# Less Can Be More: Rethinking Message-Passing for Algorithmic Alignment on Graphs

## Abstract

Most Graph Neural Networks are based on the principle of message-passing, where all neighboring nodes exchange messages with each other simultaneously. We introduce the Flood and Echo Net, a novel architecture that aligns neural computation with the principles of distributed algorithms directly on the level of message-passing. In our method, nodes sparsely activate upon receiving a message, leading to a wave-like activation pattern that traverses the entire graph. Through these sparse but parallel activations, the Net becomes provably more efficient in terms of message complexity. Moreover, the mechanism's structure to generalize across graphs of varying sizes positions it as a practical architecture for the task of graph algorithmic reasoning. We empirically validate the Flood and Echo Net improves generalization to larger graph sizes, including the SALSA-CLRS benchmark, improving graph accuracy for instances 100 times larger than during training.

## 1 Introduction

The message-passing paradigm has become the cornerstone of graph learning, with Message-Passing Neural Networks (MPNNs) emerging as a dominant framework (Gilmer et al., 2017). In these networks, nodes iteratively update their states by simultaneously exchanging messages with all neighboring nodes, providing the necessary flexibility to process arbitrary graph topologies of different sizes. Executing one message-passing round propagates information by exactly one hop. To properly exchange information throughout the entire graph, this procedure has to be performed repeatedly. As messages are sent over all edges, all nodes throughout the entire graph have to update their state after every single step. This can result in unnecessary computations, especially if the majority of the nodes do not play an active part in the computation and should maintain their current state. This phenomenon is amplified if the network is applied to larger graphs.

We propose a new execution framework, the Flood and Echo Net (FE Net). While still rooted in the message-passing paradigm, our approach employs a distinct message exchange strategy inspired by the *flooding and echo* algorithmic design pattern from the field of distributed computing. It provides a mechanism that is efficient as it exchanges fewer messages and naturally extends to graphs of larger sizes. The computation unfolds in two phases, initiated by a single origin node. First, messages are propagated away from the origin towards the rest of the graph. In this flooding part nodes only send messages to nodes that are farther away from the origin. Once all nodes have received a message, the propagation flow reverses. Now, nodes only send messages to neighbors closer to the origin, starting with the nodes that are farthest away. This process creates a wave-like activation pattern that expands equally in all directions before returning to the origin, as illustrated in Figure 1. This unique activation pattern forms the core of the FE Net offering a more structured and algorithmically aligned computation at the level of message-passing.

Compared to regular MPNNs, the FE Net offers three distinct advantages that are of interest for graph learning: improved message complexity, enhanced expressivity, and a natural way to generalize the computation to larger graph sizes. Standard MPNNs exchange information with their one-hop neighborhood in each round, sending $\mathcal{O}(m)$ messages in total along all $m$ edges. In contrast, a single phase of a FE Net also exchanges $\mathcal{O}(m)$ messages, but crucially, it updates node states with information collected throughout the graph, thus going beyond the immediate local neighborhoods.

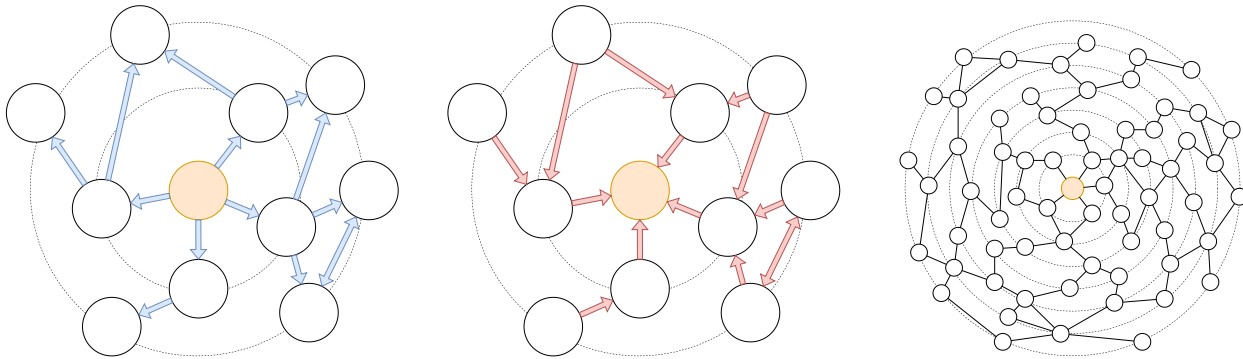

Figure 1: The FE Net propagates messages in a wave-like pattern throughout the entire graph. Starting from an origin (orange), messages are sent towards its neighbors and then continuously "flooded" outwards (blue). Once the farthest nodes are reached, the flow reverses, and messages are "echoed" back (red) toward the origin. Throughout the computation, only a small subset of nodes is active at any given time, passing messages efficiently throughout the entire graph. Moreover, the mechanism naturally generalizes to graphs of larger sizes.

Finally, as the main application of the proposed method, we study how the mechanism generalises to larger graph instances beyond what was seen during training. When MPNNs are applied to graphs of larger sizes, they have to adapt the number of rounds to retain the same relative field of perception. This results in more computation for each node. In comparison, the execution of the FE Net adapts to graphs of larger sizes more naturally as the computation inherently involves the entire graph. As a result, from a node's perspective there is less of a shift, as the computation can be done with the same amount of phases.

We hypothesize that the algorithmic alignment of the underlying mechanism makes the FE Net particularly well-suited for the challenge of graph neural algorithmic reasoning, where models must generalize learned algorithms across much larger graph sizes. Advancing the field of Neural Algorithmic Reasoning can have an impact in neural reasoning itself, but also across wider domains for downstream applications where such architectures have shown improvements in tasks related to biological vessel networks Numeroso et al. (2023), configuring networks Beurer-Kellner et al. (2022) or improving traffic engineering AlQiam et al. (2024). We test our method on a diverse set of algorithmic problems, including SALSA-CLRS - a benchmark specifically designed to evaluate scalable algorithmic reasoning on graphs. Our results demonstrate that the FE Net significantly enhances the ability to generalize learned algorithms to larger graph instances. Thus, the proposed algorithmic alignment on the level of message-passing offers a promising new direction for algorithmic reasoning on graphs.

We outline our main contributions as follows:

- We introduce the FE Net, a new execution framework aligned with principles of distributed algorithm design. The computation follows a special node activation pattern, which allows it to send fewer messages throughout the graph.

- We provide theoretical insights into the alternative computation flow, which proves that the FE Net is more efficient in terms of message complexity and more expressive than common MPNNs.

- We empirically demonstrate that the algorithmic alignment of the architecture is beneficial for size generalization in graph algorithm learning. This finding is empirically validated through extensive experiments on a variety of synthetic tasks and the SALSA-CLRS benchmark where we improve graph accuracy even for instances 100 times larger than encountered during training.

---

**Algorithm 1** Flood and Echo Net

$D \leftarrow \text{distances}(G, \text{origin})$
$\text{maxD} \leftarrow \max(D)$
$x \leftarrow \text{Encoder}(x)$
**for** $t = 1$ **to** phases **do**
    **for** $d = 1$ **to** maxD **do**
        $x[d] \leftarrow \text{FConv}^t(d - 1 \rightarrow d)$
        $x[d] \leftarrow \text{FCrossConv}^t(d \rightarrow d)$
    **end for**
    **for** $d = \text{maxD}$ **to** 1 **do**
        $x[d] \leftarrow \text{ECrossConv}^t(d \rightarrow d)$
        $x[d - 1] \leftarrow \text{EConv}^t(d \rightarrow d - 1)$
    **end for**
    $x \leftarrow \text{Update}(x)$
**end for**
$x \leftarrow \text{Decoder}(x)$

---

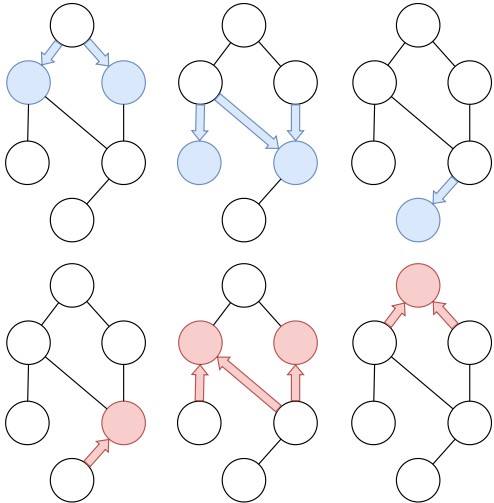

Figure 2: On the left, an algorithm describing the FE Net. First, the distances are pre-computed to activate and update the proper nodes. The convolutions $d - 1 \rightarrow d$ send messages from nodes at distance $d - 1$ to nodes at distance $d$, with only the nodes $x[d]$ at distance $d$ updating their state. On the right, an illustration of a single phase of a FE Net. At every update step, only a subset of nodes is active and changes its state. The origin is the top node of the graph, and the blue arrows depict the information flow in the flooding, while the red arrows represent the echo part. Note that a single phase activates all nodes in the graph, regardless of the graph size, while sending only a constant number of messages across each edge.

## 2 Related Work

Originally proposed by Scarselli et al. (2008) and Micheli (2009), Graph Neural Networks have seen a resurgence with applications across multiple domains (Veličković et al., 2017; Kipf & Welling, 2016; Neun et al., 2022). Notably, this line of research has gained theoretical insights through its connection to message-passing models from distributed computing (Sato et al., 2019; Loukas, 2020; Papp & Wattenhofer, 2022a). This includes strengthening existing architectures to achieve maximum expressiveness (Xu et al., 2018; Sato et al., 2021) or going beyond traditional models by changing the graph topology (Papp et al., 2021; Alon & Yahav, 2021b). In this context, multiple architectures have been investigated to combat information bottlenecks in the graph (Alon & Yahav, 2021a), i.e. using graph transformers (Rampasek et al., 2022). Similarly, higher order propagation mechanisms (Zhang et al., 2023b; Maron et al., 2020; Zhao et al., 2022), which sometimes also include distance information, have been proposed to tackle this issue or gain more expressiveness. Note that our work is orthogonal to this, as we focus on simple message-passing design on the original graph topology. In recent work, even the synchronous message-passing among all nodes has been questioned (Martinkus et al., 2023; Faber & Wattenhofer, 2023), giving rise to alternative neural graph execution models.

How GNNs can generalize across graph sizes (Yehudai et al., 2021) and their generalization capabilities for algorithmic tasks, attributed to their structurally aligned computation (Xu et al., 2020) has been of much interest. This has led to investigations into the proper alignment of parts of the architecture (Dudzik & Veličković, 2022; Engelmayer et al., 2023; Dudzik et al., 2023). A central focus has been on neural algorithmic reasoning, the study how such networks can learn to solve algorithms (Veličković et al., 2022; Ibarz et al., 2022; Minder et al., 2023; Bohde et al., 2024; Numeroso et al., 2023; Georgiev et al., 2024). Moreover, the ability to extrapolate (Xu et al., 2021) and dynamically adjust the computation in order to reason when confronted with more challenging instances remains a key aspect (Schwarzschild et al., 2021; Grötschla et al., 2022; Tang et al., 2020).

# 3 Flood and Echo Net

The fields of distributed computing and graph learning share a fundamental connection through their use of message-passing-based computation. Despite differences, the equivalence between certain models in these domains has been established (Papp & Wattenhofer, 2022a). This enables the direct translation of results such as theoretical bounds on width, number of rounds, and approximation ratios from the field of distributed computing to the study of GNNs (Sato et al., 2019; Loukas, 2020). Furthermore, it has been demonstrated that the alignment of neural network architectures with their underlying learning objectives can significantly enhance both performance and sample complexity (Xu et al., 2020; Dudzik & Veličković, 2022). This synergy between distributed computing and GNNs raises an intriguing question: can we leverage additional insights from the distributed computing community to advance graph learning? We propose the Flood and Echo Net (FE Net), a novel execution framework that directly incorporates design patterns from distributed algorithms. To illustrate the differences of our method, let us first review the conventional MPNN approach. Whenever we refer to an MPNN throughout this paper, we will refer to a GNN that operates on the original graph topology and exchanges messages in the following way:

$$a_v^t = \text{AGGREGATE}^k(\{\{x_u^t \mid u \in N(v)\}\})$$
$$x_v^{t+1} = \text{UPDATE}(x_v^t, a_v^t)$$

Where $x_v^t$ denotes the state of node $v$ in round $t$ and $N(v)$ its neighborhood. In this traditional approach, all nodes exchange messages simultaneously with all their neighbors in every round. We challenge this paradigm by taking inspiration from a design pattern called *flooding and echo* (Chang, 1982), a common building block in distributed algorithms (Kuhn et al., 2007). This pattern introduces a two-phase process: first, messages are broadcast (flooded) throughout the entire graph (Dalal & Metcalfe, 1978), and then information is gathered back (echoed) from all nodes. This approach allows for more structured and potentially more efficient information propagation encompassing the entire graph.

The Flood and Echo Net aligns its computation flow directly with the flooding and echo design pattern. The process begins at an origin node and proceeds through $T$ phases, each comprising a flooding and an echo part. Figure 2 provides a pseudo-code outline of the FE Net algorithm. Initially, nodes are partitioned based on their distance from the origin, then the $T$ phases are executed. During the flooding phase, messages propagate outward from the origin. We iterate through distances in ascending order, using two types of convolutions: FConv, which sends messages from nodes at distance $d-1$ to nodes at distance $d$. Crucially, only nodes at distance $d$ update their state (denoted as $x[d]$), and FCrossConv, which exchanges messages between nodes at the same distance $d$. In the subsequent echo phase, the message flow reverses and is echoed back towards the origin. Now we iterate through distances in descending order, again using two types of convolutions: ECrossConv for updating nodes at the same distance, and EConv for sending messages from nodes at distance $d$ to nodes at distance $d-1$, updating the latter. Note that only a subset of nodes, located at the same distance from the origin, are activated simultaneously. Therefore, FE Net can make use of a sparse but parallel activation pattern that propagates throughout the entire graph. Figure 2 provides a visual illustration of a complete phase, with colors indicating active edges and updated nodes. For a more in-depth discussion of the FE Net, including a comparison with regular MPNNs and their computation tree, we refer to Appendix B.

**Modes of Operation** The computation of the FE Net starts from an origin node. This allows for different usages of the proposed method. In the following, we outline three different strategies, which we will refer to as different modes of operations: *fixed, random* and *all*. Across all modes of operation, once the origin is chosen, the same flooding and echo parts are executed to compute node embeddings.

In the *fixed* mode, the origin is given or defined by the problem instance, i.e. by a marked source node specific to the task. In contrast, the *random* mode selects an origin uniformly at random from all nodes. In the *all* mode, we execute the FE Net once for every node. In every run, we keep only the node embedding for the chosen origin. This can be seen as a form of ego graph prediction (Zhao et al., 2021a) for each node. Although computationally more expensive, it could also be used for efficient inference on tasks where only a subset of nodes is of interest.

## 4    Theoretical Analysis

In this section, we provide a theoretical analysis of the Flood and Echo Net. While the FE Net is based on message-passing over the original graph topology, its unique propagation mechanism sets it apart from conventional message-passing GNNs. Our analysis focuses on two critical aspects: message complexity and expressiveness. We show that through the sparse activation of nodes the FE Net achieves improved efficiency in terms of message complexity. This enables it to solve tasks with significantly fewer messages than traditional MPNNs. Furthermore, we demonstrate that the FE Net not only matches but exceeds the expressiveness of regular MPNNs, surpassing the limitations of the 1-WL test. The complete proofs of these theoretical insights are contained in Appendix F.

### 4.1    Message Complexity

The FE Net significantly differs regarding the number of messages it needs to exchange in order to involve the entire graph in its computation. In standard MPNNs, a single round of message-passing updates all $n$ node states by exchanging messages over all edges. Therefore, every single round exchanges $\mathcal{O}(m)$ messages while propagating information **by exactly one hop**. Consequently, if any information needs to be propagated over a distance of $K$ hops, the total number of node updates is $\mathcal{O}(Kn)$ and the total number of exchanged messages is $\mathcal{O}(Km)$.

The FE Net, in contrast, employs a more efficient message-passing strategy. During its execution, only a subset of nodes is active during each timestep, sending messages either away from or towards the origin. This key difference results in nodes being sequentially activated, with messages passing information throughout the entire graph instead of only their immediate one-hop neighborhood. More precisely, in a single phase of a FE Net, consisting of one flooding followed by one echo part, each node is activated a constant number of times, while there are also at most a constant number of messages passed along each edge. Therefore, a single phase performs $\mathcal{O}(n)$ node updates and exchanges $\mathcal{O}(m)$ messages. Crucially, using a constant number of phases, the information can be propagated **throughout the entire graph**. Therefore, it is possible to exchange information over a distance of $K$ hops using only $\mathcal{O}(m)$ messages compared to $\mathcal{O}(Km)$ messages used by MPNNs.

**Theorem 4.1.** *There exist tasks that Flood and Echo Net can solve using $\mathcal{O}(m)$ messages, whereas no MPNN can solve them using less than $\mathcal{O}(nm)$ messages.*

As a consequence of this insight, it follows that there exist tasks that can be solved more efficiently using the FE Net. If information must be exchanged throughout the entire graph, it can be that MPNNs must use $\mathcal{O}(nm)$ messages, while a constant amount of Flood and Echo phases with $\mathcal{O}(m)$ messages each would suffice. Moreover, we will later proof in Theorem 4.2, that by simulating the execution of other MPNNs, FE Net also uses at most the same number of messages. For a more detailed discussion on the runtime and message complexity, we refer to Appendix H.

### 4.2    Expressiveness

The expressiveness of GNNs is tightly linked to the Weisfeiler-Lehman (WL) test (Leman & Weisfeiler, 1968). Most common message-passing architectures, which work on the original graph topology without higher-order message-passing, are typically bounded by the expressiveness of the 1-WL test (Papp & Wattenhofer, 2022b). First, we show that the FE Net, despite its distinct operational mechanism, not only matches the expressiveness of MPNNs but does so with at most the same number of messages:

**Theorem 4.2.** *On connected graphs, the Flood and Echo Net is at least as expressive as any MPNN . Furthermore, it exchanges at most as many messages.*

However, while MPNNs are limited by the 1-WL test, the FE Net is more expressive. Although it also exchanges messages solely on the original graph topology, the mechanism can implicitly leverage more information to distinguish more nodes through the alignment of the message propagation with the distance to the origin in the graph.

**Theorem 4.3.** *On connected graphs, Flood and Echo Net is strictly more expressive than 1-WL and, by extension, standard MPNNs.*

This enhanced expressiveness comes from the FE Net's ability to implicitly leverage additional information through its unique message propagation strategy. From a single node's perspective, the flooding and echo mechanism introduces a notion of edge "direction" relative to the origin. This allows to differentiate between edges leading towards or away from the origin (or those at equal distances). This leads to more possibilities to distinguish nodes in the local neighborhood and leverage non-local information as the wave pattern transitions through the whole graph. At the same time, the net could ignore this additional directionality information of the edges and simulate the execution of a standard MPNN. Next to these theoretical insights, we also empirically validate that the FE Net is more expressive on a variety of datasets which we include in Appendix D.

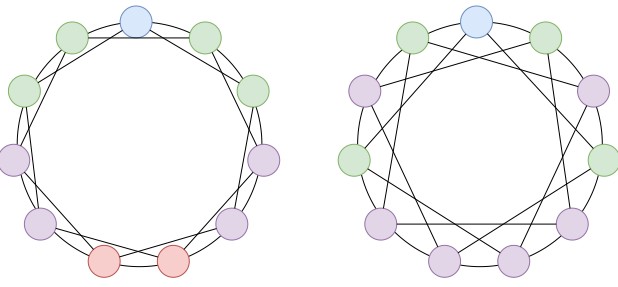

Figure 3: Example of two 4-regular graphs which cannot be distinguished using standard MPNNs as they are 1-WL equivalent. However, no matter which origin is chosen, the FE Net can easily distinguish them through the derived distance to the origin.

Note, that the FE Net's design intentionally breaks certain symmetries present in traditional MPNNs through the introduction of the origin node. This can be emulated to a certain extent with MPNNs as we show in Theorem F.1 in the Appendix. However, the FE Net still requires fewer messages and structures the mechanism for algorithmic alignment. Importantly, our theoretical results hold regardless of the FE Net operational mode. As seen in Figure 3, no matter the origin, the graphs can always be distinguished. The key insight is that the origin gives an ego perspective of the graph, similar to Identity-aware GNNs (You et al., 2021) or Subgraph GNNs (Zhao et al., 2021b). However, while these mechanisms share similar ideas, our design differs as it leverages this information implicitly and makes changes on the message-passing flow itself. While this design choice affects the equivariance properties, this symmetry breaking benefits the FE Net to be algorithmically aligned and leverage additional structural information, contributing to its enhanced expressiveness, efficiency and algorithmic alignment.

## 5 Generalization in Algorithmic Tasks

In this section, we study neural algorithmic reasoning and specifically graph algorithm learning. The concept of an algorithm is best understood as a sequence of instructions that can be applied to compute a desired output given the respective input. Algorithms have the inherent advantage to generalize across their entire domain. If we want to multiply two numbers, we can easily illustrate and explain the multiplication algorithm using small numbers. However, the same procedure generalizes, i.e. the algorithm can be used to extrapolate and multiply together much larger numbers using the same algorithmic steps. Neural algorithmic reasoning aims to grasp these underlying principles and incorporate them into machine learning architectures. The ultimate aim is to combine both domains to get models that can learn the algorithmic principles and generalize them properly, even for unseen larger inputs.

A key challenge in studying generalization is properly adapting the architecture to larger problem sizes. Without any adjustment, it might be that the amount of compute does not suffice to solve the task at hand, or in the case of graph tasks, that the required information is no longer located in the same receptive field, but is farther away. Therefore, a common strategy is to adjust the compute, or number of rounds, according to the increase of the problem size. The FE Net offers an alternative on how to adapt when generalizing to larger graphs. In fact, during a single phase, messages propagate throughout the entire graph and can therefore be updated using information beyond the immediate neighborhood.

Table 1: Extrapolation experiments on algorithmic datasets, all models were trained with graphs of size 10 and then tested on larger graphs of size 100. We compare the different Flood and Echo models against a regular GIN, which executes $L$ rounds, PGN and RecGNN, which adapts the number of rounds. The *random* mode picks a starting node at random, while the *fixed* mode starts at a predefined location. The *all* chooses each node as a start once. We report both the node accuracy with $n()$ and the graph accuracy with $g()$.

| Model | Messages | PrefixSum | | | Distance | | | Path Finding | | |
|---|---|---|---|---|---|---|---|---|---|---|
| | | n(10) | n(100) | g(100) | n(10) | n(100) | g(100) | n(10) | n(100) | g(100) |
| GIN | $\mathcal{O}(Lm)$ | $0.78 \pm 0.01$ | $0.53 \pm 0.00$ | $0.00 \pm 0.00$ | $0.97 \pm 0.01$ | $0.91 \pm 0.01$ | $0.04 \pm 0.06$ | $0.99 \pm 0.01$ | $0.70 \pm 0.05$ | $0.00 \pm 0.00$ |
| PGN | $\mathcal{O}(nm)$ | $0.94 \pm 0.12$ | $0.52 \pm 0.01$ | $0.00 \pm 0.00$ | $0.99 \pm 0.01$ | $0.89 \pm 0.01$ | $0.01 \pm 0.02$ | $1.00 \pm 0.00$ | $0.77 \pm 0.03$ | $0.00 \pm 0.00$ |
| RecGNN | $\mathcal{O}(nm)$ | $1.00 \pm 0.00$ | $0.93 \pm 0.07$ | $0.66 \pm 0.31$ | $1.00 \pm 0.00$ | $0.99 \pm 0.02$ | $0.93 \pm 0.15$ | $1.00 \pm 0.00$ | $0.95 \pm 0.04$ | $0.45 \pm 0.33$ |
| Flood and Echo *all* | $\mathcal{O}(nm)$ | $1.00 \pm 0.00$ | $\mathbf{1.00 \pm 0.01}$ | $0.96 \pm 0.07$ | $1.00 \pm 0.00$ | $0.99 \pm 0.03$ | $0.87 \pm 0.25$ | $1.00 \pm 0.00$ | $0.92 \pm 0.05$ | $0.14 \pm 0.22$ |
| Flood and Echo *random* | $\mathcal{O}(m)$ | $1.00 \pm 0.00$ | $\mathbf{1.00 \pm 0.00}$ | $0.99 \pm 0.01$ | $1.00 \pm 0.00$ | $0.97 \pm 0.04$ | $0.77 \pm 0.30$ | $1.00 \pm 0.00$ | $0.82 \pm 0.01$ | $0.01 \pm 0.00$ |
| Flood and Echo *fixed* | $\mathcal{O}(m)$ | $1.00 \pm 0.00$ | $\mathbf{1.00 \pm 0.00}$ | $\mathbf{1.00 \pm 0.00}$ | $1.00 \pm 0.00$ | $\mathbf{1.00 \pm 0.00}$ | $\mathbf{0.99 \pm 0.02}$ | $1.00 \pm 0.00$ | $\mathbf{1.00 \pm 0.00}$ | $\mathbf{1.00 \pm 0.00}$ |

Previous work has indicated that changes in the architecture or so-called "algorithmic alignment" (Engelmayer et al., 2023; Dudzik & Veličković, 2022; Xu et al., 2020) can be beneficial for learning and generalization. In our work, we propose to incorporate such **an alignment on the architectural level**, adjusting the message-passing itself to match the flooding and echo paradigm, an algorithm design pattern from distributed computing.

In the following, we empirically validate our hypothesis on a variety of tasks related to graph algorithm learning. First, we test the architecture on synthetic algorithmic tasks, which allow us both fine-grained control and theoretical insights into what is needed to solve the tasks at hand. Then, we proceed to test our method on the more challenging SALSA-CLRS benchmark, which consists of well-known graph algorithms and is specifically designed to test graph algorithms at scale. One aspect how the method differs in generalisation is that a node in an MPNNs performs more computations (as the number of rounds is increased to cover the graph). In the FE Net, the number of phases can remain unchanged. As a consequence, for a node, there is no change in the computation, even though from a graph perspective more steps are executed.

## 5.1 Algorithmic Tasks

Our initial study focuses on three algorithmic tasks PrefixSum, Distance and Path Finding adapted to the graph domain by Grötschla et al. (2022). In the Distance task, nodes have to infer their distance to a marked node modulo 2. For the Path Finding task, nodes in a tree have to predict whether they are part of the path between two given nodes. Finally, in the PrefixSum task, the cumulative sum modulo 2 has to be computed on a path graph. For a more detailed description of the datasets, we refer to Appendix K.1. Although these tasks may appear simple compared to more elaborate algorithms, their simplicity enables a rigorous analysis of the requirements to complete the task, thus providing crucial insights into the fundamental capabilities of our FE Net architecture. For a more thorough analysis of the FE Net on the PrefixSum task, including a theoretical analysis of the exchanged information, we refer to Appendix E.

**Corollary 5.1.** *Let $D$ be the diameter of the graph. In order to correctly solve the Distance, Path-finding, and PrefixSum tasks, nodes require information that is $\mathcal{O}(D)$ hops away.*

We evaluate the performance of the different FE Net modes: *fixed, random* and *all*. All modes execute two phases, which results in $\mathcal{O}(m)$ messages exchanged per chosen origin. Moreover, we choose the marked nodes in the tasks for the origin in the *fixed* mode. Note that the *all* mode, requires $n$ executions, one for each node, therefore, we only consider it for graphs of size at most one hundred. Nevertheless, the other modes can scale more easily and we believe them to be better suited for the study of algorithm learning. As a baseline comparison, we consider three models also used later on in the SALSA-CLRS evaluation. Most importantly, their architectures should be scalable to larger graph sizes and should operate on the original graph topology. We consider GIN as a representative of a maximal expressive MPNN which executes a fixed number of rounds. More precisely, five rounds are executed as the model begins to destabilize for more rounds.

**Corollary 5.2.** *Let $D$ be the diameter of the Graph. Every MPNN that correctly solves the PrefixSum, Distance, or Path Finding for all graph sizes $n$ must execute at least $\mathcal{O}(D)$ rounds and exchange $\mathcal{O}(mD)$ messages.*

Due to the above corollary, we also consider two recurrent baselines, which adapt the number of rounds according to the graph size. Therefore, we consider RecGNN Grötschla et al. (2022) and PGN (Veličković et al., 2020). We scale the number of rounds by $1.2n$, where $n$ denotes the number of nodes in the graph.

In our experimental setup, we train all models on small graphs of size 10 to assess their ability to learn underlying algorithmic patterns and then evaluate their generalization capabilities on larger graphs of size 100. From the results in Table 1, we can observe that the baseline using a fixed number of layers already struggles to fit the training data and deteriorates when tested on larger instances. Similarly, the performance of PGN drops for larger graphs. The other models exhibit better generalization, especially the node accuracy remains high. To provide a more comprehensive evaluation, we also report graph accuracy, which quantifies the proportion of graph instances where all nodes are correctly classified. This metric offers insights into the models' ability to maintain consistent performance across entire graph structures, which is required in order to solve an instance correctly in algorithmic reasoning. There, we can see that the overall model performance of the baselines drops compared to the fixed variant of FE Net. Moreover, we can test extrapolation to even larger instances of size 1000, as shown

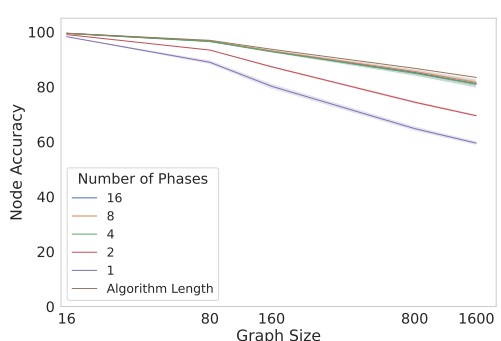

Figure 4: We illustrate node accuracy on the Dijkstra task. Adjusting the number of phases can have a positive impact on performance, both on node (Dijkstra) or graph level (MIS). All models are run on Erdős–Rényi graphs for a different amount of phases, Algorithm Length indicates that the number of phases is set equal to the given algorithm sequence length.

in the Appendix. Note that even though the node accuracy for many entries is quite high, the graph accuracy deteriorates as the graph sizes increase. The contrast between node and graph accuracies underscores a critical aspect of graph algorithm learning: while models may perform well on individual node classifications, ensuring correct and consistent performance across the entire graph becomes increasingly challenging as graphs grow in size. The FE Net 's seem to be more robust to this phenomena, especially for the fixed origin variant. This underscores that our proposed algorithmic alignment is beneficial for size generalization.

We continue our empirical evaluation of the FE Net on more challenging algorithmic tasks. We note that it is crucial to consider whether the inductive bias of the architecture aligns with the requirements of the task. Our focus lies on the study of the new algorithmic alignment of our method, especially in the context of size generalization. In many real-world graph settings such size generalization or involvement of the entire graph might not be of primary interest. In these settings, it makes more sense to utilize the standard message-passing paradigm due to its emphasis on local relations and features. As such, we leave the study how our proposed method could be adjusted or combined with existing techniques to tackle such challenges to future work. Instead, we focus on graph algorithm learning, where we expect the applications and effects of the algorithmic alignment to be directly applicable.

## 5.2 SALSA - CLRS

Building on our previous findings, where the FE Net architecture demonstrated strong generalization to larger graph instances on simple algorithmic tasks, we now extend our evaluation to more challenging and complex graph algorithms. Our goal is to test if the FE Net can face more intricate tasks that demand more sophisticated algorithmic techniques. We evaluate on the SALSA-CLRS benchmark (Minder et al., 2023), which comprises a diverse set of six graph algorithms derived from the CLRS (Veličković et al., 2022) collection.

Table 2: We evaluate the FE Net on the SALSA-CLRS benchmark, all models are trained on graphs of size 16 and then tested on larger graph sizes. We report the graph accuracy over 5 runs on Erdős–Rényi graphs of different sizes. The FE Net achieves good performance, especially on the BFS and Eccentricity task on which it exhibits strong generalization.

| Task | Model | 16 | 80 | 160 | 800 | 1600 |
|------|-------|----|----|-----|-----|------|
| BFS | FE Net | $100.0 \pm 0.0$ | $\mathbf{99.7} \pm \mathbf{0.3}$ | $\mathbf{96.6} \pm \mathbf{1.7}$ | $\mathbf{22.9} \pm \mathbf{12.5}$ | $\mathbf{4.4} \pm \mathbf{5.7}$ |
| | GIN(E) | $99.4 \pm 0.8$ | $84.3 \pm 13.9$ | $57.5 \pm 15.3$ | $2.2 \pm 4.1$ | $0.1 \pm 0.2$ |
| | PGN | $100.0 \pm 0.0$ | $88.7 \pm 5.9$ | $54.9 \pm 21.5$ | $0.2 \pm 0.1$ | $0.0 \pm 0.0$ |
| | RecGNN | $99.9 \pm 0.2$ | $87.9 \pm 8.8$ | $55.8 \pm 24.8$ | $4.6 \pm 6.5$ | $0.4 \pm 0.6$ |
| DFS | FE Net | $88.9 \pm 3.0$ | $0.0 \pm 0.0$ | $0.0 \pm 0.0$ | $0.0 \pm 0.0$ | $0.0 \pm 0.0$ |
| | GIN(E) | $0.1 \pm 0.1$ | $0.0 \pm 0.0$ | $0.0 \pm 0.0$ | $0.0 \pm 0.0$ | $0.0 \pm 0.0$ |
| | PGN | $18.4 \pm 37.7$ | $0.0 \pm 0.0$ | $0.0 \pm 0.0$ | $0.0 \pm 0.0$ | $0.0 \pm 0.0$ |
| | RecGNN | $0.0 \pm 0.0$ | $0.0 \pm 0.0$ | $0.0 \pm 0.0$ | $0.0 \pm 0.0$ | $0.0 \pm 0.0$ |
| Dijkstra | FE Net | $91.8 \pm 0.7$ | $13.2 \pm 1.7$ | $0.5 \pm 0.2$ | $0.0 \pm 0.0$ | $0.0 \pm 0.0$ |
| | GIN(E) | $73.4 \pm 2.6$ | $0.2 \pm 0.2$ | $0.0 \pm 0.0$ | $0.0 \pm 0.0$ | $0.0 \pm 0.0$ |
| | PGN | $94.6 \pm 1.1$ | $\mathbf{37.8} \pm \mathbf{6.9}$ | $\mathbf{5.2} \pm \mathbf{1.9}$ | $0.0 \pm 0.0$ | $0.0 \pm 0.0$ |
| | RecGNN | $81.7 \pm 16.1$ | $6.8 \pm 6.1$ | $0.3 \pm 0.5$ | $0.0 \pm 0.0$ | $0.0 \pm 0.0$ |

| Task | Model | 16 | 80 | 160 | 800 | 1600 |
|------|-------|----|----|-----|-----|------|
| Eccentricity | FE Net | $99.9 \pm 0.0$ | $99.9 \pm 0.1$ | $98.8 \pm 0.4$ | $99.5 \pm 0.3$ | $\mathbf{81.1} \pm \mathbf{5.4}$ |
| | GIN(E) | $57.3 \pm 21.2$ | $77.1 \pm 17.5$ | $72.3 \pm 18.0$ | $51.3 \pm 34.2$ | $36.7 \pm 17.6$ |
| | PGN | $100.0 \pm 0.0$ | $\mathbf{100.0} \pm \mathbf{0.0}$ | $\mathbf{100.0} \pm \mathbf{0.0}$ | $\mathbf{100.0} \pm \mathbf{0.0}$ | $64.6 \pm 14.9$ |
| | RecGNN | $75.8 \pm 26.2$ | $80.5 \pm 35.0$ | $75.0 \pm 39.1$ | $72.7 \pm 27.9$ | $63.0 \pm 24.8$ |
| MIS | FE Net | $98.3 \pm 0.5$ | $\mathbf{91.5} \pm \mathbf{2.4}$ | $\mathbf{83.8} \pm \mathbf{4.5}$ | $\mathbf{27.9} \pm \mathbf{12.5}$ | $\mathbf{13.9} \pm \mathbf{9.6}$ |
| | GIN(E) | $6.2 \pm 3.2$ | $0.0 \pm 0.0$ | $0.0 \pm 0.0$ | $0.0 \pm 0.0$ | $0.0 \pm 0.0$ |
| | PGN | $98.8 \pm 0.2$ | $89.2 \pm 4.6$ | $74.1 \pm 10.1$ | $10.7 \pm 10.5$ | $2.0 \pm 2.5$ |
| | RecGNN | $56.1 \pm 13.1$ | $5.5 \pm 7.1$ | $0.8 \pm 1.6$ | $0.0 \pm 0.0$ | $0.0 \pm 0.0$ |
| MST | FE Net | $58.5 \pm 4.6$ | $0.1 \pm 0.1$ | $0.0 \pm 0.0$ | $0.0 \pm 0.0$ | $0.0 \pm 0.0$ |
| | GIN(E) | $43.2 \pm 4.6$ | $0.0 \pm 0.0$ | $0.0 \pm 0.0$ | $0.0 \pm 0.0$ | $0.0 \pm 0.0$ |
| | PGN | $79.2 \pm 4.3$ | $\mathbf{2.0} \pm \mathbf{1.2}$ | $0.0 \pm 0.0$ | $0.0 \pm 0.0$ | $0.0 \pm 0.0$ |
| | RecGNN | $56.8 \pm 15.9$ | $0.6 \pm 0.8$ | $0.0 \pm 0.0$ | $0.0 \pm 0.0$ | $0.0 \pm 0.0$ |

The SALSA-CLRS benchmark is particularly relevant for our study as it emphasizes sparsity and scalability, two critical aspects in for graph algorithm learning. While it builds upon tasks from the CLRS-30 benchmark, it is important to clarify why we chose the SALSA-CLRS extension for our evaluation. The CLRS-30 collection aims to capture a wide range of algorithmic concepts, including geometry, sorting, and string tasks, not limited to graphs. To provide a unified interface for these diverse tasks, it employs an abstract graph view using fully connected graphs, enabling the modeling of relationships and reasoning between objects through generalist algorithmic reasoners and suitable architectures such as the triplet reasoner (Ibarz et al., 2022; Bohde et al., 2024). These methods heavily rely on the fully connected graph structures and use processors with higher order computations. We focus on graph algorithm learning, where the graph structure is crucial to the task and carries inherent information. A key aspect of our study is the generalization to substantially larger graphs, which extends beyond the typical 4x size increase evaluated in the CLRS framework. While CLRS-30 can accommodate larger tests, its fully connected, dense graph structure and computationally intensive baselines present scalability challenges. Moreover, applying the FE Net on fully connected graphs would be ineffective, as it relies on leveraging inherent graph structure and alignment.

In our evaluation, we use the fixed variant of the FE Net and choose the origin to match the starting node $s$ provided by the SALSA-CLRS data whenever possible, i.e. in the Dijkstra or BFS task. Otherwise, we choose the node with id 0 to be the origin. For all runs of the FE Net , unless explicitly stated otherwise, we **do not use hints** during training and execute a constant number of phases. Note that compared to the other baselines, the FE Net does not explicitly rely on being given the number of steps to be executed. All models are trained on graphs of size at most 16 and then tested on larger graph sizes. We conduct a parameter search between 1 and 16 phases for each task. In Table 2, we report the mean graph accuracy and standard deviation across 5 runs. We report the baseline performances from Minder et al. (2023). For further details on the technical setup, we refer to Appendix J. The FE Net achieves good performance across the algorithms, improving graph accuracy even to the largest graphs 100 times larger than during training. Most notably, the BFS and Eccentricity task can benefit from the algorithmic alignment. This is further underlined for the BFS task, where FE Net achieves almost perfect scores on graphs up to size 160, while the baselines already experience a significant drop off. To further investigate the impact of number of phases, we run an additional ablation on the Dijkstra task illustrated in Figure 4 and find that the performance increases when the number of phases is increased. For the complete results, we refer to Appendix J which also include tests on a variety of different graph distributions.

Overall, the FE Net architecture demonstrates strong performance on SALSA-CLRS. It achieves the best scores in 9 out of 15 extrapolation settings where non-trivial performance is achieved, while being within a percent in another 3. For some algorithms, we observe enhanced performance when increasing the number of executed phases, suggesting a potential adaptation strategy for tasks less naturally aligned with the flooding and echo paradigm. Remarkably, even without relying on predetermined step counts or intermediate hints during training, our method can achieve superior results on multiple tasks. This improvement extends to graph accuracy, underscoring the FE Net's capacity to enhance generalization of graphs 100 times

larger. These findings highlight that the algorithmic alignment on the level of message-passing benefits graph algorithm learning.

## 6 Conclusion

In this work, we challenge the standard message-passing paradigm commonly used in graph learning and introduce the Flood and Echo Net. Our method aligns its execution to send fewer messages throughout the entire graph in a wave-like activation. This improves message complexity, as it can facilitate messages throughout the entire graph more easily and can also increase expressivity. Crucially, the execution of the FE Net naturally generalizes to graphs of larger sizes, which we find to be beneficial in improving generalization in graph algorithm learning. We empirically validate our findings on simple algorithmic tasks as well as more challenging graph algorithms from the SALSA-CLRS benchmark. Our results demonstrate that the algorithmic alignment of the FE Net significantly enhances performance on multiple algorithms even in the challenging graph accuracy, particularly when generalizing to larger graphs where it improves results on extrapolation on graphs 100 times larger. These findings underscore our method's potential to improve performance through algorithmic alignment on the level of message-passing.

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

## A  Appendix

## B  Flood and Echo Net Definition

*Remark:* In the main part of the paper there is a pseudo algorithm which outlines the computation using for loops. We recommend the code view (or source code) for a more intuitive understanding as the formal definition can seem quite complex.

Let $r$ be the origin of the computation phase and let $d(v)$ denote the shortest path distance from $v$ to $r$. Then, the update rule for of the FE Net looks is defined as follows, assume $T$ phases are executed. At the beginning of each phase $t$, the flooding is performed, where the nodes are sequentially activated one after another depending on their distance towards the root. Each convolution is either from nodes at distance $d$ to $d+1$ (flood), from $d+1$ to $d$ (echo) or between nodes at the same distance (floodcross, echocross). The term $x[d]$ denotes that only nodes at distance $d$ update their state. The variable $x_v^{d,t}$ denotes the state of node $v$ after $t$ phases and $d$ updates within that phase. Unless updated in the formula, we have that $x_v^{d,t} = x_v^{d-1,t}$ and $x_v^{0,0}$ are the initial features. For each distance $d$ from 1 to the max distance $D$ in the graph the following update is performed:

$$\textbf{for each } d = 1...D\textbf{:}$$
$$f_v^{d,t} = \text{AGGREGATE}_{Flood}^t(\{\{x_u^{d-1,t} \mid d(u) = d - 1, u \in N(v)\}\})$$
$$fupd_v^{d,t}[d] = \text{UPDATE}_{Flood}^t(x_v^{d-1,t}, f_v^{d,t})$$
$$fc_v^{d,t} = \text{AGGREGATE}_{FloodCross}^t(\{\{fupd_v^{d,t} \mid d(u) = d, u \in N(v)\}\})$$
$$x_v^{d,t}[d] = \text{UPDATE}_{FloodCross}^t(x_v^{d-1,t}, fc_v^{d,t})$$

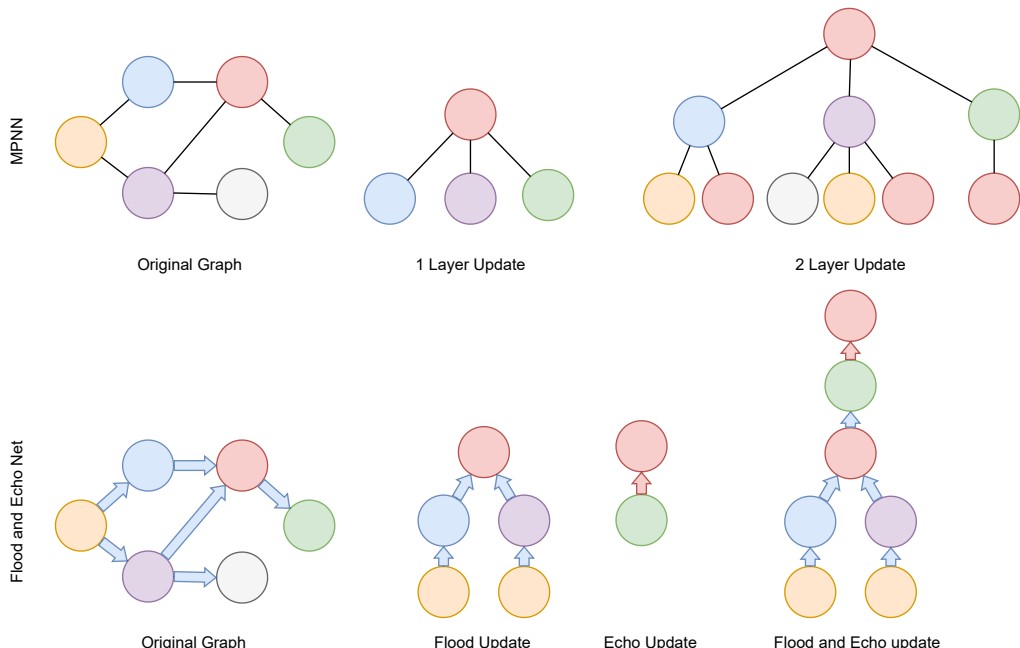

Figure 5: Visualization of the computation executed on the same graph for a regular MPNN and a FE Net from the perspective of the red node. The top row shows the computation for regular MPNN both for 1 and 2 layers of message-passing. Note that executing $l$ layers takes into account the $l$-Hop neighborhood. On the bottom row, the computation from the perspective of the red node in a FE Net is shown. Note that the origin of the FE Net is the orange node. The two middle figures illustrate the updates in the flood and the echo part respectively. Furthermore, the figure on the right shows the combined computation for an entire phase.

And similarly for each distance $d$ from max distance -1 to 0 the Echo phase

**for each** $d = 1...D$**:**, $let\ d' = D - d$

$$ec_v^{d,t} = \text{AGGREGATE}_{EchoCross}^t(\{\{x_u^{D+d-1,t} \mid d(u) = d' + 1, u \in N(v)\}\})$$
$$eupd_v^{d,t+1}[d] = \text{UPDATE}_{EchoCross}^t(x_v^{D+d-1,t}, ec_v^{d,t})$$
$$e_v^{d,t} = \text{AGGREGATE}_{Echo}^t(\{\{eupd_v^{d,t} \mid d(u) = d', u \in N(v)\}\})$$
$$x_v^{D+d,t}[d] = \text{UPDATE}_{Echo}^t(x_v^{D+d-1,t}, e_v^{d,t})$$

The phase is completed after another update for all nodes.

$$x_v^{0,t+1} = \text{UPDATE}^t(x_v^{2D,t})$$

Note that the node activations are done in a sparse way, therefore, for all updates that take an empty neighborhood set as the second argument no update is performed and the state is maintained. Furthermore, in practise we did not find a significant difference in performing the last update step, which is why in the implementation we do not include it. In Figure 5 we outline the differences between the computation of an MPNN and a FE Net.

## C    Extended Related Work

A variety of GNNs that do not follow the 1 hop neighborhood aggregation scheme have been unified under the view of so-called Subgraph GNNs. The work of Zhang et al. (2023a) analyses these models in terms of their expressiveness and gives the following general definition:

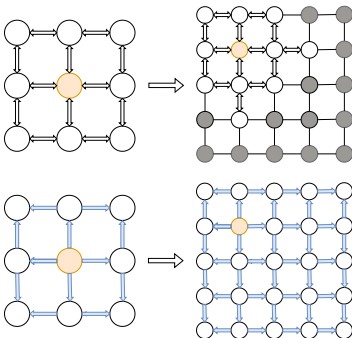

Figure 6: The top row shows a standard MPNN from the perspective of a specific node (orange) with 2 rounds of message-passing. The arrows denote messages. When applied to larger graphs, the same model will have nodes (gray) that lie outside of the receptive field and cannot be part of the computation. The bottom row shows the flooding phase of an FE Net . It generalises naturally to larger graphs, involving the entire graph in its computation. Both models send the same number of messages in their computations.

**Definition C.1.** A general subgraph GNN layer has the form

$$h_G^{(l+1)}(u, v) = \sigma^{(l+1)}(\mathsf{op}_1(u, v, G, h_G^{(l)}), \cdots, \mathsf{op}_r(u, v, G, h_G^{(l)})),$$

where $\sigma^{(l+1)}$ is an arbitrary (parameterized) continuous function, and each atomic operation $\mathsf{op}_i(u, v, G, h)$ can take any of the following expressions:

- Single-point: $h(u, v)$, $h(v, u)$, $h(u, u)$, or $h(v, v)$;
- Global: $\sum_{w \in \mathcal{V}_G} h(u, w)$ or $\sum_{w \in \mathcal{V}_G} h(w, v)$;
- Local: $\sum_{w \in \mathcal{N}_{G^u}(v)} h(u, w)$ or $\sum_{w \in \mathcal{N}_{G^v}(u)} h(w, v)$.

We assume that $h(u, v)$ is always present in some $\mathsf{op}_i$.

This allows us to capture a more general class of Graph Neural Networks, i.e., the work of Zhang et al. (2023b), which can incorporate distance information into the aggregation mechanism this way. Note that the proposed mechanism of the FE Net differs from that of this particular notion of subgraph GNNs. At each update step, only a subset of nodes is active. This allows nodes to take into account nodes that are activated earlier, which is not directly comparable to subgraph GNNs where the node updates still happen simultaneously for the nodes in question.

Another important issue that GNNs often struggle with is the so-called phenomenon of oversquashing (Alon & Yahav, 2021a). In simple terms, if too much information has to be propagated through the graph using a few edges, a bottleneck occurs, squashing the relevant information together, leading to information loss and subsequent problems for learning. Recent work of (Giovanni et al., 2023) theoretically analyses the reasons leading to the oversquashing phenomena and identifies the width and depth of the network but also the graph topology as key contributors. Note that the proposed FE Net is not designed to tackle the problem of oversquashing. Rather, it tries to facilitate information throughout the graph, assuming that there is no inherent (topological) bottleneck. It only affects the aforementioned depth aspect of the network. However, as outlined by (Giovanni et al., 2023), the depth is likely to have a marginal effect compared to the graph topology.

The works of Martinkus et al. (2023), namely AgentNet, and Faber & Wattenhofer (2023), who proposes AMP (Asynchronous Message-Passing), also draw inspiration from the field of distributed computing. Although they share some aspects in their mechanisms, their respective settings differ quite a bit. In AgentNet, there exist agents which traverse the graph which gives them the possibility to solve problems on the graph in sublinear time. In contrast, our approach tries to enable communication throughout the whole graph, especially in the context of different graph sizes. On the other hand, AMP activates nodes one at a time, benefiting from a similar computational sparsity as our method. However, note that the FE Net's execution is more structured. On one side, this leads to less flexible activation patterns, however, on the other hand, it

Table 3: As the theory predicts, the GIN model cannot go beyond trivial performance. Whereas both the *single* and *all* execution mode go beyond the limits of 1-WL. Note, that the datasets are imbalanced and can contain multiple components, which can explain the performance of GIN and the account for the drop of the single mode compared to the all execution.

| Model | GIN | | Flood and Echo *single* | | Flood and Echo *all* | |
|---|---|---|---|---|---|---|
| | Train | Test | Train | Test | Train | Test |
| Triangles | $0.80 \pm 0.00$ | $0.78 \pm 0.00$ | $0.92 \pm 0.00$ | $0.92 \pm 0.00$ | $1.00 \pm 0.00$ | $1.00 \pm 0.00$ |
| LCC | $0.79 \pm 0.00$ | $0.79 \pm 0.00$ | $0.92 \pm 0.00$ | $0.91 \pm 0.00$ | $1.00 \pm 0.00$ | $1.00 \pm 0.00$ |
| 4-Cycles | $0.49 \pm 0.02$ | $0.50 \pm 0.00$ | $0.95 \pm 0.01$ | $0.95 \pm 0.02$ | $1.00 \pm 0.00$ | $0.96 \pm 0.02$ |
| Limits-1 | $0.50 \pm 0.00$ | $0.50 \pm 0.00$ | $0.70 \pm 0.06$ | $0.80 \pm 0.27$ | $1.00 \pm 0.00$ | $1.00 \pm 0.00$ |
| Limits-2 | $0.50 \pm 0.00$ | $0.50 \pm 0.00$ | $0.79 \pm 0.05$ | $0.90 \pm 0.22$ | $1.00 \pm 0.00$ | $1.00 \pm 0.00$ |
| Skip-Circles | $0.10 \pm 0.00$ | $0.10 \pm 0.00$ | $1.00 \pm 0.00$ | $1.00 \pm 0.00$ | $1.00 \pm 0.00$ | $1.00 \pm 0.00$ |
| Messages | $\mathcal{O}(Lm)$ | | $\mathcal{O}(m)$ | | $\mathcal{O}(nm)$ | |

translates naturally across graph sizes. Whereas AMP has to additionally learn a termination criteria which must generalize.

## D  1-WL Expressive Experiments

We empirically validate our findings for the FE Net on multiple expressive datasets that go beyond 1-WL. The tasks span both graph and node predictions, which include graphs that have multiple disconnected components. We test two modes on these datasets. One variant performs an execution from a single node using the random variant, while the other performs the all mode. Both modes compute node embeddings and can be used for the node prediction tasks without modification. Whereas for graph prediction tasks, the sum of all node class predictions is used for the final graph prediction. Note that the second variant is fairer for comparison against MPNNs, since for some datasets like Limits-1, Limits-2, and 4-Cycles, the graph is not connected. Therefore, the single start mode struggles, as it cannot access all components.

In Table 3 we can see that the Flood and Echo all starts manages to almost perfectly solve all tasks. The single start performs worse in the Limits-1 and Limits-2 due to the lack of access to all components. The GIN model, as predicted by theory, performs no better than random guessing. The higher scores in the Triangles and LCC datasets are due to the fact that these datasets are imbalanced. For an in-depth explanation of the individual datasets, we refer to Appendix K.2. Comparing the message complexities of the different methods, a GIN with $L$ layers exchanges $\mathcal{O}(Lm)$ messages while the Flood and Echo model either exchanges $\mathcal{O}(m)$ or $\mathcal{O}(nm)$ messages based on whether it executes the single or all starts mode.

## E  Information Propagation

In this section, we analyze the ability of the FE Net to distribute the available information throughout the whole graph. We use a synthetic algorithmic dataset, the PrefixSum task. For this task, we can provably determine what pieces of information must be gathered for each node to make correct predictions. If we choose an appropriate origin point, we could easily send the information and solve the task. However, more interestingly, what happens if we choose a random origin node instead? Can the Flood and Echo model still distribute the relevant information, even if it does not suffice to fully solve the task? We derive theoretical upper bounds for the best-performing instance given the information that theoretically could be available during the execution depending on the number of origin nodes. Interestingly, even if the full information is not available, the FE Net achieves performance that closely follows the theoretical upper bound. This showcases the ability of our proposed method to distribute all available information throughout the whole graph.

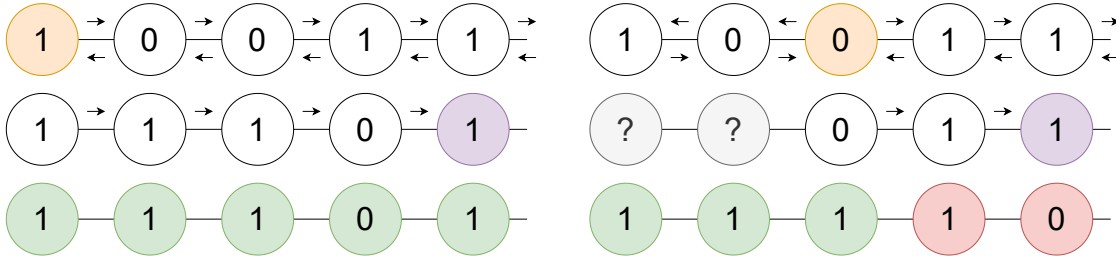

Figure 7: Visualization of the information exchange in the PrefixSum task when choosing different origin nodes for FE Net. We can derive theoretical upper bounds for the performance of FE Net depending on the number of random origin nodes for a single phase. We show that the empirical performance closely follows the theoretical analysis. This confirms the ability of the FE Net to distribute the available information throughout the whole graph.

Table 4: Information propagation of the FE Net for graphs of size $n$ on the PrefixSum task. As the number of random origin points $s$ increases, the model can distribute the additional information, as seen by the increase in accuracy. Moreover, it can do so very effectively as the performance closely follows the theoretical upper bound.

| Model | $n = 10$ | | | | $n = 100$ | | | |
|---|---|---|---|---|---|---|---|---|
| | $s = 1$ | $s = 2$ | $s = 3$ | $s = 5$ | $s = 1$ | $s = 2$ | $s = 3$ | $s = 5$ |
| THEORETICAL UPPER BOUND | 82.00 | 89.80 | 93.52 | 96.91 | 75.75 | 84.07 | 88.23 | 92.39 |
| FLOOD AND ECHO | 81.69 ± 0.51 | 88.10 ± 2.34 | 89.99 ± 0.28 | 93.90 ± 0.23 | 75.39 ± 0.29 | 83.43 ± 0.44 | 87.79 ± 0.34 | 91.86 ± 0.28 |

**PrefixSum Task**  For this analysis, we use the PrefixSum dataset, which follows the task introduced by Schwarzschild et al. (2021) and was later adapted for the graph setting (Grötschla et al., 2022). It consists of a path graph, where one end is marked to distinguish left form right. Each node $v$ independently and uniformly at random gets assigned one bit $x_v$, which is either 1 or 0, chosen with probability $\frac{1}{2}$ each. The task is to compute the prefix sum from left to right modulo 2. Therefore, the output $y_v$ of each node $v$ is the sum of the bits of all nodes to the left $y_v \equiv_2 \left(\sum_{i \leq v} x_i\right)$. Note, that in order to correctly predict a node output, it has to take all bits left of it into consideration.

**Lemma E.1.** *In the PrefixSum task, for every node $v$, the computation of the output $o_v$ must be dependent on all bits of the nodes to its left. If not all bits are considered for the computation, the probability of a correct prediction is bounded by $\Pr[o_v = y_v] \leq \frac{1}{2}$*

Note that from this lemma, it immediately follows that to solve the task correctly, information needs to be exchanged throughout the whole graph. Nodes towards the end of the path must consider almost all nodes of the graph for their computation.

**Corollary E.2.** *The PrefixSum task requires information of nodes that are $\mathcal{O}(D)$ hops apart and therefore must exchange information throughout the entire graph.*

From Lemma E.1, we know that nodes can only correctly predict their output if the information of all nodes left to them is taken into account. Whenever the initial origin of the FE Net is chosen at one of the ends, this information should be available in either the flooding or echo part. However, what happens if we choose one of the nodes in the graph at random to be the origin? Then, there will always be a right side whose predictions are dependent on the computation of the left, which has not yet been exchanged. An example is depicted in Figure 7. The top row indicates the origin node (orange) and illustrates the message exchange in the flooding (top arrows) and echo phase (bottom arrows). The middle row indicates what parts of the graph the purple-marked node can know about after a single phase. Note that on the right-hand side, it cannot infer the initial features of the two leftmost nodes. Because of the missing information, the configuration on the right can only correctly predict the nodes up to the initial origin node.

We can formally derive a theoretical upper bound for the expected number of correctly predicted nodes depending on $n$, the number of nodes, and $s$, the number of origins. For the entire derivation and formula, we refer to the Appendix.

In Table 4, we can compare the empirical performance of the FE Net with the theoretical upper bound. Moreover, the measured performance closely follows the theoretical upper bound. The experiment clearly shows that the accuracy of the model strictly increases when more starting nodes are chosen. This indicates, that the model can make use of the additional provided information. Therefore, it can effectively incorporate the information and propagate it in a sensible way throughout the graph.

## F  Proofs and Derivations

Derivation of $\mathbb{E}[X]$:
Let us assume $s$ starting nodes are chosen uniformly at random and $s_j$ denote the index of the $j$-th starting nodes. If the beginning is chosen, then all nodes could be classified correctly. Otherwise, nodes can only be correctly inferred up to $t = \max_j s_j$, the starting node farthest to the right. Moreover, the rest of the $n - t$ nodes can only be guessed correctly with probability $\frac{1}{2}$ as the cumulative sum to the left is missingWe can derive the closed-form solution for $X$, the expected number of correctly predicted nodes for a perfect solution.

$$\mathbb{E}[X] = \Pr[\min_j s_j = 1]n + (1 - \Pr[\min_j s_j = 1]) \sum_{i=2}^{n} \frac{n + \max_j s_j}{2} \Pr[\max_j s_j = i]$$

$$= \left(1 - \left(\frac{n-1}{n}\right)^s\right) n + \left(\frac{n-1}{n}\right)^s \sum_{i=2}^{n} \frac{n+i}{2} (\Pr[\max_j s_j < i+1] - \Pr[\max_j s_j < i])$$

$$= \left(1 - \left(\frac{n-1}{n}\right)^s\right) n + \left(\frac{n-1}{n}\right)^s \sum_{i=2}^{n} \frac{n+i}{2} \left(\left(\frac{i-1}{n-1}\right)^s - \left(\frac{i-2}{n-1}\right)^s\right)$$

*Proof of Theorem 4.2.* It has been shown by the work of Xu et al. (2018) that the *Graph Isomorphism Network* (GIN) achieves maximum expressiveness amongst MPNN. In the following, we will show that a FE Net can simulate the execution of a GIN on connected graphs, therefore matching it in its expressive power. Let $G_I$ be a GIN using a node state vector $h_v^k$ of dimension $d_i$.

$$h_v^{(k)} = \text{MLP}^{(k)}\left((1+\epsilon)h_v^{(k-1)} + \sum_{u \in \mathcal{N}(v)} h_u^{(k-1)}\right)$$

Let $G_F$ be a FE Net using node state vector $q_v^{(k)}$ of dimension $d_f = 2 \cdot d_i$. We partition the vector $q_v^{(k)} = o_v^{(k)} \,\|\, n_v^{(k)}$ into two vectors of dimension $d_i$. Initially, we assume that the encoder gives us $o_v^{(0)} = h_v^{(0)}$ and $n_v = 0^{d_i}$ the zero vector. We now define the updates of flood, floodcross, echo, and echocross in a special way, that after the flood and echo part $o_v^{(k)}$ is equal to $h_v^{(k)}$ and $n_v^{(k)}$ is equal to $\sum_{u \in \mathcal{N}(v)} h_u^{(k-1)}$. If this is ensured, the final update in a flood and echo phase can update $q_v^{(k)} = \text{MLP}^{(k)}((1+\epsilon)o_v^{(k-1)} + n_v^{(k-1)}) \,\|\, 0^{d_i}$, which exactly mimics the GIN update. It is easy to verify that if we set the echo and flood updates to add the full sum of the $o_v^{(k)}$ part of the incoming messages (and similarly half of the sum of the incoming messages during the cross updates) to $n_v^{(k-1)}$ the desired property is fulfilled. Moreover, there are at most four messages exchanged over each edge of the graph. Specifically, four is for cross edges and two is for all other edges. Therefore, a total of $\mathcal{O}(m)$ messages are exchanged, which is asymptotically the same number of messages GIN exchanges in a single update step. This enables a single phase of the FE Net to mimic the execution of a single GIN round. Repeating this process the whole GIN computation can be simulated by the FE Net.

Therefore, given a GIN network $G_I$ of width $d_i$, we can construct a FE Net $G_F$ of width $\mathcal{O}(d)$ that can simulate one round of $G_I$ in a single flood and echo phase using $\mathcal{O}(m)$ messages.

$\square$

*Proof of Theorem 4.3.* To show that the FE Net goes beyond 1-WL, it suffices to find two different graphs that are equivalent under the 1-WL test but can be distinguished by a FE Net. Observe that a FE Net can calculate its distance, in number of hops, to the root for each node. See the graphs illustrated in Figure 3 for a comparison. On the left is a cycle with 11 nodes, which have additional connections to the nodes that are at distance two away. Similarly, the graph on the right has additional connections at a distance of three. Both graphs are four regular and can, therefore, not be distinguished using the 1-WL test. However, no matter where the starting node for Flood and Echo is placed, it can distinguish that there are nodes which have distance four to the starting root in one graph, which is not the case in the other graph. Therefore, FE Net can distinguish the two graphs and is more expressive than the 1-WL test. Moreover, due to the Theorem 4.2 it matches the expressiveness of the 1-WL test on connected graphs by a reduction to the graph isomorphism network. $\square$

**Theorem F.1.** *On connected graphs, an MPNN that is given a uniquely marked node $r$ in the graph and a sufficient number of rounds is as expressive as a Flood and Echo Net with origin $r$.*

*Proof of Theorem F.1.* We assume we have an MPNN that operates on a connected Graph $G$ and executes a sufficient number of rounds $L$. Moreover, the graph has one node $q$ which is uniquely differentiated from the rest of the nodes. This node will act as the origin node. Assume for now that each node knows its distance to node $q$, the maximal distance $d$ of any node to $q$ and the overall number of rounds that has already passed $l'$. If this were the case, the MPNN could simulate the FE Net by appropriately matching the procedure outlined in the pseudocode of Figure 2. Each round would correspond to one of the convolutions, which can be done as each node knows $l', d$ and its own distance to properly emulate the corresponding computation. This simulation can be done as long as $L = \mathcal{O}(Td)$, where $T$ denotes the number of phases of the corresponding FE Net .

Each node can easily keep track of $l'$ during its execution. Furthermore, an MPNN could derive the distances and $d$ as follows. Distances are iteratively updated, the marked node marks itself as distance 0. All other nodes will update their own distance to be the minimum distance of their neighbors plus one. If a node only has neighbors that are of smaller distance, it will send a "return" message to its neighbors containing its distance. Once a node has received such a return message from all its nodes with a higher distance it forwards the maximum of these distances as a return message itself. The maximum can be easily determined in the aggregation, however, it requires an additional (in-between) round where nodes communicate their distance and if they have already sent such a return message. As the nodes only receive the multiset of messages we have to do such a check which doubles the number of rounds. Once the node $q$ has received a return message from each of its neighbors it can determine $d$. It will send an "initiate" message (which will be forwarded to all nodes), that in $c \cdot d$ steps, the simulation of the FE Net will begin, where $c$ is an appropriate constant so all nodes will receive the initiate message. Therefore all nodes know their own distance, $d$ and $l'$ and can simulate the process as outlined above.

Because the MPNN can simulate the FE Net computation given the described assumptions, it will be at least as expressive as the FE Net . Further, we already know that the FE Net can simulate the computation of any MPNN, which includes the described circumstances. Thus, they are equally expressive, given the uniquely marked node and sufficient number of rounds. $\square$

*Proof of Lemma 4.1.* Consider either one of the PrefixSum, Distance, or Path Finding tasks presented in Appendix K.1. All of them require information that is $\mathcal{O}(D)$ apart and must be exchanged. It follows that all MPNNs must execute at least $\mathcal{O}(D)$ rounds of message-passing to facilitate this information. Moreover, in these graphs, the graph diameter can be $\mathcal{O}(n)$. As in each round, there are $\mathcal{O}(m)$ messages exchanged, MPNNs must use at least $\mathcal{O}(nm)$ messages to solve these tasks. Furthermore, from Lemma F.2, it follows that FE Net can solve the task in a single phase using $\mathcal{O}(m)$ messages. $\square$

*Proof of Lemma E.1.* For the sake of contradiction, assume that not all bits of the nodes to the left have to be considered for the computation. Therefore, at least one bit at a node $u$ exists, which is not considered for the computation of $o_v$. We know that all bits $x$ are drawn uniformly at random and are independent of each other. Furthermore, we can rewrite the groundtruth $y_v \equiv_2 \sum_{i \leq v} x_i \equiv_2 x_u + \sum_{i \leq v, i \neq u} x_i \equiv_2 x_u + s$ as the sum of $x_u$ and the rest of the nodes. From there, it follows that the ground truth is dependent on $x_u$, even if all other bits are known $\Pr[y_v = 0 \mid s] = \Pr[s = x_u] = \frac{1}{2}$. On the other hand, we know that $o_v$ must be completely determined by the information of the nodes that make up $s$ and cannot change depending on $x_u$. Therefore, $\Pr[o_v = y_v \mid o_v \text{ does not consider } x_u] \leq \frac{1}{2}$. $\square$

*Proof of Corollary E.2.* According to Lemma E.1, for each node $v$ to derive the correct prediction, all $x_u$ for nodes $u$ that are left of $v$ have to be considered. Therefore, look at the node $r$ on the very right end of the path graph. It has to take the bits of all other nodes into consideration. However, the leftmost bit at node $l$ is $n-1$ hops away, which is also the diameter of the graph. Therefore, in order to solve the PrefixSum task, information has to be exchanged throughout the entire graph by propagating it for at least $\mathcal{O}(D)$ hops. $\square$

*Proof of Corollary 5.1.* For the task PrefixSum, the statement follows from E.2. For the other tasks we outline the proof as follows: Assume for the sake of contradiction that this is not the case and only information has to be exchanged, which is $d' = o(D)$ hops away to solve the task. Therefore, as both tasks are node prediction tasks, the output of each node is defined by its $d'$-hop neighborhood. For both tasks, we construct a star-like graph $G$, which consists of a center node $c$ and $k$ paths of length $\frac{n}{k}$, which are connected to $c$ for a constant $k$. For the Path Finding task, let the center $c$ be one marked node, and the end of path $j$ be the other marked node. Consider the nodes $x_i$, $i = 1, 2, ..., k$ which lie on the $i$-th path at distance $\frac{n}{2k}$ from $c$. Note that all $x_i$ are $\frac{n}{2k}$ away from both their end of the path and $c$ the root. Moreover, the diameter of the graph is $\frac{2n}{k}$. This means that neither the end of the $i$-th path nor the center $c$ will ever be part of the $d'$hop neighborhood. Therefore, if we can only consider the $d'$-hop neighborhood for each $x_i$, they are all the same and as a consequence will predict the same solution. However, $x_j$ lies on the path between the marked nodes while the other $x_i$'s do not. So they should have different solutions, a contradiction. A similar argument holds for the Distance task. Again let $c$ be the marked node in the graph and $x_i$ for $i = 1, 2, ..., k$ be the nodes which lie on the $i$-th path at distance $\frac{n}{2k}$ for even $i$ and $\frac{n}{2k} + 1$ for odd $i$. Again, note that the $d'$-hop neighborhood of all $x_i$ is identical and therefore must compute the same solution. However, the solution of even $x_i$ should be different from the odd $x_i$, a contradiction. $\square$

*Proof of Corollary 5.2.* From Corollary 5.1 and E.2 it directly follows that information must be exchanged for at least $\mathcal{O}(D)$ hops to infer a correct solution. As MPNNs only exchange information one hop and exchange $\mathcal{O}(m)$ messages per round, the claim follows immediately. $\square$

**Lemma F.2.** *FE Net can facilitate the required information for the PrefixSum, Distance and Path Finding task in a single phase, which can be executed using $\mathcal{O}(m)$ messages.*

*Proof of Lemma F.2.* We will prove that in all three mentioned tasks, there exists a configuration for a Flood and Echo phase, which can propagate the necessary information throughout the graph in a single phase. Let the origin $s$ correspond to the marked node in the graph, or in the case of the Path Finding, any of the two suffices. First, we consider the PrefixSum task. Note that in the flooding phase, information is propagated from the start, which is the left end, towards the right. Therefore, in principle, each bit can be propagated to the right, which suffices to solve the task according to E.1. For the Distance task, it is necessary that the length of the shortest path between the root and each node can be inferred. Note that this is exactly the path which is taken by the flooding messages, therefore, this should be sufficient to solve the task. Similarly, for the Path Finding task, one phase is sufficient. Note that starting from the leaves of the graph during the echo phase, nodes can decide that they are not part of the path between the two marked nodes (as only marked leaves can be part of the path). However, when such a message is received at one of the marked nodes, they can ignore it and tell their predecessor that they are on the path. This is correct, as one of the marked ends is at the start of our computation, and this echo message travels from the other marked end on the to-be-marked path back toward the root. This shows that for each of the above-mentioned tasks, there exists a FE Net configuration that solves the task in a single phase, which exchanges $\mathcal{O}(m)$ messages. $\square$

## G   Model Architecture and Training

The following describes the setup of our experiments for PrefixSum, Path-Finding and Distance. We use a GRUMLP convolution for all Flood and Echo models and the RecGNN, which is defined in equation 1. It concatenates both endpoints of an edge for its message and passes it into a GRU cell (Cho et al., 2014). All models use a hidden node state of 32. We use a multilayer perceptron with a hidden dimension 4 times the input dimension and map back to the hidden node state. Further, we use LayerNorm introduced by (Ba et al., 2016). We also adapt the PGN for the experiments following the implementation by Minder et al. (2023). We concatenate the current, last and original input in each step and also adapt the number of rounds to be linear in the graphs size by executing $1.2n$ rounds. For the expressiveness tasks, we perform one phase of Flood and Echo to compute our node embeddings, while for the algorithmic tasks, we perform two phases of Flood and Echo. We run for a maximum of 200 epochs, but do an early stop whenever the validation loss does not increase for 25 epochs. We use the Adam optimizer with an initial learning rate of

$$x_v^{t+1} = \text{GRUCell}\left(x_v^t, \sum_{u \in N(v)} \phi(x_v^t || x_u^t)\right) \tag{1}$$

In all our experiments, we train our model using the ADAM optimizer Kingma & Ba (2015) with a learning rate of $4 \cdot 10^{-4}$ and batch size of 32 for 200 epochs. We also use a learning rate scheduler where we decay the learning rate with patience of 3 epochs and perform early stopping if the validation loss does not decrease for more than 25 epochs. All reported values are reported over the mean of 5 runs.

The model is implemented in pytorch lightning using the pyg library, and the code will be made public upon publication.

## H   Runtime

### H.1   Runtime Complexity

We denote $n$ the number of nodes, $m$ the number of edges and $D$ the diameter of the graph. Furthermore, let $T$ be the number of phases for a FE Net and $L$ be the number of layers for an MPNN.

A single round of regular message-passing exchanges $\mathcal{O}(m)$ messages. Therefore, executing $L$ such rounds results in $\mathcal{O}(L)$ steps and $\mathcal{O}(Lm)$ messages. Note that in order for communication between any two nodes $L$ has to be in the order of $\mathcal{O}(D)$.

A single phase of a FE Net, consisting of one starting node, exchanges $\mathcal{O}(m)$ messages and does so in $\mathcal{O}(D)$ steps. Therefore, executing $T$ phases of a FE Net results in $\mathcal{O}(Tm)$ messages exchanged in $\mathcal{O}(TD)$ steps. Note, that it is sufficient for $T$ to be constant $\mathcal{O}(1)$ in order to communicate throughout the whole graph and does not necessarily have to be scaled according to the size of the graph.

The variations *fixed* and *random* perform their executions only for a specific single node. Contrary, the *all* variation performs such an execution for each of its nodes individually. Therefore, both the number of messages and the number of steps is increased by a factor of $n$, resulting in a total of $\mathcal{O}(Tnm)$ messages.

### H.2   Measurements

The FE Net is implemented using PyTorch Lightning and PyG, the code will be made publicly available upon acceptance. The flood and echo are implemented in such a way, that they make use of the GPU operations provided by PyTorch Geometric by masking out the non-relevant messages. This precomputation implemented through message-passing on the GPU as well. In Table H we measure the execution of the forward pass of all models on the PrefixSum task for graphs of size $10, 20, 50,$ and $100$. Each run consists of a 1000 graphs for which we report the mean execution time per graph and the standard deviation.

Table A: Runtime on the SALSA-CLRS benchmark on graphs of size 1600. Time reported is mean time per graph in [ms] over the entire test set. The FE Net (FE-1) performs a single phase, which generalizes across different graph sizes. Whereas the baseline (PGN) has to rely on the number of steps dictated by the ground truth algorithm. Across BFS, MIS and Eccentricity the runtime is very comparable, for DFS, Dijkstra and MST it is much faster - however, a single phase is likely not sufficient to solve the task. We refer to the performance ablation on the number of rounds. Note that the main aim of the FE Net is not to have a faster execution, but to leverage a new mechanism other than standard message-passing.

| Task | Model | ER 1600 | WS 1600 | DELAUNAY 1600 |
|------|-------|---------|---------|---------------|
| BFS | PGN | $40.2 _{\pm 3.0}$ | $32.0 _{\pm 2.0}$ | $65.1 _{\pm 2.6}$ |
| | FE-1 | $47.0 _{\pm 2.0}$ | $50.1 _{\pm 2.3}$ | $97.7 _{\pm 3.6}$ |
| DFS | PGN | $22360.4 _{\pm 1107.0}$ | $10624.2 _{\pm 247.1}$ | $11471.3 _{\pm 8.7}$ |
| | FE-1 | $44.6 _{\pm 1.8}$ | $73.1 _{\pm 6.9}$ | $105.5 _{\pm 3.9}$ |
| Dijkstra | PGN | $11021.9 _{\pm 384.2}$ | $4094.0 _{\pm 96.8}$ | $4471.2 _{\pm 23.4}$ |
| | FE-1 | $49.8 _{\pm 1.2}$ | $87.5 _{\pm 7.0}$ | $108.2 _{\pm 3.9}$ |
| Eccentricity | PGN | $56.8 _{\pm 5.5}$ | $62.1 _{\pm 3.7}$ | $116.2 _{\pm 3.2}$ |
| | FE-1 | $40.2 _{\pm 2.1}$ | $55.5 _{\pm 2.8}$ | $94.7 _{\pm 3.2}$ |
| MIS | PGN | $45.7 _{\pm 3.7}$ | $27.0 _{\pm 1.2}$ | $22.4 _{\pm 1.2}$ |
| | FE-1 | $40.2 _{\pm 2.6}$ | $46.5 _{\pm 2.2}$ | $98.1 _{\pm 8.4}$ |
| MST | PGN | $9162.6 _{\pm 531.3}$ | $4589.2 _{\pm 116.8}$ | $4793.6 _{\pm 90.8}$ |
| | FE-1 | $47.0 _{\pm 2.5}$ | $75.8 _{\pm 5.6}$ | $106.1 _{\pm 3.8}$ |

Note that we take the exact same setup as in the PrefixSum task. Therefore, GIN always executes 5 layers and its runtime is not really impacted on larger graph. The RecGNN baseline performs $1.2n$ rounds of message-passing, where $n$ denotes the graph size. As the graphs grow larger, the runtime increases roughly linear. A similar behaviour can be seen in the *random* and *fixed* variations of the FE Net. Note, that they execute two phases, each consisting of a flooding and echoing part. Therefore, there are about $4n$ steps of message-passing. Together with the precomputation of the distances for appropriate masking, this can account for the relative difference in performance. The *all* variation of Flood and Echo performs $n$ single executions in a sequential order. It might be possible to at least partially parallelize these executions. However, as the number of different runs scales with the number of nodes, we believe that the fixed and random variants of the FE Net are more suited for the study of extrapolation.

Note that in this specific experiment, the diameter of the graph is $n$. Due to the way the mechanism couples the number of iterations to the graph diameter, this is the worst case scenario. Therefore, we expect the performance ratio compared to RecGNN (which scales the number of iterations to the graph size) to be upper bounded by our measurements. While the current implementation is a bit slower compared to the standard MPNNS, due to the GPU support, the performance is still reasonable and practical for further research. Further, recall that the achieved performance of the models drastically differ. Moreover, while this is not yet the case for the current implementation, future implementations could leverage that the set of simultaneously active nodes is much smaller than the graph itself. This could drastically improve the overall usage of the GPU memory and open up further applications.

## H.3 Standard Deviation of *random* Variation

By using the *random* variant, we introduce a certain randomness in the computation, which could result in different outcomes depending on the chosen origin node.

We measure the deviation of the random variant in the PrefixSum task. Each model performs 50 runs over 1000 graphs, we report the node and graph accuracy in percent as well as the minimum and maximum

Table B: Graph Accuracy on the SALSA-CLRS benchmark for the FE Net and PGN on ER graphs. FE-X denotes that the model executes X phases. The FE Net can achieve significant improvements in graph accuracies on tasks such as BFS or Eccentricity. Furthermore, the performance is greatly increased on other tasks when the number of phases is increased.

| Task | Model | 16 | 80 | 160 | 800 | 1600 |
|---|---|---|---|---|---|---|
| BFS | FE - 1 | $100.0 \pm 0.0$ | $98.7 \pm 1.0$ | $87.0 \pm 9.3$ | $8.2 \pm 7.6$ | $1.3 \pm 2.1$ |
| | FE - 4 | $100.0 \pm 0.0$ | $99.8 \pm 0.3$ | $94.3 \pm 4.3$ | $17.2 \pm 11.1$ | $2.3 \pm 3.1$ |
| | FE - 16 | $100.0 \pm 0.0$ | $99.5 \pm 0.8$ | $97.5 \pm 2.2$ | $33.9 \pm 16.6$ | $7.7 \pm 6.7$ |
| | PGN | $100.0 \pm 0.0$ | $88.7 \pm 5.9$ | $54.9 \pm 21.5$ | $0.2 \pm 0.1$ | $0.0 \pm 0.0$ |
| DFS | FE - 1 | $0.9 \pm 0.2$ | $0.0 \pm 0.0$ | $0.0 \pm 0.0$ | $0.0 \pm 0.0$ | $0.0 \pm 0.0$ |
| | FE - 4 | $82.1 \pm 1.2$ | $0.0 \pm 0.0$ | $0.0 \pm 0.0$ | $0.0 \pm 0.0$ | $0.0 \pm 0.0$ |
| | FE - 16 | $51.4 \pm 27.1$ | $0.0 \pm 0.0$ | $0.0 \pm 0.0$ | $0.0 \pm 0.0$ | $0.0 \pm 0.0$ |
| | PGN | $18.4 \pm 37.7$ | $0.0 \pm 0.0$ | $0.0 \pm 0.0$ | $0.0 \pm 0.0$ | $0.0 \pm 0.0$ |
| Dijkstra | FE - 1 | $74.4 \pm 3.7$ | $0.7 \pm 0.4$ | $0.0 \pm 0.0$ | $0.0 \pm 0.0$ | $0.0 \pm 0.0$ |
| | FE - 4 | $91.2 \pm 1.0$ | $11.7 \pm 2.6$ | $0.2 \pm 0.2$ | $0.0 \pm 0.0$ | $0.0 \pm 0.0$ |
| | FE - 16 | $91.2 \pm 1.6$ | $11.9 \pm 3.7$ | $0.3 \pm 0.3$ | $0.0 \pm 0.0$ | $0.0 \pm 0.0$ |
| | PGN | $94.6 \pm 1.1$ | $37.8 \pm 6.9$ | $5.2 \pm 1.9$ | $0.0 \pm 0.0$ | $0.0 \pm 0.0$ |
| Eccentricity | FE - 1 | $99.8 \pm 0.1$ | $99.9 \pm 0.1$ | $98.9 \pm 0.3$ | $99.4 \pm 0.2$ | $81.7 \pm 9.4$ |
| | FE - 4 | $99.9 \pm 0.2$ | $99.7 \pm 0.5$ | $98.5 \pm 1.5$ | $98.6 \pm 2.4$ | $73.4 \pm 13.5$ |
| | FE - 16 | $99.8 \pm 0.2$ | $99.5 \pm 1.2$ | $98.0 \pm 2.1$ | $95.8 \pm 7.8$ | $66.9 \pm 16.1$ |
| | PGN | $100.0 \pm 0.0$ | $100.0 \pm 0.0$ | $100.0 \pm 0.0$ | $100.0 \pm 0.0$ | $64.6 \pm 14.9$ |
| MIS | FE - 1 | $39.5 \pm 1.4$ | $0.2 \pm 0.2$ | $0.0 \pm 0.0$ | $0.0 \pm 0.0$ | $0.0 \pm 0.0$ |
| | FE - 4 | $90.7 \pm 2.7$ | $36.4 \pm 8.8$ | $18.3 \pm 7.8$ | $0.0 \pm 0.0$ | $0.0 \pm 0.0$ |
| | FE - 16 | $97.9 \pm 0.8$ | $89.7 \pm 4.9$ | $79.9 \pm 9.2$ | $23.1 \pm 15.3$ | $12.7 \pm 10.2$ |
| | PGN | $98.8 \pm 0.2$ | $89.2 \pm 4.6$ | $74.1 \pm 10.1$ | $10.7 \pm 10.5$ | $2.0 \pm 2.5$ |
| MST | FE - 1 | $18.3 \pm 0.4$ | $0.0 \pm 0.0$ | $0.0 \pm 0.0$ | $0.0 \pm 0.0$ | $0.0 \pm 0.0$ |
| | FE - 4 | $53.3 \pm 6.0$ | $0.1 \pm 0.1$ | $0.0 \pm 0.0$ | $0.0 \pm 0.0$ | $0.0 \pm 0.0$ |
| | FE - 16 | $58.5 \pm 4.6$ | $0.1 \pm 0.1$ | $0.0 \pm 0.0$ | $0.0 \pm 0.0$ | $0.0 \pm 0.0$ |
| | PGN | $79.2 \pm 4.3$ | $2.0 \pm 1.2$ | $0.0 \pm 0.0$ | $0.0 \pm 0.0$ | $0.0 \pm 0.0$ |

Table 7: Runtime measurements performed on the PrefixSum task on 1000 graphs per graph size. We report the mean time per graph in ms and the corresponding standard deviation. All measurements were performed on a NVIDIA GeForce RTX 3090.

| Model | Time Measurement [ms] | | | |
|---|---|---|---|---|
| | n(10) | n(20) | n(50) | n(100) |
| GIN | $0.003 \pm 0.023$ | $0.003 \pm 0.022$ | $0.003 \pm 0.027$ | $0.003 \pm 0.023$ |
| RecGNN | $0.008 \pm 0.023$ | $0.015 \pm 0.022$ | $0.034 \pm 0.028$ | $0.066 \pm 0.023$ |
| Flood and Echo *all* | $0.304 \pm 0.025$ | $1.284 \pm 0.031$ | $7.995 \pm 0.050$ | $31.169 \pm 0.168$ |
| Flood and Echo *random* | $0.031 \pm 0.025$ | $0.066 \pm 0.033$ | $0.160 \pm 0.042$ | $0.315 \pm 0.066$ |
| Flood and Echo *fixed* | $0.040 \pm 0.025$ | $0.084 \pm 0.030$ | $0.212 \pm 0.029$ | $0.422 \pm 0.029$ |

achieved accuracy for each model instance. From the results in Table 8, we can see that there are differences between the models, however, the variance due to the chosen origins within each model is quite small.

## I  Extrapolation

In Table 10 we report the full results for the Path-Finding task and in Table 9 for the Distance task.

Table 8: Measurement of the standard deviation of the Flood and Echo *random* variant. Each model performs 50 runs over 1000 graphs, we report the node and graph accuracy in percent as well as the minimum and maximum achieved accuracy for each model instance.

| Model | PREFIXSUM | | | |
|---|---|---|---|---|
| | n(100) | min,max | g(100) | min,max |
| Model A | $98.78 \pm 0.19$ | (98.28, 99.11) | $96.43 \pm 0.34$ | (95.70, 97.20) |
| Model B | $100.00 \pm 0.00$ | (100.00, 100.00) | $100.00 \pm 0.00$ | (100.00, 100.00) |
| Model C | $100.00 \pm 0.00$ | (100.00, 100.00) | $100.00 \pm 0.00$ | (100.00, 100.00) |
| Model D | $91.37 \pm 0.44$ | (90.40, 92.48) | $74.97 \pm 0.82$ | (73.50, 77.40) |
| Model E | $100.00 \pm 0.00$ | (100.00, 100.00) | $100.00 \pm 0.00$ | (100.00, 100.00) |

Table 9: Extrapolation on the Distance task. All models are trained with graphs of size 10 and then tested on larger graphs. The Flood and Echo models are able to generalize well to graphs 100 times the sizes encountered during training. We report both the node accuracy with $n()$ and the graph accuracy with $g()$.

| Model | MESSAGES | DISTANCE | | | | | |
|---|---|---|---|---|---|---|---|
| | | n(10) | g(10) | n(100) | g(100) | n(1000) | g(1000) |
| GIN | $\mathcal{O}(Lm)$ | $0.99 \pm 0.01$ | $0.92 \pm 0.06$ | $0.70 \pm 0.05$ | $0.00 \pm 0.00$ | $0.53 \pm 0.01$ | $0.00 \pm 0.00$ |
| PGN | $\mathcal{O}(nm)$ | $1.00 \pm 0.00$ | $1.00 \pm 0.00$ | $0.77 \pm 0.03$ | $0.00 \pm 0.00$ | $0.50 \pm 0.00$ | $0.00 \pm 0.00$ |
| RecGNN | $\mathcal{O}(nm)$ | $1.00 \pm 0.00$ | $1.00 \pm 0.00$ | $0.95 \pm 0.04$ | $0.45 \pm 0.33$ | $0.78 \pm 0.13$ | $0.00 \pm 0.00$ |
| Flood and Echo *random* | $\mathcal{O}(m)$ | $1.00 \pm 0.00$ | $1.00 \pm 0.00$ | $0.82 \pm 0.01$ | $0.01 \pm 0.00$ | $0.58 \pm 0.01$ | $0.00 \pm 0.00$ |
| Flood and Echo *fixed* | $\mathcal{O}(m)$ | $1.00 \pm 0.00$ | $1.00 \pm 0.00$ | $1.00 \pm 0.00$ | $1.00 \pm 0.00$ | $1.00 \pm 0.00$ | $1.00 \pm 0.00$ |

Table 10: Extrapolation on the Path-Finding task. All models are trained with graphs of size 10 and then tested on larger graphs. The Flood and Echo models are able to generalize well to graphs 100 times the sizes encountered during training. We report both the node accuracy with $n()$ and the graph accuracy with $g()$.

| Model | MESSAGES | PATH-FINDING | | | | | |
|---|---|---|---|---|---|---|---|
| | | n(10) | g(10) | n(100) | g(100) | n(1000) | g(1000) |
| GIN | $\mathcal{O}(Lm)$ | $0.97 \pm 0.01$ | $0.77 \pm 0.08$ | $0.91 \pm 0.01$ | $0.04 \pm 0.06$ | $0.95 \pm 0.01$ | $0.00 \pm 0.01$ |
| PGN | $\mathcal{O}(nm)$ | $0.99 \pm 0.01$ | $0.91 \pm 0.05$ | $0.89 \pm 0.01$ | $0.01 \pm 0.02$ | $0.96 \pm 0.00$ | $0.00 \pm 0.00$ |
| RecGNN | $\mathcal{O}(nm)$ | $1.00 \pm 0.00$ | $1.00 \pm 0.00$ | $0.99 \pm 0.02$ | $0.93 \pm 0.15$ | $0.99 \pm 0.01$ | $0.79 \pm 0.37$ |
| Flood and Echo *random* | $\mathcal{O}(m)$ | $1.00 \pm 0.00$ | $1.00 \pm 0.00$ | $0.97 \pm 0.04$ | $0.77 \pm 0.30$ | $0.98 \pm 0.02$ | $0.48 \pm 0.38$ |
| Flood and Echo *fixed* | $\mathcal{O}(m)$ | $1.00 \pm 0.00$ | $1.00 \pm 0.00$ | $1.00 \pm 0.00$ | $0.99 \pm 0.02$ | $1.00 \pm 0.00$ | $0.89 \pm 0.13$ |

## J  SALSA

We follow the training setup from Minder et al. (2023). If not specified otherwise, we run a single phase of the FE Net using batchsize 8, max aggregation, the AdamW optimizer with an initial learning rate around 0.0004 while also reducing the learning rate by a factor of 0.1 if the validation loss does not decrease for 10 epochs. We employ an early stop if the validation loss does not decrease for 25 epochs and run the training for at most 100 epochs. All reported mean accuracies are taken across 5 model run on a NVIDIA GeForce RTX 3090.

The full results for all tasks on all graph distributions is depicted in Table 13 for node accuracy and in Table 12 for graph accuracy. Further in Tables 14,15,16 and 17 we report the exact figures for the performance on MIS and Dijkstra if the number of rounds is increased. We test 1, 2, 4, 8 and 16 phases for the selection of the best FE Net model as reported in Table 11.

Table 11: Best performing number of phases for the FE Net on the different tasks of SALSA-CLRS.

| | ALGORITHMS | | | | | |
| --- | --- | --- | --- | --- | --- | --- |
| | BFS | DFS | Dijkstra | Eccentricity | MIS | MST |
| Number of Phases | 2 | 8 | 8 | 8 | 8 | 16 |

Table 12: We test the FE Net across multiple rounds on the SALSA-CLRS benchmark across six graph based algorithmic tasks. Flood and Echo - X, denotes that All models are trained on graphs of size 16 and then tested on larger graphs. We report the graph accuracy on Erdős–Rényi graphs of different sizes. All numbers are taken across 5 runs.

| Task | Model | ER 16 | 80 | 160 | 800 | 1600 | WS 16 | 80 | 160 | 800 | 1600 | DELAUNAY 16 | 80 | 160 | 800 | 1600 |
| --- | --- | --- | --- | --- | --- | --- | --- | --- | --- | --- | --- | --- | --- | --- | --- | --- |
| BFS | Flood and Echo - 1 | 100.0 ±0.0 | 98.7 ±1.0 | 87.0 ±9.3 | 8.2 ±7.6 | 1.3 ±2.1 | 100.0 ±0.0 | 33.2 ±11.5 | 4.6 ±3.9 | 0.0 ±0.0 | 0.0 ±0.0 | 100.0 ±0.0 | 59.5 ±15.0 | 9.6 ±9.7 | 0.0 ±0.0 | 0.0 ±0.0 |
| | Flood and Echo - 2 | 100.0 ±0.0 | 99.7 ±0.3 | 96.6 ±1.7 | 22.9 ±12.5 | 4.4 ±5.7 | 100.0 ±0.0 | 57.7 ±13.5 | 13.7 ±9.7 | 0.0 ±0.0 | 0.0 ±0.0 | 100.0 ±0.0 | 83.6 ±10.1 | 21.3 ±19.2 | 0.0 ±0.0 | 0.0 ±0.0 |
| | Flood and Echo - 4 | 100.0 ±0.0 | 99.8 ±0.3 | 94.3 ±4.3 | 17.2 ±11.1 | 2.3 ±3.1 | 100.0 ±0.0 | 54.6 ±11.3 | 15.4 ±10.6 | 0.0 ±0.0 | 0.0 ±0.0 | 100.0 ±0.0 | 79.9 ±6.1 | 17.5 ±14.5 | 0.0 ±0.0 | 0.0 ±0.0 |
| | Flood and Echo - 8 | 100.0 ±0.0 | 99.0 ±0.4 | 90.3 ±7.0 | 11.4 ±9.9 | 1.4 ±2.0 | 100.0 ±0.0 | 45.0 ±10.8 | 7.5 ±6.6 | 0.0 ±0.0 | 0.0 ±0.0 | 100.0 ±0.0 | 73.2 ±11.6 | 13.1 ±13.8 | 0.0 ±0.0 | 0.0 ±0.0 |
| | Flood and Echo - 16 | 100.0 ±0.0 | 99.5 ±0.8 | 97.5 ±2.2 | 33.9 ±16.6 | 7.7 ±6.7 | 100.0 ±0.0 | 46.3 ±11.9 | 13.9 ±8.3 | 0.0 ±0.0 | 0.0 ±0.0 | 100.0 ±0.0 | 78.2 ±6.6 | 21.1 ±19.3 | 0.0 ±0.0 | 0.0 ±0.0 |
| | Flood and Echo - 0 | 100.0 ±0.0 | 99.0 ±1.0 | 95.0 ±3.5 | 17.5 ±12.3 | 2.6 ±2.4 | 100.0 ±0.0 | 42.5 ±12.6 | 9.2 ±8.3 | 0.0 ±0.0 | 0.0 ±0.0 | 100.0 ±0.0 | 69.9 ±5.9 | 5.3 ±2.4 | 0.0 ±0.0 | 0.0 ±0.0 |
| | GIN(E) | 99.4 ±0.8 | 84.3 ±13.9 | 57.5 ±15.3 | 2.2 ±4.1 | 0.1 ±0.2 | 98.0 ±4.2 | 5.7 ±8.7 | 0.2 ±0.5 | 0.0 ±0.0 | 0.0 ±0.0 | 99.3 ±1.0 | 25.1 ±28.6 | 0.7 ±1.4 | 0.0 ±0.0 | 0.0 ±0.0 |
| | PGN | 100.0 ±0.0 | 88.7 ±5.9 | 54.9 ±21.5 | 0.2 ±0.1 | 0.0 ±0.0 | 100.0 ±0.0 | 13.1 ±3.3 | 0.1 ±0.1 | 0.0 ±0.0 | 0.0 ±0.0 | 100.0 ±0.0 | 35.1 ±6.3 | 0.3 ±0.4 | 0.0 ±0.0 | 0.0 ±0.0 |
| | RecGNN | 99.9 ±0.2 | 87.9 ±8.8 | 55.8 ±24.8 | 4.6 ±6.5 | 0.4 ±0.6 | 100.0 ±0.0 | 32.5 ±18.3 | 1.0 ±1.2 | 0.0 ±0.0 | 0.0 ±0.0 | 100.0 ±0.0 | 53.4 ±11.5 | 1.7 ±1.2 | 0.0 ±0.0 | 0.0 ±0.0 |
| BFS (H) | GIN(E) | 92.5 ±13.9 | 59.4 ±38.3 | 37.8 ±37.9 | 0.9 ±1.4 | 0.0 ±0.1 | 92.8 ±12.0 | 10.2 ±13.8 | 0.4 ±0.7 | 0.0 ±0.0 | 0.0 ±0.0 | 85.2 ±3.9 | 17.5 ±17.7 | 0.2 ±0.3 | 0.0 ±0.0 | 0.0 ±0.0 |
| | PGN | 100.0 ±0.0 | 88.1 ±3.8 | 66.3 ±8.7 | 0.2 ±0.3 | 0.0 ±0.0 | 100.0 ±0.0 | 14.2 ±3.6 | 0.2 ±0.2 | 0.0 ±0.0 | 0.0 ±0.0 | 100.0 ±0.0 | 26.2 ±11.5 | 0.1 ±0.1 | 0.0 ±0.0 | 0.0 ±0.0 |
| | RecGNN | 99.9 ±0.1 | 81.7 ±13.0 | 49.6 ±25.2 | 1.8 ±2.3 | 0.0 ±0.1 | 99.4 ±1.3 | 20.7 ±13.5 | 1.3 ±2.3 | 0.0 ±0.0 | 0.0 ±0.0 | 99.9 ±0.2 | 18.7 ±8.4 | 0.0 ±0.0 | 0.0 ±0.0 | 0.0 ±0.0 |
| DFS | Flood and Echo - 1 | 0.9 ±0.2 | 0.0 ±0.0 | 0.0 ±0.0 | 0.0 ±0.0 | 0.0 ±0.0 | 0.0 ±0.0 | 0.0 ±0.0 | 0.0 ±0.0 | 0.0 ±0.0 | 0.0 ±0.0 | 0.2 ±0.1 | 0.0 ±0.0 | 0.0 ±0.0 | 0.0 ±0.0 | 0.0 ±0.0 |
| | Flood and Echo - 2 | 14.3 ±4.3 | 0.0 ±0.0 | 0.0 ±0.0 | 0.0 ±0.0 | 0.0 ±0.0 | 6.5 ±5.6 | 0.0 ±0.0 | 0.0 ±0.0 | 0.0 ±0.0 | 0.0 ±0.0 | 8.6 ±3.3 | 0.0 ±0.0 | 0.0 ±0.0 | 0.0 ±0.0 | 0.0 ±0.0 |
| | Flood and Echo - 4 | 82.1 ±1.2 | 0.0 ±0.0 | 0.0 ±0.0 | 0.0 ±0.0 | 0.0 ±0.0 | 92.0 ±2.7 | 0.0 ±0.0 | 0.0 ±0.0 | 0.0 ±0.0 | 0.0 ±0.0 | 63.7 ±1.5 | 0.0 ±0.0 | 0.0 ±0.0 | 0.0 ±0.0 | 0.0 ±0.0 |
| | Flood and Echo - 8 | 88.9 ±3.0 | 0.0 ±0.0 | 0.0 ±0.0 | 0.0 ±0.0 | 0.0 ±0.0 | 81.2 ±12.5 | 0.0 ±0.0 | 0.0 ±0.0 | 0.0 ±0.0 | 0.0 ±0.0 | 68.3 ±3.0 | 0.0 ±0.0 | 0.0 ±0.0 | 0.0 ±0.0 | 0.0 ±0.0 |
| | Flood and Echo - 16 | 51.4 ±27.1 | 0.0 ±0.0 | 0.0 ±0.0 | 0.0 ±0.0 | 0.0 ±0.0 | 35.2 ±31.7 | 0.0 ±0.0 | 0.0 ±0.0 | 0.0 ±0.0 | 0.0 ±0.0 | 32.1 ±18.5 | 0.0 ±0.0 | 0.0 ±0.0 | 0.0 ±0.0 | 0.0 ±0.0 |
| | Flood and Echo - 0 | 0.2 ±0.2 | 0.0 ±0.0 | 0.0 ±0.0 | 0.0 ±0.0 | 0.0 ±0.0 | 0.0 ±0.0 | 0.0 ±0.0 | 0.0 ±0.0 | 0.0 ±0.0 | 0.0 ±0.0 | 0.0 ±0.1 | 0.0 ±0.0 | 0.0 ±0.0 | 0.0 ±0.0 | 0.0 ±0.0 |
| | GIN(E) | 0.1 ±0.1 | 0.0 ±0.0 | 0.0 ±0.0 | 0.0 ±0.0 | 0.0 ±0.0 | 0.0 ±0.0 | 0.0 ±0.0 | 0.0 ±0.0 | 0.0 ±0.0 | 0.0 ±0.0 | 0.0 ±0.0 | 0.0 ±0.0 | 0.0 ±0.0 | 0.0 ±0.0 | 0.0 ±0.0 |
| | PGN | 18.4 ±37.7 | 0.0 ±0.0 | 0.0 ±0.0 | 0.0 ±0.0 | 0.0 ±0.0 | 9.5 ±21.2 | 0.0 ±0.0 | 0.0 ±0.0 | 0.0 ±0.0 | 0.0 ±0.0 | 13.9 ±29.4 | 0.0 ±0.0 | 0.0 ±0.0 | 0.0 ±0.0 | 0.0 ±0.0 |
| | RecGNN | 0.0 ±0.0 | 0.0 ±0.0 | 0.0 ±0.0 | 0.0 ±0.0 | 0.0 ±0.0 | 0.0 ±0.0 | 0.0 ±0.0 | 0.0 ±0.0 | 0.0 ±0.0 | 0.0 ±0.0 | 0.0 ±0.0 | 0.0 ±0.0 | 0.0 ±0.0 | 0.0 ±0.0 | 0.0 ±0.0 |
| DFS (H) | GIN(E) | 0.0 ±0.0 | 0.0 ±0.0 | 0.0 ±0.0 | 0.0 ±0.0 | 0.0 ±0.0 | 0.0 ±0.0 | 0.0 ±0.0 | 0.0 ±0.0 | 0.0 ±0.0 | 0.0 ±0.0 | 0.0 ±0.0 | 0.0 ±0.0 | 0.0 ±0.0 | 0.0 ±0.0 | 0.0 ±0.0 |
| | PGN | 19.9 ±30.7 | 0.0 ±0.0 | 0.0 ±0.0 | 0.0 ±0.0 | 0.0 ±0.0 | 3.2 ±7.2 | 0.0 ±0.0 | 0.0 ±0.0 | 0.0 ±0.0 | 0.0 ±0.0 | 13.8 ±23.0 | 0.0 ±0.0 | 0.0 ±0.0 | 0.0 ±0.0 | 0.0 ±0.0 |
| | RecGNN | 4.5 ±7.8 | 0.0 ±0.0 | 0.0 ±0.0 | 0.0 ±0.0 | 0.0 ±0.0 | 0.0 ±0.0 | 0.0 ±0.0 | 0.0 ±0.0 | 0.0 ±0.0 | 0.0 ±0.0 | 5.8 ±11.5 | 0.0 ±0.0 | 0.0 ±0.0 | 0.0 ±0.0 | 0.0 ±0.0 |
| Dijkstra | Flood and Echo - 1 | 74.4 ±3.7 | 0.7 ±0.4 | 0.0 ±0.0 | 0.0 ±0.0 | 0.0 ±0.0 | 53.4 ±5.4 | 1.2 ±0.8 | 0.0 ±0.0 | 0.0 ±0.0 | 0.0 ±0.0 | 67.1 ±5.1 | 0.1 ±0.1 | 0.0 ±0.0 | 0.0 ±0.0 | 0.0 ±0.0 |
| | Flood and Echo - 2 | 84.8 ±1.1 | 4.2 ±1.0 | 0.0 ±0.1 | 0.0 ±0.0 | 0.0 ±0.0 | 71.2 ±3.2 | 4.4 ±0.7 | 0.0 ±0.0 | 0.0 ±0.0 | 0.0 ±0.0 | 79.7 ±2.5 | 1.7 ±0.8 | 0.0 ±0.0 | 0.0 ±0.0 | 0.0 ±0.0 |
| | Flood and Echo - 4 | 91.2 ±1.0 | 11.7 ±2.6 | 0.2 ±0.2 | 0.0 ±0.0 | 0.0 ±0.0 | 78.7 ±2.9 | 12.3 ±2.2 | 0.3 ±0.1 | 0.0 ±0.0 | 0.0 ±0.0 | 87.8 ±1.2 | 6.6 ±1.4 | 0.0 ±0.0 | 0.0 ±0.0 | 0.0 ±0.0 |
| | Flood and Echo - 8 | 91.8 ±0.7 | 13.2 ±1.7 | 0.5 ±0.2 | 0.0 ±0.0 | 0.0 ±0.0 | 79.0 ±2.5 | 13.1 ±3.5 | 0.3 ±0.5 | 0.0 ±0.0 | 0.0 ±0.0 | 89.2 ±1.0 | 7.3 ±3.1 | 0.0 ±0.1 | 0.0 ±0.0 | 0.0 ±0.0 |
| | Flood and Echo - 16 | 91.2 ±1.6 | 11.9 ±3.7 | 0.3 ±0.0 | 0.0 ±0.0 | 0.0 ±0.0 | 76.0 ±5.1 | 10.5 ±4.6 | 0.2 ±0.2 | 0.0 ±0.0 | 0.0 ±0.0 | 87.9 ±1.6 | 6.3 ±2.4 | 0.0 ±0.1 | 0.0 ±0.0 | 0.0 ±0.0 |
| | Flood and Echo - 0 | 91.0 ±1.1 | 10.9 ±3.6 | 0.3 ±0.4 | 0.0 ±0.0 | 0.0 ±0.0 | 76.3 ±4.8 | 10.9 ±4.1 | 0.2 ±0.0 | 0.0 ±0.0 | 0.0 ±0.0 | 87.1 ±2.4 | 6.0 ±3.8 | 0.0 ±0.1 | 0.0 ±0.0 | 0.0 ±0.0 |
| | GIN(E) | 73.4 ±2.6 | 0.2 ±0.2 | 0.0 ±0.0 | 0.0 ±0.0 | 0.0 ±0.0 | 51.6 ±3.0 | 0.0 ±0.0 | 0.0 ±0.0 | 0.0 ±0.0 | 0.0 ±0.0 | 66.6 ±4.3 | 0.0 ±0.0 | 0.0 ±0.0 | 0.0 ±0.0 | 0.0 ±0.0 |
| | PGN | 94.6 ±1.1 | 37.8 ±6.9 | 5.2 ±1.9 | 0.0 ±0.0 | 0.0 ±0.0 | 76.4 ±4.0 | 17.2 ±2.8 | 0.9 ±0.8 | 0.0 ±0.0 | 0.0 ±0.0 | 93.0 ±1.4 | 19.2 ±4.2 | 0.1 ±0.0 | 0.0 ±0.0 | 0.0 ±0.0 |
| | RecGNN | 81.7 ±16.1 | 6.8 ±6.1 | 0.3 ±0.5 | 0.0 ±0.0 | 0.0 ±0.0 | 60.4 ±22.7 | 8.4 ±7.4 | 0.2 ±0.2 | 0.0 ±0.0 | 0.0 ±0.0 | 74.4 ±19.9 | 4.4 ±4.5 | 0.0 ±0.0 | 0.0 ±0.0 | 0.0 ±0.0 |
| Dijkstra (H) | GIN(E) | 49.8 ±10.8 | 0.0 ±0.0 | 0.0 ±0.0 | 0.0 ±0.0 | 0.0 ±0.0 | 28.7 ±9.9 | 0.0 ±0.0 | 0.0 ±0.0 | 0.0 ±0.0 | 0.0 ±0.0 | 40.3 ±10.4 | 0.0 ±0.0 | 0.0 ±0.0 | 0.0 ±0.0 | 0.0 ±0.0 |
| | PGN | 89.5 ±1.0 | 3.3 ±3.7 | 0.0 ±0.1 | 0.0 ±0.0 | 0.0 ±0.0 | 70.8 ±2.4 | 0.4 ±0.6 | 0.0 ±0.0 | 0.0 ±0.0 | 0.0 ±0.0 | 87.6 ±0.7 | 0.4 ±0.8 | 0.0 ±0.0 | 0.0 ±0.0 | 0.0 ±0.0 |
| | RecGNN | 73.8 ±1.6 | 0.0 ±0.0 | 0.0 ±0.0 | 0.0 ±0.0 | 0.0 ±0.0 | 50.9 ±5.6 | 0.0 ±0.0 | 0.0 ±0.0 | 0.0 ±0.0 | 0.0 ±0.0 | 66.4 ±3.3 | 0.0 ±0.0 | 0.0 ±0.0 | 0.0 ±0.0 | 0.0 ±0.0 |
| Eccentricity | Flood and Echo - 1 | 99.8 ±0.1 | 99.9 ±0.1 | 98.9 ±0.3 | 99.4 ±0.2 | 81.7 ±9.4 | 100.0 ±0.0 | 88.6 ±0.8 | 93.2 ±6.0 | 36.2 ±6.6 | 29.2 ±0.0 | 100.0 ±0.0 | 80.8 ±11.3 | 73.7 ±6.6 | 0.0 ±0.0 | 0.0 ±0.0 |
| | Flood and Echo - 2 | 99.9 ±0.1 | 100.0 ±0.0 | 99.1 ±0.1 | 99.2 ±1.4 | 70.1 ±15.3 | 100.0 ±0.0 | 87.5 ±0.4 | 97.7 ±1.8 | 38.7 ±2.7 | 25.1 ±10.7 | 100.0 ±0.0 | 95.3 ±2.1 | 72.9 ±12.6 | 0.0 ±0.0 | 0.0 ±0.0 |
| | Flood and Echo - 4 | 99.9 ±0.2 | 99.7 ±0.5 | 98.5 ±1.5 | 98.6 ±2.4 | 73.4 ±13.5 | 100.0 ±0.0 | 88.5 ±1.8 | 96.0 ±4.6 | 40.4 ±3.8 | 22.8 ±9.3 | 100.0 ±0.0 | 93.5 ±6.7 | 73.4 ±16.3 | 0.0 ±0.0 | 0.0 ±0.0 |
| | Flood and Echo - 8 | 99.9 ±0.0 | 99.9 ±0.1 | 98.8 ±0.4 | 99.5 ±0.3 | 81.1 ±5.4 | 100.0 ±0.0 | 87.4 ±3.1 | 92.3 ±7.9 | 29.7 ±11.1 | 20.8 ±7.2 | 100.0 ±0.0 | 82.7 ±15.5 | 54.9 ±28.5 | 0.0 ±0.0 | 0.0 ±0.0 |
| | Flood and Echo - 16 | 99.8 ±0.2 | 99.5 ±1.2 | 98.0 ±2.1 | 95.8 ±7.8 | 66.9 ±16.1 | 100.0 ±0.0 | 88.4 ±2.2 | 95.7 ±2.6 | 36.5 ±6.2 | 29.4 ±6.2 | 100.0 ±0.0 | 89.1 ±9.4 | 69.6 ±11.6 | 0.0 ±0.0 | 0.0 ±0.0 |
| | Flood and Echo - 0 | 99.9 ±0.1 | 100.0 ±0.0 | 99.4 ±0.6 | 99.4 ±0.6 | 75.6 ±11.0 | 100.0 ±0.0 | 88.7 ±1.3 | 97.7 ±1.3 | 36.9 ±2.2 | 30.4 ±7.2 | 100.0 ±0.0 | 90.8 ±7.3 | 66.2 ±4.8 | 0.0 ±0.0 | 0.0 ±0.0 |
| | GIN(E) | 57.3 ±21.2 | 77.1 ±17.5 | 72.3 ±18.0 | 51.3 ±34.2 | 36.7 ±17.6 | 78.0 ±18.7 | 27.6 ±19.5 | 3.6 ±8.0 | 0.0 ±0.0 | 0.0 ±0.0 | 84.8 ±12.4 | 0.0 ±0.0 | 0.0 ±0.0 | 0.0 ±0.0 | 0.0 ±0.0 |
| | PGN | 100.0 ±0.0 | 100.0 ±0.0 | 100.0 ±0.0 | 100.0 ±0.0 | 64.6 ±14.9 | 100.0 ±0.0 | 100.0 ±0.0 | 93.8 ±2.1 | 0.0 ±0.0 | 0.0 ±0.0 | 100.0 ±0.0 | 76.9 ±19.8 | 0.0 ±0.0 | 0.0 ±0.0 | 0.0 ±0.0 |
| | RecGNN | 75.8 ±26.2 | 80.5 ±35.0 | 75.0 ±39.1 | 72.7 ±27.9 | 63.0 ±24.8 | 86.7 ±25.7 | 60.8 ±29.1 | 57.4 ±38.7 | 27.6 ±29.4 | 15.2 ±13.7 | 89.9 ±19.4 | 25.2 ±37.6 | 8.3 ±11.9 | 0.0 ±0.0 | 0.0 ±0.0 |
| Eccentricity (H) | GIN(E) | 25.3 ±41.0 | 23.8 ±39.0 | 26.1 ±36.8 | 17.1 ±32.9 | 16.0 ±21.7 | 25.3 ±42.2 | 19.0 ±18.8 | 18.6 ±18.9 | 4.6 ±6.9 | 9.8 ±10.2 | 24.8 ±42.5 | 17.0 ±12.5 | 3.0 ±5.8 | 0.0 ±0.0 | 0.0 ±0.0 |
| | PGN | 100.0 ±0.0 | 100.0 ±0.0 | 100.0 ±0.0 | 100.0 ±0.0 | 83.0 ±6.5 | 100.0 ±0.0 | 88.3 ±1.8 | 100.0 ±0.1 | 34.8 ±7.2 | 9.2 ±4.8 | 100.0 ±0.0 | 99.7 ±0.3 | 64.4 ±14.2 | 0.0 ±0.0 | 0.0 ±0.0 |
| | RecGNN | 95.0 ±6.3 | 96.6 ±3.6 | 95.8 ±4.6 | 93.4 ±10.3 | 72.1 ±20.9 | 99.0 ±1.2 | 66.4 ±22.4 | 46.2 ±40.9 | 14.1 ±6.3 | 8.3 ±4.9 | 99.6 ±0.6 | 51.0 ±36.0 | 19.4 ±11.7 | 0.0 ±0.0 | 0.0 ±0.0 |
| MIS | Flood and Echo - 1 | 39.5 ±1.4 | 0.2 ±0.2 | 0.0 ±0.0 | 0.0 ±0.0 | 0.0 ±0.0 | 40.9 ±0.5 | 1.0 ±0.1 | 0.0 ±0.0 | 0.0 ±0.0 | 0.0 ±0.0 | 43.7 ±0.6 | 1.0 ±0.2 | 0.0 ±0.0 | 0.0 ±0.0 | 0.0 ±0.0 |
| | Flood and Echo - 2 | 47.6 ±2.1 | 0.5 ±0.3 | 0.0 ±0.0 | 0.0 ±0.0 | 0.0 ±0.0 | 46.4 ±1.9 | 2.8 ±0.6 | 0.0 ±0.0 | 0.0 ±0.0 | 0.0 ±0.0 | 48.7 ±1.9 | 2.3 ±0.4 | 0.0 ±0.1 | 0.0 ±0.0 | 0.1 ±0.1 |
| | Flood and Echo - 4 | 90.7 ±2.7 | 36.4 ±8.8 | 18.3 ±7.8 | 0.0 ±0.0 | 0.0 ±0.0 | 92.3 ±1.8 | 72.6 ±5.9 | 47.3 ±8.6 | 3.0 ±2.5 | 0.1 ±0.1 | 94.5 ±2.0 | 69.2 ±5.4 | 45.6 ±6.8 | 2.0 ±1.4 | 0.1 ±0.1 |
| | Flood and Echo - 8 | 98.3 ±0.5 | 91.5 ±2.4 | 83.8 ±4.5 | 27.9 ±12.5 | 13.9 ±9.6 | 98.3 ±0.5 | 96.4 ±0.9 | 88.0 ±3.3 | 54.7 ±10.9 | 30.5 ±11.8 | 98.7 ±0.5 | 94.7 ±1.2 | 88.5 ±3.2 | 52.9 ±11.7 | 28.5 ±14.0 |
| | Flood and Echo - 16 | 97.9 ±0.8 | 89.7 ±4.9 | 79.9 ±9.2 | 23.1 ±15.3 | 12.7 ±10.2 | 98.3 ±0.6 | 95.4 ±2.5 | 85.7 ±6.1 | 50.5 ±16.7 | 28.8 ±15.9 | 98.2 ±0.8 | 93.6 ±3.5 | 87.7 ±5.5 | 50.0 ±15.4 | 27.8 ±15.8 |
| | Flood and Echo - 0 | 98.2 ±0.4 | 90.9 ±2.5 | 83.6 ±5.2 | 30.0 ±12.9 | 15.9 ±8.2 | 98.3 ±0.5 | 96.1 ±1.7 | 89.0 ±5.0 | 59.7 ±12.9 | 38.7 ±15.0 | 98.9 ±0.3 | 95.0 ±2.1 | 89.8 ±3.6 | 57.3 ±10.7 | 34.5 ±11.9 |
| | GIN(E) | 6.2 ±3.2 | 0.0 ±0.0 | 0.0 ±0.0 | 0.0 ±0.0 | 0.0 ±0.0 | 6.5 ±2.8 | 0.0 ±0.0 | 0.0 ±0.0 | 0.0 ±0.0 | 0.0 ±0.0 | 6.1 ±3.8 | 0.0 ±0.0 | 0.0 ±0.0 | 0.0 ±0.0 | 0.0 ±0.0 |
| | PGN | 98.8 ±0.2 | 89.2 ±4.6 | 74.1 ±10.1 | 10.7 ±10.5 | 2.0 ±2.5 | 98.1 ±0.6 | 84.4 ±8.4 | 58.3 ±14.1 | 4.6 ±4.3 | 0.5 ±0.6 | 98.9 ±0.6 | 93.9 ±2.2 | 87.2 ±4.9 | 41.2 ±8.9 | 17.4 ±7.4 |
| | RecGNN | 56.1 ±13.1 | 5.5 ±7.1 | 0.8 ±1.6 | 0.0 ±0.0 | 0.0 ±0.0 | 52.6 ±14.6 | 9.0 ±9.8 | 2.0 ±2.9 | 0.0 ±0.0 | 0.0 ±0.0 | 56.0 ±13.3 | 9.6 ±7.8 | 1.7 ±2.2 | 0.0 ±0.0 | 0.0 ±0.0 |
| MIS (H) | GIN(E) | 3.3 ±2.5 | 0.0 ±0.0 | 0.0 ±0.0 | 0.0 ±0.0 | 0.0 ±0.0 | 4.4 ±2.3 | 0.0 ±0.0 | 0.0 ±0.0 | 0.0 ±0.0 | 0.0 ±0.0 | 3.3 ±2.2 | 0.0 ±0.0 | 0.0 ±0.0 | 0.0 ±0.0 | 0.0 ±0.0 |
| | PGN | 98.6 ±0.4 | 88.9 ±3.1 | 76.7 ±3.6 | 18.0 ±8.6 | 5.2 ±4.3 | 98.2 ±0.3 | 82.2 ±8.7 | 54.1 ±6.6 | 2.3 ±1.7 | 0.1 ±0.0 | 98.6 ±0.2 | 87.5 ±4.0 | 85.1 ±3.4 | 40.1 ±9.1 | 15.1 ±6.5 |
| | RecGNN | 44.1 ±5.8 | 2.6 ±1.5 | 0.1 ±0.2 | 0.0 ±0.0 | 0.0 ±0.0 | 46.5 ±5.7 | 4.2 ±1.1 | 0.4 ±0.4 | 0.0 ±0.0 | 0.0 ±0.0 | 46.9 ±6.2 | 4.8 ±1.4 | 0.3 ±0.3 | 0.0 ±0.0 | 0.0 ±0.0 |
| MST | Flood and Echo - 1 | 18.3 ±0.4 | 0.0 ±0.0 | 0.0 ±0.0 | 0.0 ±0.0 | 0.0 ±0.0 | 10.8 ±2.5 | 0.0 ±0.0 | 0.0 ±0.0 | 0.0 ±0.0 | 0.0 ±0.0 | 17.5 ±1.8 | 0.0 ±0.0 | 0.0 ±0.0 | 0.0 ±0.0 | 0.0 ±0.0 |
| | Flood and Echo - 2 | 33.4 ±6.5 | 0.0 ±0.0 | 0.0 ±0.0 | 0.0 ±0.0 | 0.0 ±0.0 | 22.6 ±5.2 | 0.0 ±0.0 | 0.0 ±0.0 | 0.0 ±0.0 | 0.0 ±0.0 | 35.2 ±5.9 | 0.0 ±0.0 | 0.0 ±0.0 | 0.0 ±0.0 | 0.0 ±0.0 |
| | Flood and Echo - 4 | 53.3 ±6.0 | 0.1 ±0.1 | 0.0 ±0.0 | 0.0 ±0.0 | 0.0 ±0.0 | 40.2 ±6.3 | 0.0 ±0.0 | 0.0 ±0.0 | 0.0 ±0.0 | 0.0 ±0.0 | 54.8 ±5.3 | 0.0 ±0.1 | 0.0 ±0.0 | 0.0 ±0.0 | 0.0 ±0.0 |
| | Flood and Echo - 8 | 57.5 ±5.9 | 0.1 ±0.1 | 0.0 ±0.0 | 0.0 ±0.0 | 0.0 ±0.0 | 46.7 ±6.5 | 0.0 ±0.0 | 0.0 ±0.0 | 0.0 ±0.0 | 0.0 ±0.0 | 59.3 ±6.2 | 0.1 ±0.1 | 0.0 ±0.0 | 0.0 ±0.0 | 0.0 ±0.0 |
| | Flood and Echo - 16 | 58.5 ±4.6 | 0.1 ±0.1 | 0.0 ±0.0 | 0.0 ±0.0 | 0.0 ±0.0 | 46.5 ±6.1 | 0.0 ±0.1 | 0.0 ±0.0 | 0.0 ±0.0 | 0.0 ±0.0 | 59.7 ±4.3 | 0.1 ±0.1 | 0.0 ±0.0 | 0.0 ±0.0 | 0.0 ±0.0 |
| | Flood and Echo - 0 | 57.5 ±3.4 | 0.1 ±0.1 | 0.0 ±0.0 | 0.0 ±0.0 | 0.0 ±0.0 | 43.8 ±2.9 | 0.1 ±0.1 | 0.0 ±0.0 | 0.0 ±0.0 | 0.0 ±0.0 | 58.4 ±1.1 | 0.0 ±0.0 | 0.0 ±0.0 | 0.0 ±0.0 | 0.0 ±0.0 |
| | GIN(E) | 43.2 ±4.6 | 0.0 ±0.0 | 0.0 ±0.0 | 0.0 ±0.0 | 0.0 ±0.0 | 30.0 ±4.1 | 0.0 ±0.0 | 0.0 ±0.0 | 0.0 ±0.0 | 0.0 ±0.0 | 43.0 ±5.0 | 0.0 ±0.0 | 0.0 ±0.0 | 0.0 ±0.0 | 0.0 ±0.0 |
| | PGN | 79.2 ±4.3 | 2.0 ±1.2 | 0.0 ±0.0 | 0.0 ±0.0 | 0.0 ±0.0 | 73.2 ±9.1 | 0.3 ±0.3 | 0.0 ±0.0 | 0.0 ±0.0 | 0.0 ±0.0 | 78.8 ±4.1 | 6.6 ±0.4 | 0.0 ±0.0 | 0.0 ±0.0 | 0.0 ±0.0 |
| | RecGNN | 56.8 ±15.9 | 0.6 ±0.8 | 0.0 ±0.0 | 0.0 ±0.0 | 0.0 ±0.0 | 44.4 ±18.0 | 0.1 ±0.1 | 0.0 ±0.0 | 0.0 ±0.0 | 0.0 ±0.0 | 58.7 ±15.8 | 0.1 ±0.2 | 0.0 ±0.0 | 0.0 ±0.0 | 0.0 ±0.0 |
| MST (H) | GIN(E) | 29.7 ±5.6 | 0.0 ±0.0 | 0.0 ±0.0 | 0.0 ±0.0 | 0.0 ±0.0 | 20.4 ±5.0 | 0.0 ±0.0 | 0.0 ±0.0 | 0.0 ±0.0 | 0.0 ±0.0 | 34.6 ±6.0 | 0.0 ±0.0 | 0.0 ±0.0 | 0.0 ±0.0 | 0.0 ±0.0 |
| | PGN | 69.9 ±6.1 | 0.0 ±0.1 | 0.0 ±0.0 | 0.0 ±0.0 | 0.0 ±0.0 | 65.7 ±8.8 | 0.0 ±0.0 | 0.0 ±0.0 | 0.0 ±0.0 | 0.0 ±0.0 | 72.6 ±5.2 | 0.0 ±0.0 | 0.0 ±0.0 | 0.0 ±0.0 | 0.0 ±0.0 |
| | RecGNN | 24.5 ±7.5 | 0.0 ±0.0 | 0.0 ±0.0 | 0.0 ±0.0 | 0.0 ±0.0 | 14.8 ±5.4 | 0.0 ±0.0 | 0.0 ±0.0 | 0.0 ±0.0 | 0.0 ±0.0 | 26.0 ±7.5 | 0.0 ±0.0 | 0.0 ±0.0 | 0.0 ±0.0 | 0.0 ±0.0 |

Table 13: We test the FE Net across multiple rounds on the SALSA-CLRS benchmark across six graph based algorithmic tasks. Flood and Echo - X, denotes that All models are trained on graphs of size 16 and then tested on larger graphs. We report the node accuracy on Erdős–Rényi graphs of different sizes. All numbers are taken across 5 runs.

| Task | Model | ER | | | | | WS | | | | | Delaunay | | | | |
|---|---|---|---|---|---|---|---|---|---|---|---|---|---|---|---|---|
| | | 16 | 80 | 160 | 800 | 1600 | 16 | 80 | 160 | 800 | 1600 | 16 | 80 | 160 | 800 | 1600 |
| BFS | Flood and Echo - 1 | 100.0±0.0 | 100.0±0.0 | 99.9±0.1 | 99.5±0.2 | 99.4±0.3 | 100.0±0.0 | 97.8±0.6 | 95.2±1.2 | 84.5±1.7 | 83.1±2.4 | 100.0±0.0 | 99.2±0.4 | 96.8±2.0 | 69.3±6.6 | 59.5±12.6 |
| | Flood and Echo - 2 | 100.0±0.0 | 100.0±0.0 | 100.0±0.0 | 99.8±0.1 | 99.7±0.1 | 100.0±0.0 | 99.0±0.4 | 97.4±0.7 | 88.6±0.6 | 84.6±0.7 | 100.0±0.0 | 99.7±0.2 | 97.8±1.1 | 77.0±10.1 | 65.9±13.2 |
| | Flood and Echo - 4 | 100.0±0.0 | 100.0±0.0 | 100.0±0.0 | 99.7±0.1 | 99.6±0.1 | 100.0±0.0 | 98.8±0.3 | 97.3±0.7 | 88.1±1.5 | 83.9±0.9 | 100.0±0.0 | 99.7±0.1 | 98.1±1.3 | 76.4±10.2 | 61.2±8.1 |
| | Flood and Echo - 8 | 100.0±0.0 | 100.0±0.0 | 99.9±0.1 | 99.6±0.1 | 99.5±0.1 | 100.0±0.0 | 98.5±0.4 | 96.2±1.0 | 86.6±1.5 | 83.1±0.7 | 100.0±0.0 | 99.5±0.3 | 97.4±1.6 | 77.3±10.8 | 63.6±10.7 |
| | Flood and Echo - 16 | 100.0±0.0 | 100.0±0.0 | 100.0±0.0 | 99.8±0.1 | 99.7±0.2 | 100.0±0.0 | 98.6±0.3 | 96.6±1.16 | 86.2±8.0 | 85.0±1.0 | 100.0±0.0 | 99.6±0.1 | 97.5±2.1 | 79.2±15.5 | 68.6±14.5 |
| | Flood and Echo - 0 | 100.0±0.0 | 100.0±0.0 | 100.0±0.0 | 99.7±0.2 | 99.6±0.2 | 100.0±0.0 | 98.4±0.4 | 96.5±1.1 | 87.8±0.6 | 83.4±1.0 | 100.0±0.0 | 99.4±0.2 | 96.7±1.1 | 81.5±16.1 | 70.3±17.0 |
| | GIN(E) | 100.0±0.1 | 99.6±0.4 | 99.3±0.6 | 98.0±1.6 | 98.0±1.5 | 99.9±0.3 | 92.9±4.2 | 86.7±5.5 | 70.4±10.8 | 75.3±6.1 | 100.0±0.1 | 94.3±5.6 | 84.6±10.6 | 52.7±17.2 | 45.9±15.8 |
| | PGN | 100.0±0.0 | 99.8±0.1 | 99.5±0.3 | 99.0±0.2 | 98.9±0.2 | 100.0±0.0 | 95.5±0.7 | 88.7±1.5 | 75.9±3.3 | 80.6±0.7 | 100.0±0.0 | 98.2±0.7 | 90.4±4.5 | 53.6±7.0 | 40.3±6.5 |
| | RecGNN | 100.0±0.0 | 99.8±0.1 | 99.5±0.3 | 99.3±0.4 | 99.2±0.4 | 100.0±0.0 | 97.8±1.1 | 94.2±2.0 | 82.2±4.7 | 82.1±2.3 | 100.0±0.0 | 98.5±0.8 | 92.0±5.5 | 67.1±11.8 | 55.6±10.0 |
| BFS (H) | GIN(E) | 98.8±2.4 | 95.3±9.2 | 95.1±8.9 | 86.9±26.1 | 86.5±27.2 | 99.2±1.4 | 83.0±25.0 | 77.5±24.9 | 60.6±28.8 | 64.4±32.4 | 98.1±4.0 | 79.5±32.3 | 68.9±32.6 | 42.8±16.4 | 34.2±10.0 |
| | PGN | 100.0±0.0 | 99.8±0.1 | 99.6±0.1 | 98.7±0.3 | 98.5±0.3 | 100.0±0.0 | 96.1±0.5 | 90.8±0.8 | 76.4±1.6 | 80.6±1.0 | 100.0±0.0 | 97.5±0.8 | 89.4±1.6 | 53.2±2.4 | 40.8±3.2 |
| | RecGNN | 100.0±0.0 | 99.6±0.2 | 99.3±0.5 | 99.0±0.5 | 98.6±0.6 | 100.0±0.1 | 96.7±0.8 | 92.5±2.0 | 77.6±3.8 | 79.3±1.7 | 100.0±0.0 | 95.3±2.3 | 83.6±6.0 | 51.5±4.1 | 42.9±5.0 |
| DFS | Flood and Echo - 1 | 68.8±0.9 | 42.7±0.3 | 31.1±0.4 | 28.3±0.4 | 25.9±0.5 | 64.7±1.6 | 19.2±0.3 | 18.6±0.1 | 24.4±0.3 | 22.0±0.2 | 65.9±0.7 | 43.6±0.5 | 40.7±0.5 | 38.0±0.6 | 37.7±0.6 |
| | Flood and Echo - 2 | 83.1±2.3 | 47.4±0.6 | 35.0±0.8 | 31.9±0.7 | 29.5±0.9 | 83.2±5.4 | 21.0±0.2 | 19.8±0.1 | 24.6±0.3 | 22.2±0.2 | 81.0±2.2 | 48.9±1.0 | 44.7±0.8 | 41.3±0.6 | 40.8±0.6 |
| | Flood and Echo - 4 | 97.9±0.2 | 53.0±0.3 | 38.9±0.4 | 35.0±0.3 | 32.7±0.3 | 99.4±0.2 | 23.9±0.6 | 20.7±0.2 | 24.9±0.1 | 22.3±0.1 | 95.5±0.2 | 54.9±0.3 | 48.6±0.3 | 43.3±0.5 | 42.5±0.6 |
| | Flood and Echo - 8 | 98.9±0.3 | 52.8±0.6 | 38.2±0.5 | 33.5±0.5 | 31.1±0.6 | 98.4±1.3 | 24.1±1.0 | 21.0±0.6 | 24.1±0.1 | 21.6±0.4 | 96.2±0.3 | 54.3±0.9 | 46.4±0.7 | 39.8±0.7 | 38.9±0.7 |
| | Flood and Echo - 16 | 92.4±4.9 | 48.8±1.3 | 35.5±1.1 | 31.6±1.1 | 29.1±1.2 | 89.0±10.1 | 23.0±0.4 | 20.7±0.3 | 23.7±0.6 | 21.5±0.5 | 88.8±4.5 | 49.9±1.4 | 44.6±1.2 | 40.6±1.0 | 40.1±1.0 |
| | Flood and Echo - 0 | 58.6±3.5 | 38.5±1.9 | 26.4±1.3 | 24.2±1.2 | 21.9±1.1 | 42.1±3.8 | 20.6±0.8 | 19.9±0.5 | 23.3±0.5 | 21.2±0.4 | 56.9±3.8 | 39.1±1.8 | 36.4±1.6 | 34.3±1.4 | 34.1±1.4 |
| | GIN(E) | 49.3±8.1 | 30.6±4.0 | 19.7±3.9 | 18.1±3.8 | 16.5±3.5 | 29.7±4.9 | 15.9±0.9 | 16.8±0.6 | 22.3±0.6 | 20.1±0.5 | 46.7±7.3 | 28.0±3.1 | 25.1±3.1 | 23.4±2.9 | 23.2±2.9 |
| | PGN | 74.2±14.0 | 41.2±3.8 | 29.9±2.6 | 27.8±2.1 | 25.8±2.1 | 58.8±20.8 | 17.9±1.7 | 17.7±0.8 | 23.6±0.6 | 21.3±0.6 | 72.7±13.1 | 41.7±3.9 | 38.2±2.8 | 35.8±2.1 | 35.4±2.1 |
| | RecGNN | 33.4±14.5 | 28.0±6.5 | 18.7±4.1 | 18.2±4.4 | 16.8±4.3 | 22.7±6.2 | 15.9±1.5 | 16.8±1.4 | 21.5±1.6 | 19.5±1.4 | 32.3±14.9 | 26.8±5.8 | 25.2±5.3 | 24.1±5.2 | 24.0±5.2 |
| DFS (H) | GIN(E) | 41.5±7.5 | 30.4±2.3 | 20.0±3.1 | 19.5±2.6 | 17.8±2.5 | 25.0±3.7 | 15.8±0.6 | 16.8±0.4 | 22.7±0.7 | 20.6±0.6 | 39.6±9.1 | 28.3±3.1 | 26.1±3.7 | 25.3±2.9 | 25.2±2.9 |
| | PGN | 82.0±9.2 | 38.4±2.7 | 26.9±2.5 | 24.9±2.3 | 23.1±2.3 | 57.6±17.6 | 17.0±1.6 | 17.2±0.5 | 22.9±1.3 | 20.7±1.1 | 79.9±8.8 | 38.3±3.9 | 34.7±3.7 | 31.9±3.7 | 31.5±3.7 |
| | RecGNN | 48.3±19.1 | 22.8±4.7 | 13.5±4.6 | 13.1±4.1 | 12.0±3.6 | 35.3±17.7 | 13.5±2.6 | 14.7±1.9 | 19.4±2.1 | 17.9±1.7 | 50.2±21.7 | 21.8±3.2 | 19.4±3.8 | 18.7±3.6 | 18.5±3.5 |
| Dijkstra | Flood and Echo - 1 | 98.1±0.3 | 89.0±0.8 | 80.9±0.9 | 66.2±1.0 | 61.1±0.9 | 96.0±0.6 | 91.1±0.8 | 88.5±0.9 | 81.2±1.1 | 78.4±1.3 | 97.5±0.5 | 89.9±1.1 | 83.8±1.1 | 70.3±1.9 | 66.5±2.4 |
| | Flood and Echo - 2 | 98.9±0.1 | 94.0±0.6 | 88.9±1.1 | 78.2±2.0 | 73.9±2.3 | 97.9±0.3 | 94.6±0.4 | 92.4±0.7 | 82.5±0.9 | 78.1±1.4 | 98.6±0.2 | 94.1±0.7 | 89.8±0.9 | 73.7±1.9 | 66.4±3.9 |
| | Flood and Echo - 4 | 99.4±0.1 | 96.4±0.4 | 93.0±0.7 | 86.2±1.1 | 82.6±1.5 | 98.5±0.2 | 96.7±0.3 | 95.1±0.3 | 87.7±0.8 | 84.1±1.0 | 99.2±0.1 | 96.2±0.4 | 92.7±0.6 | 78.7±1.1 | 71.2±1.8 |
| | Flood and Echo - 8 | 99.4±0.0 | 96.6±0.3 | 93.1±0.7 | 85.4±2.2 | 81.1±3.0 | 98.5±0.3 | 96.4±0.5 | 95.1±0.7 | 87.5±1.5 | 84.3±2.1 | 99.3±0.1 | 96.4±0.5 | 92.9±1.3 | 78.3±2.6 | 70.1±2.3 |
| | Flood and Echo - 16 | 99.4±0.1 | 96.1±0.6 | 92.1±1.6 | 84.7±2.4 | 80.7±2.7 | 98.3±0.4 | 96.4±0.6 | 94.7±0.6 | 86.3±1.7 | 82.7±2.0 | 99.2±0.1 | 96.0±0.6 | 92.1±1.0 | 76.5±2.7 | 68.8±3.3 |
| | Flood and Echo - 0 | 99.4±0.1 | 96.1±0.7 | 92.2±1.2 | 84.1±2.5 | 79.8±3.2 | 98.4±0.4 | 96.4±0.7 | 94.2±1.4 | 85.3±3.6 | 81.7±4.0 | 99.2±0.2 | 95.9±1.0 | 91.7±2.1 | 76.4±5.0 | 69.1±5.8 |
| | GIN(E) | 98.0±0.2 | 89.8±1.1 | 84.3±1.6 | 75.8±2.2 | 72.8±2.3 | 95.4±0.7 | 85.0±1.4 | 79.9±1.9 | 61.4±4.0 | 52.6±4.1 | 97.4±0.4 | 81.6±1.3 | 70.4±2.6 | 46.5±3.7 | 39.9±3.6 |
| | PGN | 99.6±0.1 | 98.6±0.3 | 97.2±0.5 | 94.1±0.6 | 92.2±0.7 | 98.3±0.4 | 97.1±0.2 | 95.4±0.3 | 81.8±1.2 | 72.5±6.0 | 99.5±0.1 | 97.6±0.3 | 92.4±0.7 | 62.7±1.3 | 51.0±3.9 |
| | RecGNN | 98.5±1.6 | 86.8±15.4 | 76.0±22.1 | 63.7±27.7 | 60.6±27.7 | 95.8±4.2 | 89.2±14.1 | 83.9±18.9 | 71.4±20.4 | 67.3±17.7 | 98.0±1.9 | 90.4±9.7 | 85.0±10.0 | 60.2±4.4 | 50.0±3.6 |
| Dijkstra (H) | GIN(E) | 95.2±1.8 | 62.4±7.0 | 53.3±2.6 | 40.4±4.1 | 36.9±7.6 | 91.2±3.5 | 55.3±9.3 | 48.1±8.3 | 38.6±5.2 | 35.6±4.4 | 94.2±1.6 | 54.4±7.4 | 45.2±5.4 | 37.2±4.1 | 36.0±4.1 |
| | PGN | 99.3±0.1 | 94.2±2.5 | 92.0±2.3 | 87.1±2.7 | 84.5±3.4 | 97.8±0.2 | 85.8±6.0 | 80.9±7.0 | 60.5±8.3 | 52.4±8.3 | 99.2±0.1 | 84.9±6.8 | 72.8±6.9 | 50.8±4.6 | 46.4±3.1 |
| | RecGNN | 98.0±0.21 | 32.9±21.6 | 25.0±17.4 | 17.7±12.2 | 16.4±10.7 | 95.5±1.0 | 36.3±16.4 | 29.4±16.1 | 27.3±12.3 | 26.6±11.7 | 97.4±4.0 | 35.6±17.8 | 29.5±17.0 | 26.7±14.4 | 26.3±14.1 |
| Eccentricity | Flood and Echo - 1 | 99.8±0.1 | 99.9±0.1 | 98.9±0.3 | 99.4±0.2 | 81.7±9.4 | 100.0±0.0 | 88.6±0.8 | 93.2±6.0 | 36.2±6.6 | 29.2±6.0 | 100.0±0.0 | 80.8±11.3 | 73.7±6.6 | 0.0±0.0 | 0.0±0.0 |
| | Flood and Echo - 2 | 99.9±0.1 | 100.0±0.0 | 99.1±0.1 | 99.2±1.4 | 70.1±15.3 | 100.0±0.0 | 87.5±0.4 | 97.7±1.8 | 38.7±2.7 | 25.1±10.7 | 100.0±0.0 | 95.3±2.1 | 72.9±12.6 | 0.0±0.0 | 0.0±0.0 |
| | Flood and Echo - 4 | 99.9±0.2 | 99.7±0.5 | 98.5±1.5 | 98.6±2.4 | 73.4±13.5 | 100.0±0.0 | 88.5±1.4 | 96.0±4.6 | 40.4±3.8 | 22.8±9.3 | 100.0±0.0 | 93.5±6.7 | 73.4±16.3 | 0.0±0.0 | 0.0±0.0 |
| | Flood and Echo - 8 | 99.9±0.0 | 99.9±0.1 | 98.8±0.4 | 99.5±0.3 | 81.1±5.4 | 100.0±0.0 | 87.4±3.1 | 92.3±7.9 | 29.7±11.1 | 20.8±7.2 | 100.0±0.0 | 82.7±15.5 | 54.9±28.5 | 0.0±0.0 | 0.0±0.0 |
| | Flood and Echo - 16 | 99.8±0.0 | 99.5±1.2 | 98.0±2.1 | 95.8±7.8 | 66.9±16.1 | 100.0±0.0 | 88.4±2.2 | 95.7±2.6 | 36.5±6.2 | 29.4±6.2 | 100.0±0.0 | 89.1±9.4 | 69.6±11.0 | 0.0±0.0 | 0.0±0.0 |
| | Flood and Echo - 0 | 99.9±0.1 | 100.0±0.0 | 99.4±0.0 | 99.4±0.5 | 75.6±11.0 | 100.0±0.0 | 88.7±1.3 | 97.7±1.3 | 36.9±2.2 | 30.4±7.2 | 100.0±0.0 | 90.8±7.3 | 66.2±4.8 | 0.0±0.0 | 0.0±0.0 |
| | GIN(E) | 57.3±21.2 | 77.1±17.5 | 72.3±18.0 | 51.3±34.2 | 36.7±17.6 | 78.0±18.7 | 27.6±19.5 | 3.6±8.0 | 0.0±0.0 | 0.0±0.0 | 84.8±12.4 | 0.0±0.0 | 0.0±0.0 | 0.0±0.0 | 0.0±0.0 |
| | PGN | 100.0±0.0 | 100.0±0.0 | 100.0±0.0 | 100.0±0.0 | 64.6±14.9 | 100.0±0.0 | 93.8±2.1 | 100.0±0.1 | 25.6±7.5 | 5.2±3.3 | 100.0±0.0 | 100.0±0.0 | 76.9±19.8 | 0.0±0.0 | 0.0±0.0 |
| | RecGNN | 75.8±26.2 | 80.5±35.0 | 75.0±39.1 | 72.7±27.9 | 63.0±24.8 | 86.7±25.7 | 60.8±29.1 | 57.4±38.7 | 27.6±29.4 | 15.2±13.7 | 89.9±19.4 | 25.2±37.6 | 8.3±11.9 | 0.0±0.0 | 0.0±0.0 |
| Eccentricity (H) | GIN(E) | 25.3±41.0 | 23.8±39.0 | 26.1±36.8 | 17.1±32.9 | 16.0±21.7 | 25.3±42.2 | 19.0±18.8 | 18.6±18.9 | 4.6±8.9 | 9.8±10.2 | 24.8±42.5 | 17.0±12.5 | 3.0±5.8 | 0.0±0.0 | 0.0±0.0 |
| | PGN | 100.0±0.0 | 100.0±0.0 | 100.0±0.0 | 100.0±0.0 | 83.0±6.5 | 100.0±0.0 | 88.3±1.8 | 100.0±0.1 | 34.8±7.2 | 9.2±4.8 | 100.0±0.0 | 99.7±0.3 | 64.4±14.2 | 0.0±0.0 | 0.0±0.0 |
| | RecGNN | 95.0±6.3 | 96.6±3.6 | 95.8±4.6 | 93.4±10.3 | 72.1±20.9 | 99.0±1.2 | 66.4±22.4 | 46.2±40.9 | 14.1±6.3 | 8.3±4.9 | 99.6±0.6 | 51.0±36.0 | 19.4±11.7 | 0.0±0.0 | 0.0±0.0 |
| MIS | Flood and Echo - 1 | 91.3±0.3 | 87.4±0.2 | 87.7±0.1 | 88.1±0.4 | 87.3±0.2 | 92.6±0.3 | 91.9±0.2 | 91.6±0.2 | 92.3±0.3 | 92.0±0.2 | 92.7±0.1 | 91.7±0.2 | 91.4±0.2 | 91.4±0.2 | 91.3±0.2 |
| | Flood and Echo - 2 | 93.0±0.2 | 89.3±0.2 | 90.0±0.3 | 89.4±0.2 | 89.2±0.2 | 93.9±0.2 | 93.7±0.1 | 93.3±0.2 | 94.0±0.2 | 93.8±0.2 | 94.0±0.2 | 93.4±0.1 | 93.1±0.2 | 93.1±0.2 | 93.0±0.2 |
| | Flood and Echo - 4 | 98.9±0.2 | 97.3±0.3 | 97.6±0.3 | 95.7±0.4 | 96.8±0.4 | 99.2±0.1 | 99.3±0.1 | 99.2±0.1 | 99.1±0.1 | 99.1±0.1 | 99.4±0.2 | 99.2±0.1 | 99.1±0.1 | 99.1±0.1 | 99.1±0.1 |
| | Flood and Echo - 8 | 99.7±0.1 | 99.6±0.1 | 99.5±0.1 | 99.1±0.3 | 99.4±0.2 | 99.7±0.1 | 99.9±0.0 | 99.8±0.1 | 99.8±0.1 | 99.8±0.1 | 99.8±0.1 | 99.8±0.0 | 99.8±0.1 | 99.8±0.1 | 99.8±0.1 |
| | Flood and Echo - 16 | 99.6±0.1 | 99.5±0.2 | 99.4±0.2 | 98.7±0.4 | 99.1±0.3 | 99.7±0.1 | 99.9±0.0 | 99.7±0.1 | 99.8±0.1 | 99.7±0.1 | 99.8±0.1 | 99.8±0.1 | 99.8±0.1 | 99.8±0.1 | 99.7±0.1 |
| | Flood and Echo - 0 | 99.7±0.1 | 99.5±0.1 | 99.5±0.2 | 98.9±0.3 | 99.3±0.2 | 99.7±0.1 | 99.8±0.1 | 99.8±0.1 | 99.8±0.1 | 99.8±0.1 | 99.8±0.0 | 99.8±0.1 | 99.8±0.1 | 99.8±0.1 | 99.8±0.1 |
| | GIN(E) | 82.2±2.5 | 81.6±1.9 | 80.8±2.4 | 83.6±1.5 | 80.8±2.5 | 84.2±2.1 | 82.0±2.6 | 82.3±2.4 | 84.3±1.9 | 83.4±2.6 | 82.5±3.2 | 82.4±3.0 | 81.5±3.3 | 80.9±3.7 | 80.3±4.0 |
| | PGN | 99.8±0.1 | 99.6±0.2 | 99.5±0.2 | 98.8±0.6 | 98.9±0.5 | 99.8±0.1 | 99.4±0.2 | 98.8±0.6 | 95.8±2.6 | 93.3±4.4 | 99.9±0.1 | 99.8±0.1 | 99.5±0.2 | 99.3±0.3 | |
| | RecGNN | 93.6±2.2 | 90.0±2.3 | 90.1±2.5 | 87.9±1.9 | 88.2±2.6 | 93.3±2.2 | 92.6±2.6 | 92.2±2.9 | 91.8±3.3 | 91.4±3.5 | 94.3±2.0 | 93.4±2.0 | 93.0±2.5 | 92.5±3.0 | 92.1±3.4 |
| MIS (H) | GIN(E) | 79.9±2.9 | 79.9±2.2 | 78.2±2.7 | 83.4±0.9 | 79.2±1.6 | 83.1±1.9 | 79.5±3.4 | 79.8±3.3 | 83.2±2.2 | 81.8±1.9 | 80.6±3.5 | 80.6±3.5 | 79.8±3.6 | 78.9±3.7 | 78.2±3.7 |
| | PGN | 99.8±0.1 | 99.4±0.1 | 99.4±0.2 | 98.8±0.5 | 98.9±0.7 | 99.7±0.1 | 99.5±0.2 | 99.1±0.3 | 98.6±0.8 | 98.2±1.3 | 99.8±0.1 | 99.7±0.1 | 99.7±0.2 | 99.4±0.7 | 99.1±1.2 |
| | RecGNN | 92.2±0.7 | 88.7±1.7 | 88.3±2.8 | 85.6±3.5 | 84.7±5.5 | 92.5±2.1 | 91.7±2.2 | 89.8±4.6 | 88.0±6.5 | | 93.5±2.0 | | | 89.8±4.6 | 88.0±6.5 |
| MST | Flood and Echo - 1 | 86.4±0.2 | 67.7±0.7 | 63.4±0.9 | 53.1±1.0 | 49.1±1.0 | 83.0±0.7 | 69.0±1.0 | 66.7±1.2 | 60.7±1.2 | 62.4±1.2 | 87.1±0.5 | 72.7±1.2 | 68.4±1.2 | 63.4±1.5 | 62.5±1.6 |
| | Flood and Echo - 2 | 90.4±1.2 | 75.8±1.9 | 72.0±2.1 | 63.1±2.2 | 59.4±2.2 | 87.1±1.8 | 75.5±1.9 | 73.5±2.0 | 68.1±2.6 | 68.8±2.5 | 91.1±1.1 | 78.7±1.2 | 75.1±1.3 | 70.7±1.5 | 69.9±1.5 |
| | Flood and Echo - 4 | 94.0±0.8 | 82.0±1.3 | 78.2±1.5 | 69.1±2.1 | 65.3±2.4 | 92.2±1.3 | 80.6±1.6 | 77.9±1.8 | 71.9±2.6 | 71.0±3.4 | 94.4±0.7 | 82.8±1.4 | 78.6±2.0 | 72.7±3.4 | 71.3±3.8 |
| | Flood and Echo - 8 | 94.5±1.0 | 83.1±1.9 | 79.1±2.2 | 70.3±3.1 | 66.5±3.5 | 92.8±1.5 | 81.4±1.2 | 78.3±1.2 | 70.9±1.4 | 68.4±1.6 | 95.1±0.9 | 83.2±1.0 | 78.1±0.6 | 69.6±1.8 | 67.1±2.7 |
| | Flood and Echo - 16 | 94.6±0.7 | 82.0±1.9 | 77.7±2.1 | 68.5±2.3 | 64.1±2.6 | 91.0±1.1 | 77.6±1.3 | 76.6±1.9 | 70.2±1.6 | 68.3±1.4 | 95.2±0.6 | 82.8±1.2 | 77.8±1.4 | 70.2±1.3 | 68.1±1.2 |
| | Flood and Echo - 0 | 94.5±0.5 | 83.1±0.9 | 79.1±1.0 | 69.8±1.8 | 65.7±2.0 | 92.7±0.4 | 81.5±1.0 | 78.4±1.2 | 71.1±2.0 | 68.9±2.7 | 95.1±0.2 | 83.1±0.8 | 78.2±1.1 | 70.0±2.8 | 67.3±4.0 |
| | GIN(E) | 92.6±0.9 | 79.1±1.3 | 77.6±1.7 | 74.5±2.0 | 72.9±2.2 | 89.6±1.4 | 75.3±1.0 | 74.4±1.4 | 73.0±2.4 | 72.8±2.3 | 92.8±0.8 | 77.4±0.6 | 75.8±1.1 | 74.8±1.7 | 74.7±1.7 |
| | PGN | 97.3±0.4 | 89.1±1.6 | 84.6±1.7 | 75.7±2.0 | 71.9±2.1 | 96.8±1.0 | 82.5±2.4 | 77.6±2.6 | 67.4±3.1 | 65.1±3.3 | 97.4±0.5 | 85.2±1.5 | 78.5±1.4 | 68.7±1.6 | 66.8±0.9 |
| | RecGNN | 94.2±2.3 | 70.7±27.8 | 66.6±28.2 | 58.9±29.0 | 56.0±28.5 | 92.8±2.8 | 67.4±22.9 | 62.8±23.2 | 53.5±20.9 | 52.5±17.1 | 94.7±2.1 | 69.9±22.1 | 62.6±20.0 | 52.5±13.4 | 50.6±11.3 |
| MST (H) | GIN(E) | 89.6±1.7 | 51.6±4.5 | 49.5±4.3 | 45.0±4.2 | 43.2±4.0 | 86.0±2.1 | 54.9±6.2 | 52.7±6.5 | 50.9±6.4 | 54.1±6.9 | 91.1±1.5 | 58.4±5.9 | 56.4±5.6 | 55.0±5.6 | 54.9±5.5 |
| | PGN | 96.4±0.6 | 79.7±3.8 | 75.6±4.5 | 69.5±5.5 | 66.8±5.1 | 96.1±1.0 | 74.5±3.9 | 72.5±4.5 | 69.2±4.4 | 68.8±5.9 | 96.7±0.5 | 77.7±4.1 | 74.3±5.0 | 71.4±6.5 | 71.0±6.7 |
| | RecGNN | 87.5±2.4 | 29.0±6.7 | 25.7±6.6 | 21.3±6.4 | 20.1±6.3 | 82.0±4.0 | 32.0±7.3 | 29.6±6.0 | 24.9±7.3 | 28.8±8.7 | 88.2±2.1 | 34.2±8.4 | 31.9±7.1 | 28.0±7.2 | 27.8±7.3 |

Table 14: Results for the FE Net on the MIS task when the number of rounds is increased. We report node accuracy, SALSA-CLRS indicates that the number of phases matches the length of the algorithm trajectory.

| Model | Task | ER | | | | | WS | | | | | Delaunay | | | | |
|---|---|---|---|---|---|---|---|---|---|---|---|---|---|---|---|---|
| | | 16 | 80 | 160 | 800 | 1600 | 16 | 80 | 160 | 800 | 1600 | 16 | 80 | 160 | 800 | 1600 |
| Flood and Echo Net - SALSA-CLRS | MIS | 99.7±0.1 | 99.5±0.1 | 99.4±0.2 | 98.9±0.3 | 99.2±0.2 | 99.8±0.1 | 99.8±0.1 | 99.7±0.1 | 99.7±0.1 | 99.7±0.1 | 99.8±0.0 | 99.8±0.1 | 99.7±0.0 | 99.7±0.1 | 99.7±0.1 |
| Flood and Echo Net - 1 | MIS | 91.5±0.1 | 87.5±0.2 | 87.8±0.1 | 88.3±0.3 | 87.4±0.2 | 92.7±0.2 | 91.9±0.1 | 91.6±0.2 | 92.5±0.3 | 92.1±0.2 | 92.9±0.1 | 91.8±0.1 | 91.5±0.2 | 91.5±0.2 | 91.4±0.2 |
| Flood and Echo Net - 2 | MIS | 93.1±0.1 | 89.4±0.1 | 90.1±0.0 | 89.5±0.1 | 89.3±0.1 | 93.8±0.1 | 93.8±0.1 | 93.5±0.0 | 94.2±0.0 | 93.9±0.0 | 94.1±0.1 | 93.5±0.0 | 93.2±0.0 | 93.2±0.0 | 93.2±0.0 |
| Flood and Echo Net - 4 | MIS | 93.1±11.6 | 92.4±8.6 | 92.3±9.6 | 92.1±5.4 | 92.1±7.9 | 94.4±9.2 | 93.2±12.2 | 93.4±11.3 | 94.8±7.9 | 94.5±8.5 | 93.5±11.9 | 93.6±10.7 | 93.6±10.6 | 93.7±10.5 | 93.7±10.4 |
| Flood and Echo Net - 8 | MIS | 99.7±0.1 | 99.6±0.1 | 99.5±0.2 | 99.0±0.3 | 99.3±0.2 | 99.8±0.1 | 99.9±0.0 | 99.8±0.1 | 99.8±0.1 | 99.9±0.1 | 99.8±0.0 | 99.8±0.1 | 99.8±0.1 | 99.8±0.1 | 99.8±0.1 |
| Flood and Echo Net - 16 | MIS | 99.7±0.2 | 99.6±0.1 | 99.5±0.1 | 99.2±0.2 | 99.4±0.1 | 99.7±0.1 | 99.9±0.1 | 99.8±0.1 | 99.8±0.1 | 99.8±0.1 | 99.8±0.1 | 99.8±0.1 | 99.8±0.1 | 99.8±0.1 | 99.8±0.1 |

# K  Datasets

## K.1  Algorithmic Datasets

For all the below tasks, we use train set, validation set, and test set sizes of 1024, 100, and 1000, respectively. The sizes of the respective graphs in the train, validation, and test sets are 10, 20, and 100. Performance

Table 15: Results for the FE Net on the MIS task when the number of rounds is increased. We report graph accuracy, SALSA-CLRS indicates that the number of phases matches the length of the algorithm trajectory.

| Model | Task | ER | | | | | WS | | | | | Delaunay | | | | |
|---|---|---|---|---|---|---|---|---|---|---|---|---|---|---|---|---|
| | | 16 | 80 | 160 | 800 | 1600 | 16 | 80 | 160 | 800 | 1600 | 16 | 80 | 160 | 800 | 1600 |
| Flood and Echo Net - SALSA-CLRS | MIS | 98.3 ±0.6 | 90.0 ±2.0 | 81.0 ±4.4 | 24.3 ±10.4 | 10.9 ±6.6 | 98.3 ±0.7 | 95.1 ±1.6 | 85.6 ±1.7 | 48.0 ±7.4 | 24.1 ±7.3 | 98.8 ±0.3 | 93.7 ±1.2 | 87.4 ±1.2 | 48.6 ±7.0 | 24.3 ±4.7 |
| Flood and Echo Net - 1 | MIS | 40.4 ±2.1 | 0.1 ±0.1 | 0.0 ±0.0 | 0.0 ±0.0 | 0.0 ±0.0 | 40.3 ±2.4 | 1.0 ±0.3 | 0.0 ±0.0 | 0.0 ±0.0 | 0.0 ±0.0 | 44.1 ±1.2 | 1.2 ±0.1 | 0.0 ±0.0 | 0.0 ±0.0 | 0.0 ±0.0 |
| Flood and Echo Net - 2 | MIS | 49.1 ±2.0 | 0.3 ±0.2 | 0.0 ±0.0 | 0.0 ±0.0 | 0.0 ±0.0 | 46.3 ±1.6 | 3.1 ±0.6 | 0.0 ±0.0 | 0.0 ±0.0 | 0.0 ±0.0 | 49.1 ±1.7 | 2.2 ±0.5 | 0.1 ±0.1 | 0.0 ±0.0 | 0.0 ±0.0 |
| Flood and Echo Net - 4 | MIS | 70.9 ±40.0 | 25.0 ±17.2 | 11.3 ±9.0 | 0.0 ±0.0 | 0.0 ±0.0 | 70.9 ±40.0 | 49.7 ±33.3 | 29.6 ±23.5 | 1.2 ±1.2 | 0.0 ±0.0 | 73.0 ±41.2 | 46.7 ±31.4 | 27.3 ±21.5 | 0.9 ±0.9 | 0.0 ±0.0 |
| Flood and Echo Net - 8 | MIS | 98.5 ±0.5 | 90.3 ±3.6 | 82.6 ±5.4 | 25.1 ±11.0 | 12.9 ±10.6 | 98.7 ±0.4 | 95.9 ±1.8 | 86.8 ±4.0 | 51.4 ±12.8 | 27.5 ±15.1 | 99.0 ±0.4 | 94.5 ±1.5 | 88.7 ±3.5 | 52.5 ±13.1 | 28.9 ±17.2 |
| Flood and Echo Net - 16 | MIS | 98.2 ±0.6 | 91.8 ±2.6 | 83.6 ±4.4 | 33.1 ±9.6 | 16.4 ±8.8 | 98.2 ±0.6 | 95.8 ±1.6 | 87.9 ±3.7 | 56.2 ±9.3 | 33.6 ±11.5 | 98.8 ±0.6 | 94.8 ±1.1 | 88.9 ±3.2 | 54.0 ±10.9 | 31.5 ±14.1 |

Table 16: Results for the FE Net on the Dijkstra task when the number of rounds is increased. We report node accuracy, SALSA-CLRS indicates that the number of phases matches the length of the algorithm trajectory.

| Model | Task | ER | | | | | WS | | | | | Delaunay | | | | |
|---|---|---|---|---|---|---|---|---|---|---|---|---|---|---|---|---|
| | | 16 | 80 | 160 | 800 | 1600 | 16 | 80 | 160 | 800 | 1600 | 16 | 80 | 160 | 800 | 1600 |
| Flood and Echo Net - SALSA-CLRS | MIS | 99.5 ±nan | 97.0 ±nan | 93.7 ±nan | 86.8 ±nan | 83.4 ±nan | 98.6 ±nan | 96.9 ±nan | 95.0 ±nan | 85.2 ±nan | 79.3 ±nan | 99.3 ±nan | 96.5 ±nan | 92.6 ±nan | 72.4 ±nan | 61.4 ±nan |
| Flood and Echo Net - 1 | MIS | 98.3 ±0.3 | 89.0 ±0.9 | 80.2 ±1.1 | 64.8 ±1.0 | 59.5 ±0.9 | 97.6 ±0.3 | 88.1 ±0.7 | 91.0 ±0.7 | 76.5 ±1.4 | 71.7 ±2.0 | 97.8 ±0.3 | 90.4 ±1.3 | 83.2 ±1.3 | 62.6 ±2.5 | 56.8 ±3.6 |
| Flood and Echo Net - 2 | MIS | 99.0 ±0.1 | 93.4 ±0.2 | 87.3 ±0.5 | 74.4 ±0.6 | 69.5 ±0.5 | 97.6 ±0.3 | 93.8 ±0.4 | 91.4 ±0.4 | 80.6 ±1.4 | 76.4 ±1.5 | 98.7 ±0.2 | 93.4 ±0.4 | 88.2 ±0.8 | 72.2 ±1.0 | 66.0 ±1.1 |
| Flood and Echo Net - 4 | MIS | 99.5 ±0.0 | 96.5 ±0.2 | 92.8 ±0.3 | 85.0 ±1.1 | 81.1 ±1.2 | 98.6 ±0.2 | 96.5 ±0.3 | 94.7 ±0.5 | 84.9 ±1.9 | 80.5 ±2.3 | 99.3 ±0.0 | 96.1 ±0.4 | 92.1 ±0.6 | 74.3 ±1.3 | 65.4 ±2.5 |
| Flood and Echo Net - 8 | MIS | 99.6 ±0.1 | 96.7 ±0.3 | 93.1 ±0.4 | 85.6 ±1.0 | 81.6 ±1.4 | 98.7 ±0.2 | 96.5 ±0.4 | 94.5 ±0.4 | 83.8 ±1.9 | 78.7 ±2.5 | 99.3 ±0.1 | 95.8 ±0.6 | 90.7 ±1.2 | 68.7 ±4.9 | 59.6 ±5.5 |
| Flood and Echo Net - 16 | MIS | 99.5 ±0.1 | 96.6 ±0.4 | 93.0 ±0.7 | 85.2 ±1.3 | 81.1 ±1.7 | 98.7 ±0.2 | 96.6 ±0.5 | 94.7 ±0.9 | 84.4 ±2.9 | 79.7 ±3.9 | 99.4 ±0.1 | 96.2 ±0.6 | 91.5 ±1.6 | 72.2 ±6.4 | 63.5 ±7.4 |

Table 17: Results for the FE Net on the Dijkstra task when the number of rounds is increased. We report graph accuracy, SALSA-CLRS indicates that the number of phases matches the length of the algorithm trajectory.

| Model | Task | ER | | | | | WS | | | | | Delaunay | | | | |
|---|---|---|---|---|---|---|---|---|---|---|---|---|---|---|---|---|
| | | 16 | 80 | 160 | 800 | 1600 | 16 | 80 | 160 | 800 | 1600 | 16 | 80 | 160 | 800 | 1600 |
| Flood and Echo Net - SALSA-CLRS | MIS | 93.1 ±nan | 15.5 ±nan | 0.7 ±nan | 0.0 ±nan | 0.0 ±nan | 78.8 ±nan | 13.9 ±nan | 0.6 ±nan | 0.0 ±nan | 0.0 ±nan | 89.9 ±nan | 9.8 ±nan | 0.0 ±nan | 0.0 ±nan | 0.0 ±nan |
| Flood and Echo Net - 1 | MIS | 76.7 ±2.7 | 1.0 ±0.4 | 0.0 ±0.0 | 0.0 ±0.0 | 0.0 ±0.0 | 56.8 ±4.0 | 1.4 ±0.6 | 0.0 ±0.0 | 0.0 ±0.0 | 0.0 ±0.0 | 71.7 ±2.7 | 0.2 ±0.2 | 0.0 ±0.0 | 0.0 ±0.0 | 0.0 ±0.0 |
| Flood and Echo Net - 2 | MIS | 86.3 ±1.1 | 3.7 ±0.5 | 0.0 ±0.1 | 0.0 ±0.0 | 0.0 ±0.0 | 68.5 ±2.4 | 3.5 ±0.7 | 0.0 ±0.0 | 0.0 ±0.0 | 0.0 ±0.0 | 81.7 ±2.3 | 1.1 ±0.4 | 0.0 ±0.0 | 0.0 ±0.0 | 0.0 ±0.0 |
| Flood and Echo Net - 4 | MIS | 92.2 ±0.6 | 12.7 ±1.1 | 0.4 ±0.2 | 0.0 ±0.0 | 0.0 ±0.0 | 79.4 ±2.6 | 11.5 ±1.9 | 0.2 ±0.1 | 0.0 ±0.0 | 0.0 ±0.0 | 89.5 ±0.2 | 6.6 ±1.9 | 0.0 ±0.0 | 0.0 ±0.0 | 0.0 ±0.0 |
| Flood and Echo Net - 8 | MIS | 93.4 ±1.0 | 14.1 ±1.7 | 0.5 ±0.3 | 0.0 ±0.0 | 0.0 ±0.0 | 80.8 ±2.4 | 11.2 ±2.1 | 0.2 ±0.1 | 0.0 ±0.0 | 0.0 ±0.0 | 90.1 ±1.5 | 5.8 ±1.3 | 0.0 ±0.0 | 0.0 ±0.0 | 0.0 ±0.0 |
| Flood and Echo Net - 16 | MIS | 92.8 ±1.4 | 15.2 ±3.3 | 0.6 ±0.3 | 0.0 ±0.0 | 0.0 ±0.0 | 80.5 ±3.2 | 13.2 ±3.6 | 0.3 ±0.2 | 0.0 ±0.0 | 0.0 ±0.0 | 90.1 ±1.1 | 8.0 ±2.3 | 0.1 ±0.1 | 0.0 ±0.0 | 0.0 ±0.0 |

on this test set demonstrates the model's ability to extrapolate to larger graph sizes. Note that many of the tasks only require the output modulo 2. We reduce the problem to this specific setting so that all numbers involved in the computation stay within the same range, as otherwise, the values have to be interpreted almost in a symbolic way, which is very challenging for learning-based models.

**PrefixSum Task**   (Grötschla et al., 2022) Each graph in this dataset is a path graph where each node has a random binary label with one marked vertex at one end, which indicates the origin. The objective of this task is to predict whether the PrefixSum from the marked node to the node in consideration is divisible by 2.

**Distance Task**   (Grötschla et al., 2022) In this task every graph is a random graph of $n$ nodes with a source node being distinctly marked. The objective of this task is to predict for each node whether its distance to the source node is divisible by 2.

**Path Finding Task**   (Grötschla et al., 2022) In this task the dataset consists of random trees of $n$ nodes with two distinct vertices being marked separately. The objective of this task is to predict for each node whether it belongs to the shortest path between the 2 marked nodes.

### K.2   Expressive Datasets

**Skip Circles**   (Chen et al., 2023) This dataset consists of CSL(Circular Skip Link) graphs denoted by $G_{n,k}$, which is a graph of size $n$, numbered 0 to $n - 1$, where there exists an edge between node $i$ and node $j$ iff $|i - j| \equiv 1$ or $k \pmod{n}$. $G_{n,k}$ and $G_{n',k'}$ are only isomorphic when $n = n'$ and $k \equiv \pm k' \pmod{n}$. Here,

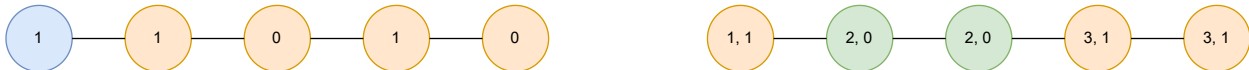

Figure 8: Example graph from the PrefixSum task. The left graph represents the input graph with a binary value associated with each node and the blue node being the starting node. The right graph represents the ground truth solution, each node contains two values the cumulative sum and the desired result which is the cumulative sum modulo 2.

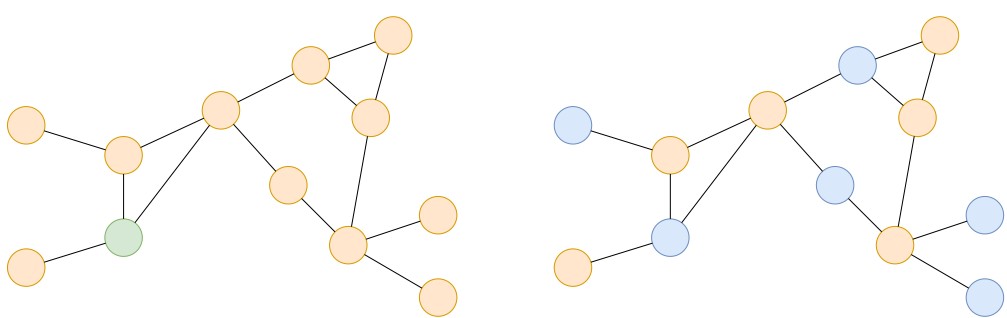

Figure 9: Example graph from the distance task. The green node in the left graph (input graph) represents the source node, and the remaining nodes are unmarked. On the right graph (ground truth) all orange nodes are at an odd distance away from the source while the blue nodes are at an even distance away from the source.

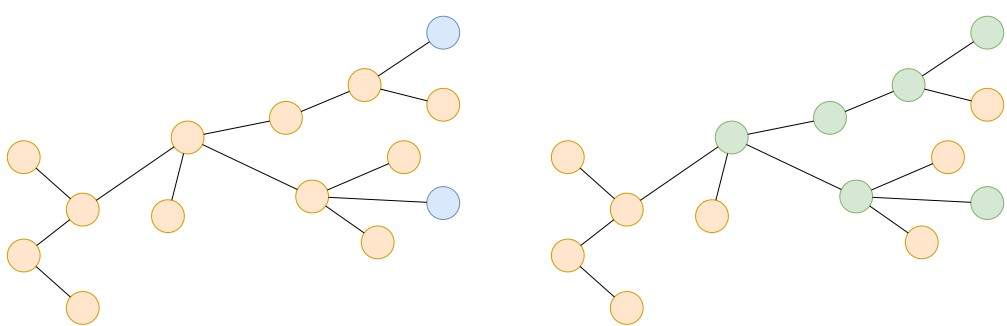

Figure 10: Example graph from the pathfinding task. The left graph represents the input graph, where the blue nodes are the marked nodes. The right is the corresponding solution, where the path between the marked nodes is highlighted in green.

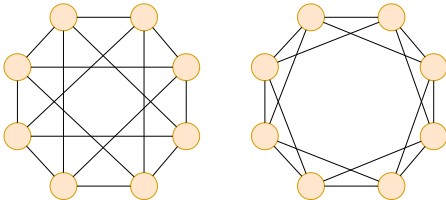

Figure 11: Example graphs from the Skip Circles dataset, namely $G_{n,5}$ and $G_{n,2}$ on the left and the right respectively.

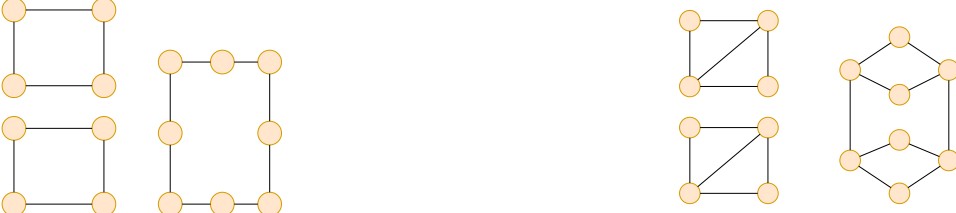

Figure 12: Counter-examples which MPNNs cannot distinguish from Garg et al. (2020), they cannot distinguish among the graphs in each example.

the number of graphs in train, validation, and test are all 10. We can see an example of this construction in Figure 11.

We follow the setup of Chen et al. (2023) where we fix $n = 41$ and set $k \in \{2, 3, 4, 5, 6, 9, 11, 12, 13, 16\}$. Each $G_{n,k}$ forms a separate isomorphism class, and the aim of the classification task is to classify the graph into its isomorphism class by the skip cycle length. Since 1-WL is unable to classify these graphs, we can see in table 3 that the GIN model cannot get an accuracy better than random guessing (10%).

**Limits1 and Limits2** (Garg et al., 2020) This dataset consists of two graphs from Garg et al. (2020) that, despite having different girth, circumference, diameter, and total number of cycles, cannot be distinguished by 1-WL. For each example, the aim is to distinguish among the disjoint graphs on the left versus the larger component on the right. The specific constructions can be seen in Figure 12.

**4-Cycles** (Loukas, 2020) This dataset introduced by Loukas (2020) originates from a construction by Korhonen & Rybicki (2017) in which two players Alice and Bob each start with a complete bipartite graph of $p = \sqrt{q}$ nodes which are numbered from 1 to $2p$ and a hidden binary key with size being $|p^2|$. The nodes from each graph with the same numbers are connected together. Each player then uses their respective binary keys to remove edges, each bipartite edge corresponding to a zero bit is removed and remaining edges are substituted by a path of length $k/2 - 1$, we use $k = 4$. The task is to determine if the resulting graph has a cycle of length $k$. In our implementation the number of train, validation and test graphs we consider are all 25. For a depiction of the construction refer to Figure 13.

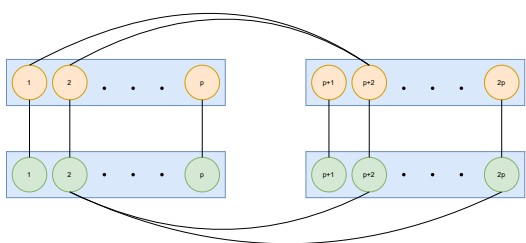

Figure 13: Example construction of Loukas (2020), where k=4.

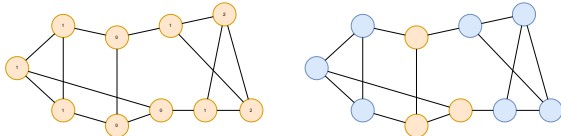

Figure 14: The graphs represent an instance from LLC and Triangles dataset respectively. For the LLC graph(left), each label denotes the ground truth for the graph while for the Triangles(right) graph, the blue nodes are ones which are a part of a triangle, while the orange nodes are not part of any triangle.

**LLC**   (Sato et al., 2021) This dataset is comprised of random 3-regular graphs and the task is to determine for each node its local clustering coefficient (Watts & Strogatz, 1998) which informally is the number of triangles the vertex is part of. The training and test set are both comprised of a 1000 graphs. The graphs in the train set have 20 nodes, while the graphs in the test set have a 100 nodes testing extrapolation. An example graph from this dataset can be seen in Figure 14.

**Triangles**   (Sato et al., 2021) This dataset akin to the previous contains random 3-regular graphs with the same train/test split and graph sizes. The task here is to classify each node as being part of a triangle or not. An example graph from this dataset can be seen in Figure 14.

