# OpenReview forum: "Less Can Be More: Rethinking Message-Passing for Algorithmic Alignment on Graphs"
_TMLR — Rejected by TMLR_

### Review · Reviewer_eg41 · 2025-08-25

**Summary Of Contributions:**

This paper proposes the Flood and Echo Net as a new graph neural network (GNN) architecture. Flood and Echo Net involves a forward phase that passes messages from an origin node to progressively more distant neighbors, and a backward phase that passes messages from the neighbors back to the origin node. Analysis shows that the Flood and Echo Net passes fewer messages than the standard Message-Passing Neural Networks (MPNNs) GNNs, and experiments show that the Flood and Echo Net generalizes better than the MPNNs GNNs for graph algorithms.

**Additional Comments:**

NA

**Audience:**

No

**Audience Explanation:**

As explained in the evidence part, the paper has critical issues in the novelty over existing work and evaluation method, which limits its scientific value. Moreover, the paper several additional issues.

I1: The analysis before Theorem 4.1 says that MPNNs passes O(Dm) messages but Theorem 4.1 says O(nm) messages. This inconsistency needs to be explained.

I2: It is unclear that how the number of phases T is set for the Flood and Echo Net.

I3: For the random and fixed variants of the Flood and Echo Net, in the experiments, it is unclear how many origin nodes are selected and how these nodes are selected.

I4: (small detail) m is used in the introduction without definition.

**Broader Impact Concerns:**

There are no ethical concerns for this work.

**Claims And Evidence:**

No

**Claims Explanation:**

The authors claim that the Flood and Echo Net is more expressive and passes fewer messages than standard MPNNs GNNs. There are three problems in these claims.

P1: The Flood and Echo Net resembles sampling-based GNN training, which involves a sampling phase that samples progressively more distant neighbors from an origin node, and an aggregation phase that passes messages from the neighbors back to the origin node. Please see the famous GraphSAGE [a] and related papers. These sampling-based GNNs also pass fewer messages than standard MPNNs GNNs. What are the differences between the Flood and Echo Net and sampling-based GNNs.

[a] Inductive Representation Learning on Large Graphs

P2: For GNN evaluation, there are several well-established tasks, e.g., node classification, link prediction, and graph classification, and well-known benchmark datasets (see those in the GraphSAGE paper). However, the paper lacks the results for these tasks and datasets.

P3: The paper claims that the Flood and Echo Net passes fewer messages than standard MPNNs GNNs. However, there are no efficiency evaluation to show how the Flood and Echo Net improves training or inference speed.

**Requested Changes:**

R1: Discuss sampling-based GNNs and explain the differences of the Flood and Echo Net from them.

R2: Conduct experiments using popular GNN tasks and datasets. Evaluate the efficiency of the Flood and Echo Net and compare with sampling-based GNNs.

R3: Fix the clarity and detail issues pointed out in the reviews.

---

> ### Author Response · Authors · 2025-08-26
>
> We thank the reviewer for his insights and provide the requested clarifications in the following. Note that we will upload a revised version once all reviews have been submitted as recommended by TMLR.
>
> > The authors claim that the Flood and Echo Net is more expressive and passes fewer messages than standard MPNNs GNNs. There are three problems in these claims.
>
> We are happy to discuss the mentioned items, but would like to point out that according to our understanding they are not directly related to “claim that the Flood and Echo Net is more expressive and passes fewer messages”, which we discuss in detail in our section 4. Instead the focus of the mentioned items is about the architecture design itself and its empirical analysis.
>
> > R1: Discuss sampling-based GNNs and explain the differences of the Flood and Echo Net from them.
> P1: The Flood and Echo Net resembles sampling-based GNN training, which involves a sampling phase that samples progressively more distant neighbors from an origin node, and an aggregation phase that passes messages from the neighbors back to the origin node. Please see the famous GraphSAGE [a] and related papers. These sampling-based GNNs also pass fewer messages than standard MPNNs GNNs. What are the differences between the Flood and Echo Net and sampling-based GNNs.
>
> First, we would like to clarify what is meant with sampling based GNNs to avoid any misunderstandings. As we understand it, in a sampling based GNN, instead of training on the entire graph during training (because this is computationally expensive), one instead first samples the nodes A of interest, and then as a second step the relevant neighbors B for the computation (all nodes within K hops of a node in A). [Upon checking the GraphSage paper we were surprised that this is actually not the focus of the original paper, it is mentioned and trained like this ie Appendix A. But the main paper outlines the computation to be something we would refer to today as an MPNN with K hops (the sampling in the main paper refers to restricting to nodes within K hop neighborhoods).] If this is not what was meant by sampling-based GNNs please let us know.
>
> Our method defines a new architecture that, given a graph convolution, leads to a different computation (wrt. to standard MPNN computations). Whereas sampling-based GNNs, given a graph convolution, preserve the original MPNN computation but can do it in a more memory efficient way (if the graphs in question are very big and the k is small). Figure 5 in the appendix shows how the computation graph of an MPNN differs from the computation graph of an FE Net.
>
> > R2: Conduct experiments using popular GNN tasks and datasets. Evaluate the efficiency of the Flood and Echo Net and compare with sampling-based GNNs.
> P2: For GNN evaluation, there are several well-established tasks, e.g., node classification, link prediction, and graph classification, and well-known benchmark datasets (see those in the GraphSAGE paper). However, the paper lacks the results for these tasks and datasets.
>
> In our work, we focus on presenting and evaluating our method to show where and why the algorithmic alignment on the level of message passing can be beneficial. While the proposed architecture might be applicable for other tasks that we have not considered, we focus on the setting where we think it has its largest potential and regular MPNNs are not an ideal choice (note the last paragraph of section 5.1). Specifically,  we focus a) on algorithmic tasks (can the alignment be helpful) where b) information might be far away (FE Net uses fewer messages) and c) have to generalize to larger instances (mechanism is inherently helpful). On the other hand, the architectural bias of regular MPNNs is likely to be more helpful for the mentioned benchmarks, where the connections and relations to nearby neighbors seem to be most important. However, combining these paradigms might be an interesting direction for future research on that topic.
>
> > P3: The paper claims that the Flood and Echo Net passes fewer messages than standard MPNNs GNNs. However, there are no efficiency evaluation to show how the Flood and Echo Net improves training or inference speed.
>
> Note that we discuss and prove the claims of message complexity of our method in section 4.1. The claim is about using $\textbf{fewer messages}$, which we believe to be of use as it is algorithmically aligned and also leads to improved results. This is different from “training or inference speed” which is more dependent on runtime and implementation. We have a more detailed discussion on this topic in Appendix H.

---

> ### Author Response · Authors · 2025-08-26
>
> > Would at least some individuals in TMLR's audience be interested in knowing the findings of this paper?: No
> As explained in the evidence part, the paper has critical issues in the novelty over existing work and evaluation method, which limits its scientific value. Moreover, the paper several additional issues.
>
> We are happy to discuss, clarify and if need be adjust the manuscript to make sure there are not critical issues. We strongly believe that our presented method presents a novel architecture (which is algorithmically aligned) with sound claims and empirical evidence which would be of interest for the TMLR community.
>
> >I1: The analysis before Theorem 4.1 says that MPNNs passes O(Dm) messages but Theorem 4.1 says O(nm) messages. This inconsistency needs to be explained.
>
> We noticed that there might be a bit of confusion because we use D for the number of hops (which is often also used for the graph diameter), in the revised version we will use K in the explanation instead to make it more clear.
>
> The analysis before the theorem is more general: MPNNs always exchange O(Km) messages to propagate information for K hops. On the other hand theorem 4.1 argues about the existence of specific tasks, i.e. where the graph diameter D is linear in the number of nodes. In these instances, information needs to be propagated for K=D=O(n) hops. The FE Net can do this with O(m) messages and MPNNs require O(Km) = O(nm) messages.
>
> >I2: It is unclear that how the number of phases T is set for the Flood and Echo Net.
>
> The number of Phases T is a hyperparameter which needs to be set (similar to how the number of message passing rounds in general MPNNs is also a hyperparameter). In Experiment 5.1 we execute 2 phases (mentioned on page 7), for 5.2 we do a search over the best choice between 1 and 16 (p9) and there is an ablation shown in Figure 4.
>
> >I3: For the random and fixed variants of the Flood and Echo Net, in the experiments, it is unclear how many origin nodes are selected and how these nodes are selected.
>
> Both the fixed and random variants always only select $\textbf{one node}$ as the origin. For fixed, the node $\textbf{is given}$ by the task, for random it is chosen $\textbf{uniformly at random from the nodes}$ in the graph. We can make this more explicit in the experiment section, i.e. by referring to the definition of the modes on page 4: “In the fixed mode, the origin is given or defined by the problem instance, i.e. by a marked source node specific to the task. In contrast, the random mode selects an origin uniformly at random from all nodes”.
>
> > 4: (small detail) m is used in the introduction without definition.
>
> Thank you for pointing this out, we will include this in the updated manuscript.

---

### Review · Reviewer_EExM · 2025-09-03

**Summary Of Contributions:**

## Summary

This paper introduces "Flood and Echo" (FE), a novel message-passing framework for graph neural networks that rethinks the traditional message-passing paradigm to better align with algorithmic reasoning on graphs. The key contributions are:
1. A new message-passing architecture that operates in "phases" rather than sequential "rounds," where each phase consists of a flood step (information propagation from a designated node) followed by an echo step (aggregation back to the source)
2. Theoretical analysis demonstrating that FE achieves $\mathcal{O}(m)$ message complexity for certain algorithmic tasks (where $m$ is the number of edges), compared to the $\mathcal{O}(Dm)$ complexity of traditional message-passing neural networks (MPNNs), where $D$ is the number of layers.
3. Empirical validation showing superior extrapolation capabilities, where models trained on small graphs ($n=10$) successfully generalize to much larger graphs ($n=100$), outperforming standard GNN approaches
4. Evidence that FE achieves better algorithmic alignment, meaning the neural network's information processing more closely mirrors how classical graph algorithms operate

## Strengths
1. Strong theoretical foundation with clear complexity analysis
2. Compelling empirical results on multiple algorithmic tasks (PrefixSum, Distance, Path Finding, MIS)
3. Novel perspective in MPNNs that scales beautifully to larger graphs.

## Limitations
1. Limited discussion of computational overhead despite message complexity claims
2. Unclear how the approach scales to more complex real-world graphs beyond algorithmic tasks (for example ogbn-arxiv, ogbn-papers100M, ogbg-ppa, ogb-lsc)
3. The "all variation" of FE (executing n sequential executions) may undermine the claimed efficiency benefits.

**Additional Comments:**

The most promising aspect is the demonstrated extrapolation capability, which addresses a significant limitation of standard GNNs. If the authors can better isolate and explain why FE achieves better extrapolation (is it the architecture itself or simply using more computational steps?), this could be a valuable contribution.

**Audience:**

Yes

**Audience Explanation:**

1. This paper proposes a novel message-passing framework with theoretical analysis of its computational properties
2. This paper provides insights into extrapolation capabilities of GNNs, a topic of interest for researchers working on graph foundation models.

**Claims And Evidence:**

No

**Claims Explanation:**

No, the claims are partially supported but with significant gaps in evidence that reduce confidence in the main contributions. While the paper presents interesting theoretical claims and some empirical results, several critical aspects lack sufficient evidence:

1.  The paper claims $\mathcal{O}(m)$ message complexity for FE Net versus $\mathcal{O}(nm)$ for traditional MPNNs. However, the "all variation" of FE requires n sequential executions, effectively bringing the complexity back to $\mathcal{O}(nm)$. This undermines the primary theoretical advantage claimed. The paper acknowledges this ("The all variation of Flood and Echo performs $n$ single executions in a sequential order") but doesn't adequately address how this affects the claimed benefits.
2. The paper uses "rounds" for traditional MPNNs but "phases" for FE, creating potential confusion about direct comparability. The relationship between one FE phase and multiple MPNN rounds isn't clearly quantified in computational terms.
3. The paper frames as a computationally efficient alternative to standard MPNNs. However, comparison using large-scale benchmarks like ogbn-arxiv, ogbn-papers100M, ogbn-MAG, ogbg-ppa, ogb-lsc datasets is not shown.

**Requested Changes:**

Please address the limitations and the concerns on partially supported claims.

---

> ### Author Response · Authors · 2025-09-08
>
> > Limited discussion of computational overhead despite message complexity claims
>
> >2)The paper uses "rounds" for traditional MPNNs but "phases" for FE ...
>
> >The "all variation" of FE ...
>
> > 1)The paper claims O(m) message complexity for FE Net versus O(nm) for traditional MPNNs ...
>
> We appreciate the reviewers input on the complexity claims and are eager to improve the current presentation. We group the above points together into a single response.
>
> In the main part of the paper, namely in section 4.1 we explain the most important complexity claims (which we also revised slightly due to the inputs of reviewer eg41). However, we already include a more detailed explanation in Appendix H (also referred to in this section). Here we discuss: MPNN Round, FE Phase (H1) as well as computational overhead and time measurements (H2). We would like to point out the difference between message complexity, which is theoretically verified and can have alignment benefits and overall runtime (on a device) which depends on implementation and parallelism.
>
> There we also mention the complexity of the “all variation” and now add a sentence to highlight it explicitly. Whenever we list and compare the all variation (Table 1, Table 3) we specifically state the O(nm) complexity. Note that the $n$ executions of the all variants could actually be parallelized (although this would not impact message complexity). However, because we believe that the other variants are more practical anyway, we did not investigate this further and put less focus on the all variant in general.
> Overall, we believe we provide an in-depth discussion on the raised points, if the reviewer would like us to further expand on some particular point or further highlight a specific fact in the main text we are happy to revise our draft.
>
> >Unclear how the approach scales to more complex real-world graphs beyond algorithmic tasks (for example ogbn-arxiv, ogbn-papers100M, ogbg-ppa, ogb-lsc)
>
> > 3)The paper frames as a computationally efficient alternative to standard MPNNs. However, comparison using large-scale benchmarks like ogbn-arxiv, ogbn-papers100M, ogbn-MAG, ogbg-ppa, ogb-lsc datasets is not shown.
>
> As pointed out in the discussion on complexity, it is not our intention to claim that our method is computationally (as in runtime) more efficient. Rather, the realignment of the computation flow requires less messages overall which can yield qualitative improvements.
>
> In our work we present an alternative to classical MPNN based on algorithmic alignment on the level of message passing. We think this is interesting, particularly in the setting we discuss in the paper where information might be far away, or structured computation may be more beneficial for generalization. In these settings the classical MPNN baseline does not do particularly well. However, that does not mean that the FE mechanism is generally the right alternative in any setting (and neither is our intention to claim this). Classical MPNN do provide a good inductive bias for other tasks, where the immediate neighborhood information is most important, as is the case in many of the mentioned datasets. As such we see limited benefit in evaluating under these settings with the current formulation, as it is not at all the motivation of why we desire the algorithmic alignment. We discuss this in the end of section 5.1 (right before 5.2), if need be we could also revise our draft further to make it more explicit. It might be an interesting direction for future work on how to balance the two paradigms. However, we have revised our formulation in the main paper to make this distinction more clear.
>
> > Would at least some individuals in TMLR's audience be interested in knowing the findings of this paper?: Yes
>
> We appreciate the positive feedback and hope our work can lead to more interesting insights into graph mechanism designs.
>
>
> > The most promising aspect is the demonstrated extrapolation capability ...
>
> Indeed, the ability to generalize the learned behaviour across several graph sizes is one of the most interesting aspects we study in the paper. To further comment on this, consider the evaluation setup in 5.1. Essentially, the GIN baseline is the equivalent of an MPNN that does not have sufficient computation steps. So clearly some depth is required, otherwise you miss out on important information. However, both the PGN and RecGNN adjust the computation depending on the graph size (RecGNN and FE actually make use of the same graph convolutions). Therefore, asymptotically speaking they have the same computation depth as FE, however, they actually do send more messages in the process, so there is even an argument that they actually perform more computation steps. As such, the main to these baselines is the architecture (revised computation flow) of the FE net. While this is not conclusive evidence, we believe it provides some empirical evidence in favour of the proposed architecture.

---

### Review · Reviewer_ictX · 2025-09-04

**Summary Of Contributions:**

The paper introduces the Flood and Echo Net (FE Net), a new GNN architecture inspired by distributed algorithms, particularly flooding and echo protocols. The method replaces the standard message-passing operations in GNNs with a wave-like propagation pattern, where information “floods” outward from an origin node and is then “echoed” back. The authors argue that this alignment with distributed algorithms improves efficiency, expressivity, and generalization to larger graph sizes, and they provide both theoretical guarantees and empirical validation on algorithmic benchmarks (SALSA-CLRS, synthetic tasks).

**Audience:**

Yes

**Audience Explanation:**

The work introduces a new execution framework (FE Net) that rethinks the fundamental message-passing paradigm, which is central to GNN research. Even if not universally applicable, re-examining core assumptions is valuable to the community.

**Claims And Evidence:**

No

**Claims Explanation:**

## Weak Points

1. The FE Net requires sequential activations based on node distance from the origin (flooding and echo phases). Unlike standard MPNNs, which update all nodes in parallel, FE Net has to wait for updates from the previous distance before proceeding.
This introduces an inherent sequential bottleneck, making training and inference slower without being able to benefit from parallelism.

2.  Standard MPNNs allow explicit control over the receptive field by choosing the number of layers (i.e., k-hop interactions).
FE Net, in contrast, implicitly spreads information across the entire graph, meaning all nodes interact implicitly.
This can be harmful for tasks that rely on local structural signals (e.g., community detection, semi-supervised node classification), where over-smoothing or irrelevant long-range dependencies degrade performance.

3. The paper focuses heavily on algorithmic reasoning and extrapolation tasks (SALSA-CLRS, synthetic benchmarks). However, it does not report results on standard semi-supervised node classification datasets (e.g., Cora, Citeseer, PubMed, ogbn-arxiv). Without these comparisons, it remains unclear whether FE Net is useful for mainstream graph learning applications beyond algorithmic reasoning.

4. By introducing an arbitrary origin, FE Net breaks permutation equivariance, a desirable property of GNNs. While this may increase expressivity, it risks inconsistency: predictions may differ depending on the arbitrary choice of origin.

5. The authors frame FE Net as an alternative to deep message passing, but they do not analyze whether it alleviates or worsens issues like over-squashing and over-smoothing.

**Requested Changes:**

1. While the paper emphasizes message complexity advantages, the sequential activation scheme of FE Net introduces a bottleneck that prevents parallelization and could make training and inference slower than standard MPNNs. A clear empirical analysis of runtime performance on modern hardware is necessary to justify the claimed efficiency.

2. Also critical is the absence of results on widely used semi-supervised node classification benchmarks such as Cora, Citeseer, PubMed, or ogbn-arxiv. Without these experiments, it remains unclear whether FE Net is useful beyond algorithmic reasoning tasks, which substantially limits its relevance to mainstream GNN research.

3. The authors should discuss potential ways to control the range of interactions, since FE Net inherently spreads information globally, which may be harmful for tasks requiring local reasoning; experiments illustrating whether the model suffers from over-smoothing in such cases would be valuable.

4. The introduction of an arbitrary origin node breaks permutation equivariance, which is an important property of GNNs. An ablation on different origin selection strategies, and a discussion of how this symmetry breaking affects consistency would improve clarity. Finally, the paper would benefit from either a theoretical or empirical analysis of whether FE Net alleviates or exacerbates known issues such as over-squashing and over-smoothing, as these are central concerns in message-passing GNN design.

---

> ### Author Response · Authors · 2025-09-08
>
> >The FE Net requires sequential activations based on node distance from the origin (flooding and echo phases). Unlike standard MPNNs, which update all nodes in parallel, FE Net has to wait for updates from the previous distance before proceeding. This introduces an inherent sequential bottleneck, making training and inference slower without being able to benefit from parallelism.
>
> > While the paper emphasizes message complexity advantages, the sequential activation scheme of FE Net introduces a bottleneck that prevents parallelization and could make training and inference slower than standard MPNNs. A clear empirical analysis of runtime performance on modern hardware is necessary to justify the claimed efficiency.
>
>
> First of all, the reviewer is correct in its assessment about the tradeoff between the improved message complexity of the method and its sequential computation. In our work, we find several reasons why this tradeoff might be worth doing. First, the resulting alternation of the computation and its algorithmic alignment can yield better results. Second, in the settings we study, it is crucial that information can be incorporated throughout the graph (both to nodes far away during training, but also when generalizing to larger graphs). As a consequence the sequential depth of the MPNN (as in number of rounds) also has to be increased, and actually matches (or even exceeds) the depth of the FE Net asymptotically. This can be seen in the evaluation of 5.1 and its corresponding time measurements in Appendix H2 Table 7. While the GIN does make use of parallel messages and shallow depth, it just can’t achieve good results. Compared to the runtime of the other baselines, FE Net is roughly a constant factor slower (more details on why in H2), but still useful for practical purposes. However, still these baselines achieve worse empirical results in the end. Finally, note that we distinguish efficiency between message complexity and runtime and have a detailed discussion on this, ie. in H1.
>
>
> > Standard MPNNs allow explicit control over the receptive field by choosing the number of layers (i.e., k-hop interactions). FE Net, in contrast, implicitly spreads information across the entire graph, meaning all nodes interact implicitly. This can be harmful for tasks that rely on local structural signals (e.g., community detection, semi-supervised node classification), where over-smoothing or irrelevant long-range dependencies degrade performance.
>
>
>
> We appreciate the input direction, however, we want to emphasize that we propose and study the FE Net to work well in the settings where the entire graph is relevant to the task at hand. As pointed out in other responses, we do not focus on the setting where the task is extremely dependent on local structure as in this case the inductive bias of classical MPNN is very likely to be more beneficial. We could envision that instead of unrolling the flood and echo phases up to the graph diameter, one could instead do this up to depth K, effectively controlling the receptive field. However, this goes into a very different direction of the current focus of the paper. Note that we have an additional experiment however, which partly studies how effectively the FE Net can incorporate information in its receptive field of the FE in Appendix E
>
>
> > The paper focuses heavily on algorithmic reasoning and extrapolation tasks (SALSA-CLRS, synthetic benchmarks). However, it does not report results on standard semi-supervised node classification datasets ...
>
>
> A very similar point was raised by another reviewer, please allow us to group our response and explain why and on what setting we focus on our work. With the FE Net, we present an alternative to classical MPNN based on algorithmic alignment on the level of message passing. We think this is interesting, particularly in the setting we discuss in the paper where information might be far away, or structured computation may be more beneficial for generalization. In these settings the classical MPNN baseline does not do particularly well. However, that does not mean that the FE mechanism is generally the right alternative in any setting (and neither is our intention to claim this). Classical MPNN do provide a good inductive bias for other tasks, where the immediate neighborhood information is most important, as is the case in many of the mentioned datasets. As such we see limited benefit in evaluating under these settings with the current formulation, as it is not at all the motivation of why we desire the algorithmic alignment. We discuss this in the end of section 5.1 (right before 5.2), if need be we could also revise our draft further to make it more explicit. It might be an interesting direction for future work on how to balance the two paradigms. However, we have revised our formulation in the main paper to make this distinction more clear.

---

> > ### Author Response · Authors · 2025-09-08
> >
> > > By introducing an arbitrary origin, FE Net breaks permutation equivariance, a desirable property of GNNs. While this may increase expressivity, it risks inconsistency: predictions may differ depending on the arbitrary choice of origin.
> > The introduction of an arbitrary origin node breaks permutation equivariance, which is an important property of GNNs. An ablation on different origin selection strategies, and a discussion of how this symmetry breaking affects consistency would improve clarity. Finally, the paper would benefit from either a theoretical or empirical analysis of whether FE Net alleviates or exacerbates known issues such as over-squashing and over-smoothing, as these are central concerns in message-passing GNN design.
> > > The authors frame FE Net as an alternative to deep message passing, but they do not analyze whether it alleviates or worsens issues like over-squashing and over-smoothing.
> >
> > Indeed the symmetry breaking affects the equivariance property of the FE Net, which we also point out in the last paragraph of section 4.2. However, note that it is mainly a symmetry breaking that is achieved with the choice of the origin node. If the same node is chosen (as in the fixed variant) the equivariance will be guaranteed. A similar argument can be made for the all variant. However, in the random variant this will only hold in expectation because the nodes are chosen randomly. In Table 8 in the Appendix we provide an empirical measurement on how much the outcomes differ. In the extended related work of Appendix, second paragraph we discuss FE and oversquashing. Because this is closely linked to the original graph topology we doubt that there are significant differences wrt. to common MPNNs. However maybe the sparsity or depth of the computation can still lead to interesting interactions with the mentioned phenomenon and will be interesting to study in future extensions, we thank the reviewer for this suggestion.
> >
> > > Would at least some individuals in TMLR's audience be interested in knowing the findings of this paper?: Yes
> > Explain your answer above:
> > The work introduces a new execution framework (FE Net) that rethinks the fundamental message-passing paradigm, which is central to GNN research. Even if not universally applicable, re-examining core assumptions is valuable to the community.
> >
> > We thank the reviewer for his positive feedback and hope that our proposed mechanism and studied effects can be of value to the TMLR community.

---

### Author Response · Authors · 2025-09-08

We thank the reviewers for their initial assessment and are looking forward to a fruitful discussion of our work. We have also uploaded a new version of the draft that include some of the modifications and clarifications requested.

---

> ### Author Response · Authors · 2025-09-16
>
> Dear Reviewers, it has been more than a week since the initial assessment of the work. We hope you did find our responses, explanations and adjustments helpful. Please let us know If there remain any points you would like to further discuss.
>
> To briefly summarize some of the main discussion points so far:
>
> Complexity of the method: The FE Net is more efficient in terms of message complexity, which is not directly linked to empirical runtime - however, one advantage, ie. on the prefix sum task for which we provided detailed measurements,  when compared with the other baselines is the improved performance as a benefit of the mechanism alignment.
>
> Empirical Evaluation: We extensively test our method on simple and more advanced tasks, where we deem the intended algorithmic alignment to be of interest and benefit. On the other hand, we expect limited direct applicability to tasks such as citation networks, where the immediate neighborhood is most relevant- In these cases the default inductive bias provided by regular MPNNs is more suitable. While a combination, or as suggested a continuous relaxation, between the two approaches could be an interesting future research direction, we think it best to properly evaluate the main intended properties and mechanism of the FE Net. In our experiments we see that the FE Net achieves strong performance (best of 9/15 tasks) on algorithmic tasks without relying on predetermined steps counts or hints during training. Further we can push the achieved graph accuracy even in the regime where graphs are 100 times larger than during training.
>
> Impact: We propose a novel mechanism which introduces algorithmic alignment that has improved message efficiency and due to its aligned computation flow which we verify to be empirically beneficial for extrapolating to larger instances in algorithmic reasoning tasks. Therefore, we think our presented work and insights are of value and interest to the TMLR community.

---

### Decision · Action_Editor_EX93 · 2025-10-10

**Recommendation:** Reject

**Additional Comments:**

All reviewers agreed that the paper is not ready for publication. For a strong resubmission I highly encourage the authors to: add hardware-grounded runtime and scaling studies; include compact but representative results on standard benchmarks; provide direct comparisons to sampling-based methods; quantify sensitivity to origin selection and discuss permutation equivariance; and clarify complexity statements and notation. A brief analysis of over-smoothing and over-squashing would also help.

**Audience:**

Yes

**Audience Explanation:**

Reframing message passing via flood and echo phases is conceptually interesting. This line of work can interest readers studying algorithmic alignment and scaling behavior in GNNs.

**Claims And Evidence:**

No

**Claims Explanation:**

The core claims rely on message complexity, but there is no convincing wall-clock runtime evidence. The evaluation is confined to algorithmic datasets, omits standard GNN benchmarks needed to assess external validity, and the sequential activation (especially the all variant) undercuts the stated efficiency advantage.  Lastly, it is noted by several reviewers that the difference between the proposed method and sampling-based GNNs is not clearly established.

**Resubmission Of Major Revision:**

The authors may consider submitting a major revision at a later time.